# FIXED-BUDGET DIFFERENTIALLY PRIVATE BEST ARM IDENTIFICATION

**Zhirui Chen[1], P. N. Karthik[2,\*] Yeow Meng Chee[1], and Vincent Y. F. Tan[1]**
[1]National University of Singapore   [2]Indian Institute of Technology, Hyderabad
zhiruichen@u.nus.edu   pnkarthik@ai.iith.ac.in   {ymchee,vtan}@nus.edu.sg

## ABSTRACT

We study best arm identification (BAI) in linear bandits in the fixed-budget regime under differential privacy constraints, when the arm rewards are supported on the unit interval. Given a finite budget $T$ and a privacy parameter $\varepsilon > 0$, the goal is to minimise the error probability in finding the arm with the largest mean after $T$ sampling rounds, subject to the constraint that the policy of the decision maker satisfies a certain $\varepsilon$-*differential privacy* ($\varepsilon$-DP) constraint. We construct a policy satisfying the $\varepsilon$-DP constraint (called DP-BAI) by proposing the principle of *maximum absolute determinants*, and derive an upper bound on its error probability. Furthermore, we derive a minimax lower bound on the error probability, and demonstrate that the lower and the upper bounds decay exponentially in $T$, with exponents in the two bounds matching order-wise in (a) the sub-optimality gaps of the arms, (b) $\varepsilon$, and (c) the problem complexity that is expressible as the sum of two terms, one characterising the complexity of standard fixed-budget BAI (without privacy constraints), and the other accounting for the $\varepsilon$-DP constraint. Additionally, we present some auxiliary results that contribute to the derivation of the lower bound on the error probability. These results, we posit, may be of independent interest and could prove instrumental in proving lower bounds on error probabilities in several other bandit problems. Whereas prior works provide results for BAI in the fixed-budget regime without privacy constraints or in the fixed-confidence regime with privacy constraints, our work fills the gap in the literature by providing the results for BAI in the fixed-budget regime under the $\varepsilon$-DP constraint.

## 1 INTRODUCTION

Multi-armed bandit problems (Thompson, 1933) form a class of sequential decision-making problems with applications in fields as diverse as clinical trials, internet advertising, and recommendation systems. The common thread in all these applications is the need to balance *exploration* (learning about the environment) and *exploitation* (making the best decision given current knowledge). The exploration-exploitation trade-off has been studied extensively in the context of regret minimisation, where the goal is to minimise the cumulative difference between the rewards of the actions taken and the best possible action in hindsight; see Lattimore and Szepesvári (2020) and the references therein for an exhaustive list of works on regret minimisation. On the other hand, the *pure exploration* framework, which is the focus of this paper, involves identifying the best arm (action) based on a certain criterion such as the highest mean reward. The pure exploration paradigm has been a subject of rigorous study in the literature, predominantly falling within two overarching regimes: the *fixed-confidence* regime and the *fixed-budget* regime. In the fixed-confidence regime, the objective is to curtail the anticipated number of trials needed to pinpoint the optimal arm, all while adhering to a predefined maximum allowable error probability. Conversely, in the fixed-budget regime, the aim is to suppress the likelihood of erroneous identification of the best arm under a predetermined budget.

**Motivation:** The task of identifying the best arm in a multi-armed bandit setting is non-trivial due to the inherent uncertainty associated with each arm's true reward distribution. This problem is amplified when *privacy* constraints are considered, such as the need to protect individual-level data in a medical trial or user data in an online advertising setting (Chan et al., 2011). In the context of such data-intensive applications, the notion of *differential privacy* (Dwork, 2006) has become the gold-standard for the modelling and analytical study of privacy. While there has been growing

---

\*This work was carried out when P. N. Karthik was a Research Fellow at the National University of Singapore.

interest in the design of privacy-preserving algorithms for regret minimisation in multi-armed bandits (Basu et al., 2019; Jain et al., 2012; Chan et al., 2011; Guha Thakurta and Smith, 2013; Mishra and Thakurta, 2015; Tossou and Dimitrakakis, 2016), a comparable level of attention has not been directed towards the domain of pure exploration. Addressing this lacuna in the literature, our research aims to investigate differentially private best arm identification within the fixed-budget regime.

**Problem Setup:** Briefly, our problem setup is as follows. We consider a multi-armed bandit in which each arm yields independent rewards supported on the unit interval $[0, 1]$. Each arm is associated with a known, $d$-dimensional *feature vector*, where $d$ is potentially much smaller than the number of arms. The *mean* reward of each arm is a *linear* function of the associated feature vector, and is given by the dot product of the feature vector with an unknown $d$-dimensional vector $\boldsymbol{\theta}^*$ which fully specifies the underlying problem instance. Given a designated budget $T$ and a parameter $\varepsilon > 0$, the objective is to minimise the probability of error in identifying the arm with the largest mean reward (best arm), while concurrently fulfilling a certain $\varepsilon$-*differential privacy* ($\varepsilon$-DP) constraint delineated in Basu et al. (2019). We explain the specifics of our model and define the $\varepsilon$-DP constraint formally in Section 2 below.

**Overview of Prior Works:** Differential privacy (DP) (Dwork, 2006) and best-arm identification (BAI) (Lattimore and Szepesvári, 2020) have both been extensively investigated in the literature, encompassing a wide array of works. In this section, we discuss a selection of more recent contributions at the intersection of these two topics. Shariff and Sheffet (2018) prove that any $\varepsilon$-DP (viz. $(\varepsilon, \delta)$-DP with $\delta = 0$) algorithm must incur an additional regret of at least $\Omega\left((K \log T)/\varepsilon\right)$, where $K$ is the number of arms. Building on this result, Sajed and Sheffet (2019) propose an elimination-based algorithm that satisfies the $\varepsilon$-DP constraint and achieves order-wise optimality in the additional regret term. Zheng et al. (2020) study regret minimisation with the $(\varepsilon, \delta)$-local differential privacy constraint, a stronger requirement than $(\varepsilon, \delta)$-DP, for contextual and generalised linear bandits. Azize and Basu (2022) study the $\varepsilon$-global differential privacy constraint for regret minimisation, and provide both minimax and problem-dependent regret bounds for general stochastic bandits and linear bandits. Chowdhury and Zhou (2023) and Solanki et al. (2023) explore differential privacy in a distributed (federated) setting. Chowdhury and Zhou (2023) explore regret minimization with the $(\varepsilon, \delta)$-DP constraint in a distributed setting, considering an untrustworthy server. They derive an upper bound on the regret which matches order-wise with the one obtainable under a centralized setting with a trustworthy server; for a similar work that studies regret minimisation in the distributed and centralised settings, see Hanna et al. (2022). Solanki et al. (2023) study federated learning for combinatorial bandits, considering a slightly different notion of privacy than the one introduced in Dwork (2006). We observe that the existing literature on bandits mainly focused on regret minimisation with DP constraint and the pure exploration counterpart has not been studied extensively.

In the pure exploration domain, Carpentier and Locatelli (2016) study the fixed-budget BAI problem and obtains a minimax lower bound on the error probability; the authors show that their bound is order-wise tight in the exponent of the error probability. Yang and Tan (2022) investigate fixed-budget BAI for linear bandits and propose an algorithm based on the G-optimal design. They prove a minimax lower bound on the error probability and obtain an upper bound on the error probability of their algorithm OD-LINBAI. Despite the significant contributions of Carpentier and Locatelli (2016), Yang and Tan (2022), Komiyama et al. (2022), and Kato et al. (2023), these works do not take into account DP constraints. Nikolakakis et al. (2021) and Rio et al. (2023) study BAI in the fixed-confidence setting with $\varepsilon$-DP constraint and propose successive elimination-type algorithms, but these works do not derive a lower bound that is a function of the privacy parameter $\varepsilon$. Our work is thus the first to study differentially private best arm identification in fixed-budget regime and provide a lower bound explicitly related to the privacy parameter $\varepsilon$.

**Our Contributions:** We present a novel algorithm for fixed-budget BAI under the $\varepsilon$-DP constraint. Our proposed algorithm, called DP-BAI, is based on the principle of maximizing absolute determinants (or MAX-DET in short). A key aspect of our algorithm is the privatisation of the empirical mean of each arm via the addition of Laplacian noise. The amount of noise added to an arm is inversely proportional to the product of the privacy parameter $\varepsilon$ and the number of times the arm is pulled. Recognising the trade-off between the number of arm pulls and the level of noise injected for privatisation, the MAX-DET principle minimises the maximum Laplacian noise injected across all arms, thereby ensuring a small probability of error in identifying the best arm. We believe our work can open for future exploration in precise control over Laplacian noise (crucial to meet the

$\varepsilon$-DP guarantee) and using other popular techniques in fixed-budget BAI, such as G-optimal designs (Kiefer and Wolfowitz, 1960; Yang and Tan, 2022) and $\mathcal{X}\mathcal{Y}$-adaptive allocations (Soare et al., 2014), with DP-constraint. We find it analytically convenient to leverage the properties of the MAX-DET collection (cf. Definition 3.1) to satisfy the $\varepsilon$-DP constraint. See Remark 2 for a brief justification on why extending other popular techniques for fixed-budget BAI such as G-optimal design (Yang and Tan, 2022) to the differential privacy setting of our work is not readily feasible.

Additionally, we establish the first-known lower bound on the error probability under the $\varepsilon$-DP constraint for a class of "hard" problem instances. We demonstrate that both the upper and lower bounds decay exponentially relative to the budget $T$. The exponents in these bounds capture the problem complexity through a certain hardness parameter, which we show can be expressed as the sum of two terms: one measuring the complexity of the standard fixed-budget BAI without privacy constraints, and the other accounting for the $\varepsilon$-DP constraint. We also present some auxiliary findings, such as the properties of the so-called *early stopping* version of a BAI policy (see Lemmas 4.3 and 4.5), that contribute to the derivation of the lower bound, which may be of independent interest and could prove instrumental in deriving lower bounds on error probabilities in several other bandit problems. Our work stands out as the first in the field to provide precise and tight bounds on the error probability for fixed-budget BAI under the $\varepsilon$-DP constraint, achieving order-wise optimal exponents in both the lower and the upper bounds.

## 2 NOTATIONS AND PRELIMINARIES

Consider a multi-armed bandit with $K > 2$ arms, in which each arm yields independent and identically distributed (i.i.d.) rewards, and the rewards are statistically independent across arms. Let $[K] := \{1, \ldots, K\}$ denote the set of arms. For $i \in [K]$, let $\nu_i$ denote the rewards distribution of arm $i$. As in several prior works (Chowdhury and Zhou, 2022; Shariff and Sheffet, 2018; Zhou and Chowdhury, 2023), we assume throughout the paper that $\nu_i$ is supported in $[0, 1]$ for all $i \in [K]$. We impose a *linear* structure on the mean rewards of the arms. That is, for each $i \in [K]$, we assume that arm $i$ is associated with a *feature vector* $\mathbf{a}_i \in \mathbb{R}^d$, where $d$ is the dimension of the feature vector, and the mean reward of arm $i$ is given by $\mu_i := \mathbf{a}_i^\top \boldsymbol{\theta}^*$ for some fixed and unknown $\boldsymbol{\theta}^* \in \mathbb{R}^d$. We assume that the feature vectors of the arms $\{\mathbf{a}_i\}_{i=1}^K$ are known beforehand to a decision maker, whose goal it is to identify the best arm $i^* = \text{argmax}_{i \in [K]} \mu_i$; we assume that the best arm is unique and defined unambiguously.

**The Fixed-Budget Regime:** The decision maker is allowed to pull the arms sequentially, one at each time $t \in \{1, 2, \ldots\}$. Let $A_t \in [K]$ denote the arm pulled by the decision maker at time $t$, and let $N_{i,t} = \sum_{s=1}^t \mathbf{1}_{\{A_s = i\}}$ denote the number of times arm $i$ is pulled up to time $t$. Upon pulling arm $A_t$, the decision maker obtains the instantaneous reward $X_{A_t, N_{A_t, t}} \in [0, 1]$; here, $X_{i,n} \sim \nu_i$ denotes the reward obtained on the $n$th pull of arm $i$. Notice that $\mathbb{E}[X_{i,n}] = \mu_i = \mathbf{a}_i^\top \boldsymbol{\theta}^*$ for all $i \in [K]$ and $n \geq 1$. For all $t$, the decision to pull arm $A_t$ is based on the history of arm pulls and rewards seen up to time $t$, i.e., $A_t$ is a (random) function of $\mathcal{H}_t := (A_1, X_{A_1, N_{A_1}, 1}, \ldots, A_{t-1}, X_{A_{t-1}, N_{A_{t-1}, t-1}})$. Given a fixed *budget* $T < \infty$, the objective of the decision maker is to minimise the probability of error in finding the best arm after $T$ rounds of arm pulls, while also satisfying a certain *differential privacy* constraint outlined below. We let $\hat{I}_T$ denote the best arm output by the decision maker.

**The $\varepsilon$-Differential Privacy Constraint:** Let $\mathcal{X} := \{\mathbf{x} = (x_{i,t})_{i \in [K], t \in [T]}\} \subseteq [0, 1]^{KT}$ denote the collection of all possible rewards outcomes from the arms. Any sequential arm selection *policy* of the decision maker may be viewed as taking inputs from $\mathcal{X}$ and producing $(A_1, \ldots, A_T, \hat{I}_T) \in [K]^{T+1}$ as outputs in the following manner: for an input $\mathbf{x} = (x_{i,t}) \in \mathcal{X}$,

$$\text{Output at time } t = 1 : A_1 = A_1,$$
$$\text{Output at time } t = 2 : A_2 = A_2(A_1, x_{A_1, N_{A_1}, 1}),$$
$$\text{Output at time } t = 3 : A_3 = A_3(A_1, x_{A_1, N_{A_1}, 1}, A_2, x_{A_2, N_{A_2}, 2}),$$
$$\vdots$$
$$\text{Output at time } t = T : A_T = A_T(A_1, x_{A_1, N_{A_1}, 1}, \ldots, A_{T-1}, x_{N_{A_{T-1}}, T-1}),$$
$$\text{Terminal output} : \hat{I}_T = \hat{I}_T(A_1, x_{A_1, N_{A_1}, 1}, \ldots, A_T, x_{N_{A_T}, T}). \tag{1}$$

We say that $\mathbf{x} = (x_{i,t})$ and $\mathbf{x}' = (x'_{i,t})$ are *neighbouring* if they differ in exactly one location, i.e., there exists $(i,t) \in [K] \times [T]$ such that $x_{i,t} \neq x'_{i,t}$ and $x_{j,s} = x'_{j,s}$ for all $(j,s) \neq (i,t)$. With the viewpoint in (1), we now introduce the notion of $\varepsilon$-*differential privacy* for a sequential policy of the decision maker, following the lines of Nikolakakis et al. (2021, Section 5).

**Definition 2.1.** Given any $\varepsilon > 0$, a randomised policy $\mathcal{M} : \mathcal{X} \to [K]^{T+1}$ satisfies $\varepsilon$-*differential privacy* if, for any pair of neighbouring $\mathbf{x}, \mathbf{x}' \in \mathcal{X}$,
$$\mathbb{P}^{\mathcal{M}}(\mathcal{M}(\mathbf{x}) \in \mathcal{S}) \leq e^{\varepsilon} \, \mathbb{P}^{\mathcal{M}}(\mathcal{M}(\mathbf{x}') \in \mathcal{S}) \quad \forall \mathcal{S} \subset [K]^{T+1}. \tag{2}$$

*Remark* 1. A generalization of the notion of $\varepsilon$-differential privacy is that of $(\varepsilon, \delta)$-differential privacy (Dwork et al., 2014, Chapter 2). For the sake of simplicity in exposition, in the main text, we provide details for the former. An extension of our algorithm, called DP-BAI-GAUSS, will be shown to be applicable to the latter (generalized) notion of differential privacy. The details and accompanying analyses of the performance of DP-BAI-GAUSS can be found in Appendix D.

While the actual sequence of rewards observed under $\mathcal{M}$ is random, it is important to note that a pair of reward sequences, say $(\mathbf{x}, \mathbf{x}')$, is fixed when specifying the $\varepsilon$-DP constraint. In (2), $\mathbb{P}^{\mathcal{M}}$ denotes the probability measure induced by the randomness arising from only the arm outputs under $\mathcal{M}$. In the sequel, we refer to the tuple $v = ((\mathbf{a}_i)_{i \in [K]}, (\nu_i)_{i \in [K]}, \boldsymbol{\theta}^*, \varepsilon)$ as a *problem instance*, and let $\mathcal{P}$ denote to be the set of all problem instances that admit a unique best arm. Given $v \in \mathcal{P}$ and a policy $\pi$, we write $\mathbb{P}_v^{\pi}$ to denote the probability measure induced under $\pi$ and under the instance $v$. When the dependence on $v$ is clear from the context, we simply write $\mathbb{P}^{\pi}$.

## 3 OUR METHODOLOGY

To meet the $\varepsilon$-DP guarantee, our approach is to add Laplacian noise to the empirical mean reward of each arm, with the magnitude of the noise inversely proportional to the product of $\varepsilon$ and the number of times the arm is pulled. Intuitively, to minimize the maximum Laplacian noise that is added (so as to minimize the failure probability of identifying the best arm), we aim to balance the number of pulls for each arm in the current active set. To this end, we employ the MAX-DET explained below.

**The MAX-DET Collection:** Fix $d' \in \mathbb{N}$. For any set $\mathcal{S} \subset \mathbb{R}^{d'}$ with $|\mathcal{S}| = d'$ vectors, each of length $d'$, let $\text{DET}(\mathcal{S})$ to denote the absolute value of the determinant of the $d' \times d'$ matrix formed by stacking the vectors in $\mathcal{S}$ as the columns of the matrix.

**Definition 3.1.** Fix $d' \in \mathbb{N}$. Given any finite set $\mathcal{A} \subset \mathbb{R}^{d'}$ with $|\mathcal{A}| \geq d'$, we say $\mathcal{B} \subset \mathcal{A}$ with $|\mathcal{B}| = d'$ is a MAX-DET *collection of* $\mathcal{A}$ if
$$\text{DET}(\mathcal{B}) \geq \text{DET}(\mathcal{B}') \quad \text{for all } \mathcal{B}' \subset \mathcal{A} \text{ with } |\mathcal{B}'| = d'. \tag{3}$$

Thus, a MAX-DET collection $\mathcal{B} \subset \mathcal{A}$ has the *maximum absolute determinant* among all subsets of $\mathcal{A}$ with the same cardinality as $\mathcal{B}$. If $\text{span}(\mathcal{A}) = d'$, the vectors in $\mathcal{B}$ are linearly independent, and any $\mathbf{b} \in \mathcal{A}$ may be expressed as a linear combination of the vectors in $\mathcal{B}$. Call the coefficients appearing in this linear combination expression for $\mathbf{b}$ as its *coordinates* (Meyer, 2000, Chapter 4). The set of coordinates of each $\mathbf{b} \in \mathcal{A}$ is unique, and $\mathbf{b}$ may be expressed alternatively as a $d'$-length vector of its coordinates. In this new system of coordinates, the vectors in $\mathcal{B}$ constitute the standard basis vectors.

### 3.1 THE DIFFERENTIALLY PRIVATE BEST ARM IDENTIFICATION (DP-BAI) POLICY

We now construct a policy based on the idea of successive elimination (SE) of arms. Our policy for Differentially Private Best Arm Identification, called DP-BAI, operates over a total of $M$ *phases*, where $M$ is designed to have order $O(\log d)$. In each phase $p \in [M]$, the policy maintains an *active set* $\mathcal{A}_p$ of arms which are potential contenders for emerging as the best arm. The policy ensures that with high probability, the true best arm lies within the active set in each phase.

**Policy-Specific Notations:** We now introduce some policy-specific notations. Let
$$\lambda = \inf\{\beta \geq 2 : \beta^{\log(d)} \geq K - \lceil d^2/4 \rceil\}, \tag{4}$$
Let $\{g_i\}_{i \geq 0}$ and $\{h_i\}_{i \geq 0}$ be defined as follows:
$$g_0 = \min\{K, \lceil d^2/4 \rceil\}, \qquad g_i = \lceil g_{i-1}/2 \rceil \qquad \forall i \geq 1, \tag{5}$$
$$h_0 = \max\{K - \lceil d^2/4 \rceil, 0\}, \quad h_i = \lceil (h_{i-1}+1)/\lambda \rceil - 1 \quad \forall i \geq 1. \tag{6}$$

Let $s_0 = g_0 + h_0$, and for each $p \in [M]$, let $s_p = |\mathcal{A}_p|$ denote the number of active arms at the beginning of phase $p$, defined via

$$s_p = \begin{cases} g_0 + h_{p-1}, & 1 \leq p \leq M_1, \\ g_{p-M_1}, & M_1 < p \leq M+1. \end{cases} \tag{7}$$

For $\alpha > 0$, let $\mathrm{Lap}\left(\frac{1}{\alpha}\right)$ denote the Laplacian distribution with density $f_\alpha(z) = \frac{\alpha}{2} e^{-\alpha |z|}$, $z \in \mathbb{R}$.

**Initialisation:** We initialise our policy with the following parameters:

$$M_1 = \min\{i \in \mathbb{N} : h_i = 0\}, \qquad M = M_1 + \min\{i \in \mathbb{N} : g_i = 1\} - 1,$$
$$T' = T - M_1 d - (M - M_1)\lceil d^2/4 \rceil, \qquad \mathbf{a}_i^{(0)} = \mathbf{a}_i \ \forall\, i \in [K],$$
$$d_0 = d, \quad T_0 = 0, \qquad\qquad\qquad\qquad \mathcal{A}_1 = [K]. \tag{8}$$

**Policy Description:** We now describe the DP-BAI policy. The policy takes as inputs the differential privacy parameter $\varepsilon$, budget $T$, the number of arms $K$, and the feature vectors of the arms $\{\mathbf{a}_i : i \in [K]\}$. With the initialisation in (8), the policy operates in *phases*. In each phase $p \in [M]$, the first step is *dimensionality reduction* (Yang and Tan, 2022), whereby the dimension of the set of vectors $\{\mathbf{a}_i^{(p-1)} : i \in \mathcal{A}_p\}$ is reduced using a linear transformation; here, $\mathbf{a}_i^{(p-1)} \in \mathbb{R}^{d_{p-1}}$ for all $i \in \mathcal{A}_p$. More specifically, suppose that $d_p := \dim(\mathrm{span}\{\mathbf{a}_i^{(p-1)} : i \in \mathcal{A}_p\})$. The policy chooses an arbitrary orthogonal basis $\mathcal{U}_p = (\mathbf{u}_1^{(p)}, \ldots, \mathbf{u}_{d_p}^{(p)})$ for $\mathrm{span}\{\mathbf{a}_i^{(p-1)} : i \in \mathcal{A}_p\}$, and obtains a new set of vectors

$$\mathbf{a}_i^{(p)} := [\mathbf{a}_i^{(p-1)}]_{\mathcal{U}_p}, \quad \text{for all} \quad i \in \mathcal{A}_p, \tag{9}$$

where $[\mathbf{v}]_{\mathcal{U}_p}$ denotes the coordinates of $\mathbf{v}$ with respect to $\mathcal{U}_p$. Subsequently, the policy checks if $d_p < \sqrt{s_p}$, where $s_p = |\mathcal{A}_p|$ is as defined in (7). If this is true, then the policy constructs a MAX-DET collection $\mathcal{B}_p \subset \mathcal{A}_p$ consisting of $|\mathcal{B}_p| = d_p$ arms, and pulls each arm $i \in \mathcal{B}_p$ for $\lceil \frac{T'}{M d_p} \rceil$ many times, and sets $T_p = T_{p-1} + d_p \lceil \frac{T'}{M d_p} \rceil$. On the other hand, if $d_p \geq \sqrt{s_p}$, then the policy pulls each arm in $\mathcal{A}_p$ for $\lceil \frac{T'}{M s_p} \rceil$ many times, and sets $T_p = T_{p-1} + s_p \lceil \frac{T'}{M s_p} \rceil$. After pulling the arms according to the preceding rule, the policy computes

$$\hat{\mu}_i^{(p)} = \frac{1}{N_{i,T_p} - N_{i,T_{p-1}}} \sum_{s=N_{i,T_{p-1}}+1}^{N_{i,T_p}} X_{i,s} \tag{10}$$

for each arm $i \in \mathcal{A}_p$ that was pulled at least once in phase $p$, and subsequently computes its *private empirical mean* $\widetilde{\mu}_i^{(p)}$ via

$$\widetilde{\mu}_i^{(p)} = \hat{\mu}_i^{(p)} + \widetilde{\xi}_i^{(p)}, \tag{11}$$

where $\widetilde{\xi}_i^{(p)} \sim \mathrm{Lap}\left(\frac{1}{(N_{i,T_p} - N_{i,T_{p-1}})\varepsilon}\right)$ is independent of the arm pulls and arm rewards. For $i \in \mathcal{A}_p$ that was not pulled in phase $p$, the policy computes its corresponding private empirical mean via

$$\widetilde{\mu}_i^{(p)} = \sum_{j \in \mathcal{B}_p} \alpha_{i,j}\, \widetilde{\mu}_j^{(p)}, \tag{12}$$

where $(\alpha_{i,j})_{j \in \mathcal{B}_p}$ is the unique set of coefficients such that $\mathbf{a}_i^{(p)} = \sum_{j \in \mathcal{B}_p} \alpha_{i,j}\, \mathbf{a}_j^{(p)}$. At the end of phase $p$, the policy retains only the top $s_{p+1}$ arms with the largest private empirical means and eliminates the remaining arms; intuitively, these arms are most likely to produce the highest rewards in the subsequent phases. At the end of the $M$th phase, the policy returns the only arm left in $\mathcal{A}_{M+1}$ as the best arm. For pseudo-code of the DP-BAI policy, see Algorithm 1.

*Remark* 2. It is natural to wonder why we do not devise a differentially private version of OD-LinBAI (Yang and Tan, 2022), the state-of-the-art linear fixed-budget BAI algorithm, which uses G-optimal designs. A proposal to do so, called DP-OD, is provided in Appendix E. However, the error probability in identifying the best arm under DP-OD depends not only on the suboptimality gaps of the arms, but is *also* a function of the *arm vectors*. For example, in a 2-armed bandit instance, let $\mathbf{a}_1 = [x, 0]^\top$, $\mathbf{a}_2 = [0, y]^\top$ with $x, y > 0$, and $\theta^* = [(0.5+\Delta)/x, \ 0.5/y]^\top$. Then, $\mu_1 = 0.5 + \Delta$, $\mu_2 = 0.5$, and the suboptimality gap $\Delta = \mu_1 - \mu_2$. For this instance, the upper bound on the error probability of DP-OD is $\exp\left(-\Omega\left(\frac{T}{\Delta^{-2} + \frac{x \vee y}{x \wedge y}(\epsilon\Delta)^{-1}}\right)\right)$. We observe that $\frac{x \vee y}{x \wedge y}$ can be made arbitrarily large. Thus, this bound is inferior to the upper bound of DP-BAI (equal to $\exp(-\Omega(\frac{T}{\Delta^{-2} + (\epsilon\Delta)^{-1}}))$ and independent of the arm vectors). See Appendix E for further details.

---

**Algorithm 1** Fixed-Budget Differentially Private Best Arm Identification (DP-BAI)

---

**Input:**
$\quad\varepsilon$: differential privacy parameter; $T$: budget; $\{\mathbf{a}_i : i \in [K]\}$: $d$-dimensional feature vectors.

**Output:** $\hat{I}_T$: best arm.

1: Initialise $T_0 = 0$, $\mathcal{A}_1 = [K]$, $\mathbf{a}_i^{(0)} = \mathbf{a}_i$ for all $i \in [K]$. Set $M$ and $T'$ as in (8).
2: **for** $p \in \{1, 2, \ldots, M\}$ **do**
3: $\quad$ Set $d_p = \dim(\mathrm{span}\{\mathbf{a}_i^{(p-1)} : i \in \mathcal{A}_p\})$.
4: $\quad$ Obtain the new vector set $\{\mathbf{a}_i^{(p)} : i \in \mathcal{A}_p\}$ from the set $\{\mathbf{a}_i^{(p-1)} : i \in \mathcal{A}_p\}$ via (9).
5: $\quad$ Compute $s_p$ using (7).
6: $\quad$ **if** $d_p < \sqrt{s_p}$ **then**
7: $\quad\quad$ Construct a MAX-DET collection $\mathcal{B}_p \subset \mathcal{A}_p$.
8: $\quad\quad$ Pull each arm in $\mathcal{B}_p$ for $\lceil \frac{T'}{Md_p} \rceil$ many times. Update $T_p \leftarrow T_{p-1} + d_p \lceil \frac{T'}{Md_p} \rceil$.
9: $\quad\quad$ Obtain the empirical means $\{\hat{\mu}_i(p) : i \in \mathcal{B}_p\}$ via (10).
10: $\quad\quad$ Generate $\widetilde{\xi}_i^{(p)} \sim \mathrm{Lap}\left( \frac{1}{\varepsilon \lceil \frac{T'}{Md_p} \rceil} \right)$ for $i \in \mathcal{B}_p$.
11: $\quad\quad$ Set $\widetilde{\mu}_i^{(p)} \leftarrow \hat{\mu}_i^{(p)} + \widetilde{\xi}_i^{(p)}$ for all $i \in \mathcal{B}_p$.
12: $\quad\quad$ For arm $i \in \mathcal{A}_p \setminus \mathcal{B}_p$, compute $\widetilde{\mu}_i^{(p)}$ via (12).
13: $\quad$ **else**
14: $\quad\quad$ Pull each arm in $\mathcal{A}_p$ for $\lceil \frac{T'}{Ms_p} \rceil$ many times. Update $T_p \leftarrow T_{p-1} + s_p \lceil \frac{T'}{Ms_p} \rceil$
15: $\quad\quad$ Obtain the empirical means $\{\hat{\mu}_i^{(p)} : i \in \mathcal{A}_p\}$ via (10).
16: $\quad\quad$ Generate $\widetilde{\xi}_i^{(p)} \sim \mathrm{Lap}\left( \frac{1}{\varepsilon \lceil \frac{T'}{Ms_p} \rceil} \right)$ for $i \in \mathcal{A}_p$.
17: $\quad\quad$ Set $\widetilde{\mu}_i^{(p)} \leftarrow \hat{\mu}_i^{(p)} + \widetilde{\xi}_i^{(p)}$ for all $i \in \mathcal{A}_p$.
18: $\quad$ **end if**
19: $\quad$ Compute $s_{p+1}$ using (7).
20: $\quad$ $\mathcal{A}_{p+1} \leftarrow$ the set of $s_{p+1}$ arms with largest private empirical means among $\{\widetilde{\mu}_i^{(p)} : i \in \mathcal{A}_p\}$.
21: **end for**
22: $\hat{I}_T \leftarrow$ the only arm remaining in $\mathcal{A}_{M+1}$
23: **return** Best arm $\hat{I}_T$.

---

## 4 THEORETICAL RESULTS

We now present theoretical results for the DP-BAI policy, followed by a minimax lower bound on the error probability. We write $\Pi_{\text{DP-BAI}}$ to denote the DP-BAI policy symbolically. The first result below, proved in Appendix F, asserts that $\Pi_{\text{DP-BAI}}$ meets the $\varepsilon$-DP constraint for any $\varepsilon > 0$.

**Proposition 4.1.** *The* DP-BAI *policy with privacy and budget parameters* $(\varepsilon, T)$ *satisfies the $\varepsilon$-DP constraint, i.e., for any pair of neighbouring* $\mathbf{x}, \mathbf{x}' \in \mathcal{X}$,

$$\mathbb{P}^{\Pi_{\text{DP-BAI}}}(\Pi_{\text{DP-BAI}}(\mathbf{x}) \in \mathcal{S}) \leq e^\varepsilon \, \mathbb{P}^{\Pi_{\text{DP-BAI}}}(\Pi_{\text{DP-BAI}}(\mathbf{x}') \in \mathcal{S}) \quad \forall \mathcal{S} \subset [K]^{T+1}. \tag{13}$$

In (13), the probabilities appearing on either sides of (13) are with respect to the randomness in the arms output by $\Pi_{\text{DP-BAI}}$ for *fixed* neighbouring reward sequences $\mathbf{x}, \mathbf{x}' \in \mathcal{X}$ (see Section 2). The use of Laplacian noise for the privatisation of the empirical means of the arms (see Lines 10-11 and 16-17 in Algorithm 1) plays a crucial role in showing (13).

### 4.1 THE HARDNESS PARAMETER

Recall that a problem instance $v$ may be expressed as the tuple $v = ((\mathbf{a}_i)_{i \in [K]}, (\nu_i)_{i \in [K]}, \boldsymbol{\theta}^*, \varepsilon)$. In this section, we capture the hardness of such an instance in terms of the instance-specific arm sub-optimality gaps and the privacy parameter $\varepsilon$. Recall that the arm means under the above instance $v$ are given by $\mu_i = \mathbf{a}_i^\top \boldsymbol{\theta}^*$ for all $i \in [K]$. Let $\Delta_i := \mu_{i^*(v)} - \mu_i$ denote the sub-optimality gap of arm $i \in [K]$. Further, let $(l_1, \ldots, l_K)$ be a permutation of $[K]$ such that $\Delta_{l_1} \leq \Delta_{l_2} \leq \ldots \leq \Delta_{l_K}$, and let $\Delta_{(i)} := \Delta_{l_i}$ for all $i \in [K]$. The *hardness* of instance $v$ is defined as

$$H(v) := H_{\text{BAI}}(v) + H_{\text{pri}}(v), \tag{14}$$

where

$$H_{\text{BAI}}(v) := \max_{2 \le i \le (d^2 \wedge K)} \frac{i}{\Delta_{(i)}^2} \quad \text{and} \quad H_{\text{pri}}(v) := \frac{1}{\varepsilon} \cdot \max_{2 \le i \le (d^2 \wedge K)} \frac{i}{\Delta_{(i)}}. \tag{15}$$

Going forward, we omit the dependence of $H, H_{\text{BAI}}$, and $H_{\text{pri}}$ on $v$ for notational brevity. It is worthwhile to mention here the quantity in (14) specialises to the hardness term "$H_2$" in Audibert et al. (2010), which is identical to $H_{\text{BAI}}$, when $K \le d^2$ and $\varepsilon \to +\infty$. The former condition $K \le d^2$ holds, for instance, for a standard $K$-armed bandit with $K = d$, $\boldsymbol{\theta}^* \in \mathbb{R}^d$ as the vector of arm means, and $\{\mathbf{a}_i\}_{i=1}^d$ as the standard basis vectors in $\mathbb{R}^d$. Intuitively, while $H_{\text{BAI}}$ quantifies the difficulty of fixed-budget BAI without privacy constraints, $H_{\text{pri}}$ accounts for the $\varepsilon$-DP constraint and captures the additional difficulty of BAI under this constraint.

## 4.2 Upper Bound on the Error Probability of DP-BAI

In this section, we provide an upper bound on the error probability of DP-BAI.

**Theorem 4.2.** *Fix $v \in \mathcal{P}$. Let $i^*(v)$ denote the unique best arm of instance $v$. For all sufficiently large $T$, the error probability of $\Pi_{\text{DP-BAI}}$ with budget $T$ and privacy parameter $\varepsilon$ satisfies*

$$\mathbb{P}_v^{\Pi_{\text{DP-BAI}}}(\hat{I}_T \ne i^*(v)) \le \exp\left(-\frac{T'}{65\,M\,H}\right), \tag{16}$$

*where $M$ and $T'$ are as defined in* (8). *In* (16), *$\mathbb{P}_v^{\Pi_{\text{DP-BAI}}}$ denotes the probability measure induced by $\Pi_{\text{DP-BAI}}$ under the instance $v$.*

This is proved in Appendix G. Since $M = \Theta(\log d)$ and $T' = \Theta(T)$ (as $T \to \infty$), (16) implies that

$$\mathbb{P}_v^{\Pi_{\text{DP-BAI}}}(\hat{I}_T \ne i^*(v)) = \exp\left(-\Omega\left(\frac{T}{H \log d}\right)\right). \tag{17}$$

## 4.3 Lower Bound on the Error Probability

In this section, we derive the first-of-its-kind lower bound on the error probability of fixed-budget BAI under the $\varepsilon$-DP constraint. Towards this, we first describe an *auxiliary* version of a generic policy that takes as input three arguments–a generic policy $\pi$, $n \in \mathbb{N}$, and $\iota \in [K]$–and pulls an auxiliary arm (arm 0) whenever arm $\iota$ is pulled $n$ or more times under $\pi$. We believe that such auxiliary policies are potentially instrumental in deriving lower bounds on error probabilities in other bandit problems.

**The Early Stopping Policy:** Suppose that the set of arms $[K]$ is augmented with an auxiliary arm (*arm* 0) which yields reward 0 each time it is pulled; recall that the arm rewards are supported in $[0, 1]$. Given a generic policy $\pi$, $n \in \mathbb{N}$ and $\iota \in [K]$, let $\text{ES}(\pi, n, \iota)$ denote the *early stopping* version of $\pi$ with the following sampling and recommendation rules.

- **Sampling rule:** given a realization $\mathcal{H}_{t-1} = (a_1, x_1, \ldots, a_{t-1}, x_{t-1})$, if $\sum_{s=1}^{t-1} \mathbf{1}_{\{a_s = \iota\}} < n$, then

$$\mathbb{P}^{\text{ES}(\pi,n,\iota)}(A_t \in \mathcal{A} \mid \mathcal{H}_{t-1}) = \mathbb{P}^\pi(A_t \in \mathcal{A} \mid \mathcal{H}_{t-1}) \quad \forall \mathcal{A} \subseteq [K], \tag{18}$$

and if $\sum_{s=1}^{t-1} \mathbf{1}_{\{a_s = \iota\}} \ge n$, then $\mathbb{P}^{\text{ES}(\pi,n,\iota)}(A_t = 0 \mid \mathcal{H}_{t-1}) = 1$. That is, as long as arm $\iota$ is pulled for a total of fewer than $n$ times, the sampling rule of $\text{ES}(\pi, n, \iota)$ is identical to that of $\pi$. Else, $\text{ES}(\pi, n, \iota)$ pulls arm $\iota$ with certainty.

- **Recommendation rule:** Given history $\mathcal{H}_T = (a_1, x_1, \ldots, a_T, x_T)$, if $\sum_{s=1}^T \mathbf{1}_{\{a_s = 0\}} = 0$, then

$$\mathbb{P}^{\text{ES}(\pi,n,\iota)}(\hat{I}_T \in \mathcal{A} \mid \mathcal{H}_T) = \mathbb{P}^\pi(\hat{I}_T \in \mathcal{A} \mid \mathcal{H}_T) \quad \forall \mathcal{A} \subseteq [K], \tag{19}$$

and if $\sum_{s=1}^T \mathbf{1}_{\{a_s = 0\}} > 0$, then $\mathbb{P}^{\text{ES}(\pi,n,\iota)}(\hat{I}_T = 0 \mid \mathcal{H}_T) = 1$. That is, if the auxiliary arm 0 is not pulled under $\pi$, the recommendation of $\text{ES}(\pi, n, \iota)$ is consistent with that of $\pi$. Else, $\text{ES}(\pi, n, \iota)$ recommends arm 0 as the best arm.

The next result below provides a "bridge" between a policy $\pi$ and its early stopped version.

**Lemma 4.3.** *Fix a problem instance $v \in \mathcal{P}$, policy $\pi$, $n \in \mathbb{N}$, and $\iota \in [K]$. For any $\mathcal{A} \subseteq [K]$ and $E = \{\hat{I}_T \in \mathcal{A}\} \cap \{N_{\iota,T} < n\}$,*

$$\mathbb{P}_v^\pi(E) = \mathbb{P}_v^{\text{ES}(\pi,n,\iota)}(E). \tag{20}$$

In addition, let $\mathcal{X}^{(n,\iota)} := \{(x_{i,t})_{i\in[K],t\in[n_i]} : (x_{i,t})_{i\in[K],t\in[T]} \in \mathcal{X}\} \subseteq \mathbb{R}^{n_1} \times \ldots \times \mathbb{R}^{n_K}$, where $n_i = T$ for all $i \neq \iota$ and $n_\iota = n$. Notice that Definition 2.1 readily extends to any randomised policy that maps $\mathcal{X}^{(n,\iota)}$ to $\{0,\ldots,K\}^{T+1}$. We then have the following corollary to Lemma 4.3.

**Corollary 4.4.** *If* $\pi : \mathcal{X} \to [K]^{T+1}$ *meets the* $\varepsilon$*-DP constraint, then* $\mathrm{ES}(\pi,n,\iota) : \mathcal{X}^{(n,\iota)} \to \{0,\ldots,K\}^{T+1}$ *also meets the* $\varepsilon$*-DP constraint.*

Given the early stopping version of a policy $\pi$, the following lemma provides a "bridge" between two problem instances $v, v' \in \mathcal{P}$.

**Lemma 4.5.** *Fix a policy* $\pi$, $n \in \mathbb{N}$, $\iota \in [K]$, *and* $\varepsilon > 0$, *and suppose that* $\mathcal{M} = \mathrm{ES}(\pi,n,\iota)$ *satisfies the* $\varepsilon$*-DP constraint with respect to* $\mathcal{X}^{(n,\iota)}$. *For any pair of instances* $v = ((\mathbf{a}_i)_{i\in[K]}, (\nu_i)_{i\in[K]}, \boldsymbol{\theta}^*, \varepsilon)$ *and* $v' = ((\mathbf{a}_i)_{i\in[K]}, (\nu'_i)_{i\in[K]}, \boldsymbol{\theta'}^*, \varepsilon)$, *with* $\boldsymbol{\theta}^* \neq \boldsymbol{\theta'}^*$, $\nu_\iota \neq \nu'_\iota$, *and* $\nu_i = \nu'_i$ *for all* $i \neq \iota$, *we have*

$$\mathbb{P}_v^{\mathcal{M}}\big(\mathcal{M}((X_{i,j})_{i\in[K],j\in[n_i]}) \in \mathcal{S}\big) \leq e^{\varepsilon'} \mathbb{P}_{v'}^{\mathcal{M}}\big(\mathcal{M}((X_{i,j})_{i\in[K],j\in[n_i]}) \in \mathcal{S}\big) \quad \forall \mathcal{S} \subseteq \{0,\ldots,K\}^{T+1} \tag{21}$$

*where in* (21), *(i)* $\varepsilon' = 6\varepsilon n\, \mathrm{TV}(v_\iota, v'_\iota)$, *with* $\mathrm{TV}(v_\iota, v'_\iota)$ *being the total variation distance between the distributions* $\nu_\iota$ *and* $\nu'_\iota$, *and (ii)* $n_i = T$ *for all* $i \neq \iota$ *and* $n_\iota = n$.

The proof of Lemma 4.5 follows exactly along the lines of the proof of Karwa and Vadhan (2018, Lemma 6.1) and is omitted. Leveraging Lemma 4.5 in conjunction with Lemma 4.3 provides us with a *change-of-measure* technique, facilitating the transition from $\mathbb{P}_v^\pi$ to $\mathbb{P}_{v'}^\pi$ under any given policy $\pi$. This change-of-measure technique serves as the foundation that enables us to derive the subsequent minimax lower bound on the error probability.

**Definition 4.6.** A policy $\pi$ for fixed-budget BAI is said to be *consistent* if

$$\lim_{T\to+\infty} \mathbb{P}_v^\pi\big(\hat{I}_T \neq i^*(v)\big) = 0, \quad \forall v \in \mathcal{P}. \tag{22}$$

**Theorem 4.7** (Lower Bound). *Fix any* $\beta_1, \beta_2, \beta_3 \in [0,1]$ *with* $\beta_1 + \beta_2 + \beta_3 < 3$, *a consistent policy* $\pi$, *and a constant* $c > 0$. *For all sufficiently large* $T$, *there exists an instance* $v \in \mathcal{P}$ *such that*

$$\mathbb{P}_v^\pi\big(\hat{I}_T \neq i^*(v)\big) > \exp\left(-\frac{T}{c(\log d)^{\beta_1}(H_{\mathrm{BAI}}(v)^{\beta_2} + H_{\mathrm{pri}}(v)^{\beta_3})}\right). \tag{23}$$

*Consequently,*

$$\inf_{\pi \text{ consistent}} \liminf_{T\to+\infty} \sup_{v\in\mathcal{P}} \left\{ \mathbb{P}_v^\pi\big(\hat{I}_T \neq i^*(v)\big) \cdot \exp\left(\frac{T}{c(\log d)^{\beta_1}(H_{\mathrm{BAI}}(v)^{\beta_2} + H_{\mathrm{pri}}(v)^{\beta_3})}\right)\right\} \geq 1, \tag{24}$$

*for any* $c > 0$ *and* $\beta_1, \beta_2, \beta_3 \in [0,1]$ *with* $\beta_1 + \beta_2 + \beta_3 < 3$.

Theorem 4.7, proved in Appendix H, implies that for any chosen $\beta \in [0,1)$ (arbitrarily close to 1), there *does not exist* a consistent policy $\pi$ with an upper bound on its error probability assuming any one of the following forms for *all* instances $v \in \mathcal{P}$: $\exp\left(-\Omega\left(\frac{T}{(\log d)^\beta(H_{\mathrm{BAI}}(v)+H_{\mathrm{pri}}(v))}\right)\right)$, $\exp\left(-\Omega\left(\frac{T}{(\log d)(H_{\mathrm{BAI}}(v)^\beta+H_{\mathrm{pri}}(v))}\right)\right)$, or $\exp\left(-\Omega\left(\frac{T}{(\log d)(H_{\mathrm{BAI}}(v)+H_{\mathrm{pri}}(v)^\beta)}\right)\right)$. In this sense, the dependencies of the upper bound in (16) on $\log d$, $H_{\mathrm{BAI}}(v)$, and $H_{\mathrm{pri}}(v)$ are "tight". Also, in this precise sense, none of these terms can be improved upon in general.

*Remark* 3. It is pertinent to highlight that the upper bound in (16) applies to *any* problem instance, whereas the lower bound in (23) is a *minimax* result that is applicable to *one or more* hard instances. An ongoing quest in fixed-budget BAI is to construct a policy with provably matching error probability upper and lower bounds for *all* problem instances.

## 5 NUMERICAL STUDY

This section presents a numerical evaluation of our proposed DP-BAI policy on synthetic data, and compares it with BASELINE, an algorithm which follows DP-BAI but for Lines 6 to 13 in Algorithm 1, i.e., BASELINE does not construct MAX-DET collections. We note that BASELINE is $\varepsilon$-DP for any $\varepsilon > 0$, and bears similarities with SEQUENTIAL HALVING (Karnin et al., 2013) when $\varepsilon \to +\infty$ (i.e.,

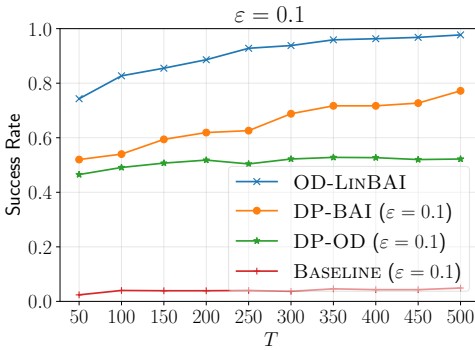 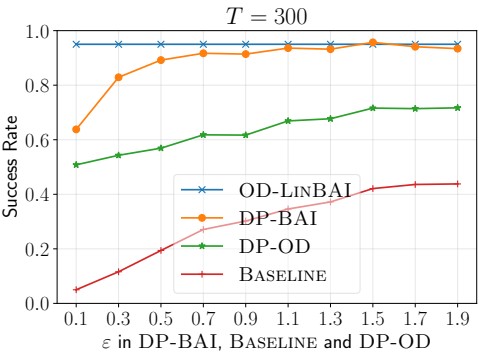

Figure 1: Comparison of DP-BAI to BASELINE, OD-LINBAI and DP-OD for different values of $\varepsilon$. Note that $\varepsilon$ is not applicable to OD-LINBAI.

non-private algorithm). However, because it does not exploit the linear structure on the arm means, we will see that it performs poorly vis-à-vis DP-BAI. In addition, we compare DP-BAI with the state-of-the-art OD-LINBAI (Yang and Tan, 2022) algorithm for fixed-budget best arm identification, which is a non-private algorithm and serves as an upper bound in performance (in terms of the error probability) of our algorithm. Also, we consider an $\varepsilon$-DP version of OD-LINBAI which we call DP-OD. A more comprehensive description of the DP-OD algorithm is presented in Appendix E.1.

Our synthetic instance is constructed as follows. We set $K = 30$, $d = 2$, and $\theta^* = [0.045\ 0.5]^\top$, $\mathbf{a}_1 = [0\ 1]^\top$, $\mathbf{a}_2 = [0\ 0.9]^\top$, $\mathbf{a}_3 = [10\ 0]^\top$, and $\mathbf{a}_i = [1\ \omega_i]^\top$ for all $i \in \{4, \dots, 30\}$, where $\omega_i$ is randomly generated from a uniform distribution on the interval $[0, 0.8]$. Clearly, $\mu_1 = 0.5$, $\mu_2 = \mu_3 = 0.45$, and $\mu_i = \omega_i/2 + 0.045$ for all $i \in \{4, \dots, 30\}$, thereby implying that arm 1 is the best arm. The sub-optimality gaps are given by $\Delta_2 = \Delta_3 = 0.05$ and $\Delta_i > 0.05$ for all $i \in \{4, \dots, 30\}$; thus, arms 2 and 3 exhibit the smallest gaps. In addition, we set $\nu_i$, the reward distribution of arm $i$, to be the uniform distribution supported on $[0, 2\mu_i]$ for all $i \in [K]$.

We run experiments with several choices for the budget $T$ and the privacy parameter $\varepsilon$, conducting 1000 independent trials for each pair of $(T, \varepsilon)$ and reporting the fraction of trials in which the best arm is successfully identified.

The experimental results are shown in Figure 1 for varying $T$ and $\varepsilon$ values respectively. As the results demonstrate, the DP-BAI policy significantly outperforms BASELINE and DP-OD, demonstrating that the utility of the MAX-DET collection in exploiting the linear structure of the arm means. We also observe that as $\varepsilon \to +\infty$ (i.e., privacy requirement vanishes), the performances of DP-BAI and the non-private state-of-the-art OD-LINBAI algorithm are similar.

## 6 CONCLUSIONS AND FUTURE WORK

This work has taken a first step towards understanding the effect of imposing a differential privacy constraint on the task of fixed-budget BAI in bandits with linearly structured mean rewards. Our contributions include the development and comprehensive analysis of a policy, namely DP-BAI, which exhibits exponential decay in error probability with respect to the budget $T$, and demonstrates a dependency on the dimensionality of the arm vectors $d$ and a composite hardness parameter, which encapsulates contributions from both the standard fixed-budget BAI task and the imposed differential privacy stipulation. A distinguishing aspect in the design of this policy is the critical utilization of the MAX-DET collection, instead of existing tools like the G-optimal designs (Yang and Tan, 2022) and $\mathcal{XY}$-adaptive allocations (Soare et al., 2014). Notably, we establish a minimax lower bound that underlines the inevitability of certain terms in the exponent of the error probability of DP-BAI.

Some interesting directions for future research include extending our work to incorporate generalized linear bandits (Azizi et al., 2022) and neural contextual bandits (Zhou et al., 2020). Additionally, we aim to tackle the unresolved question brought forth post Theorem 4.7: does there exist an efficient fixed-budget BAI policy respecting the $\varepsilon$-DP requirement, whose error probability upper bound approximately matches a *problem-dependent* lower bound?

ACKNOWLEDGEMENTS

This research/project is supported by the National Research Foundation Singapore under the AI Singapore Programme (AISG Award No: AISG2-TC-2023-012-SGIL). This work is also supported by the Singapore Ministry of Education Academic Research Fund (AcRF) Tier 2 under grant number A-8000423-00-00, and the Singapore Ministry of Education AcRF Tier 1 under grant number A-8000189-01-00.

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

# Supplementary Material for "Fixed-Budget Differentially Private Best Arm Identification"

## A    USEFUL FACTS

In this section, we collate some useful facts that will be used in the subsequent proofs.

**Lemma A.1** (Hoeffding's inequality). *Let $X_1, \ldots, X_n$ be independent random variables such that $a_i \leq X_i \leq b_i$ almost surely for all $i \in [n]$, for some fixed constants $a_i \leq b_i$, $i \in [n]$. Let*

$$S_n = X_1 + \cdots + X_n.$$

*Then, for all $\epsilon > 0$,*

$$\mathbb{P}\left(S_n - \mathbb{E}\left[S_n\right] \geq \epsilon\right) \leq \exp\left(-\frac{2\epsilon^2}{\sum_{i=1}^{n}\left(b_i - a_i\right)^2}\right), \tag{25}$$

$$\mathbb{P}\left(|S_n - \mathbb{E}\left[S_n\right]| \geq \epsilon\right) \leq 2\exp\left(-\frac{2\epsilon^2}{\sum_{i=1}^{n}\left(b_i - a_i\right)^2}\right). \tag{26}$$

**Definition A.2** (Sub-exponential random variable). Let $\tau \in \mathbb{R}$ and $b > 0$. A random variable $X$ is said to be *sub-exponential* with parameters $(\tau^2, b)$ if

$$\mathbb{E}\left[\exp(\lambda X)\right] \leq \exp\left(\frac{\lambda^2 \tau^2}{2}\right) \qquad \forall \lambda : |\lambda| < \frac{1}{b}. \tag{27}$$

**Lemma A.3** (Linear combination of sub-exponential random variables). *Let $X_1, \ldots, X_n$ be independent, zero-mean sub-exponential random variables, where $X_i$ is $(\tau_i^2, b_i)$-sub-exponential. Then, for any $a_1, \ldots, a_n \in \mathbb{R}$, the random variable $\sum_{i=1}^{n} a_i X_i$ is $(\tau^2, b)$-sub-exponential, where $\tau^2 = \sum_{i=1}^{n} a_i^2 \tau_i^2$ and $b = \max_i b_i |a_i|$.*

**Lemma A.4** (Tail bounds of sub-exponential random variables). *Suppose that $X$ is sub-exponential with parameters $(\tau^2, b)$. Then,*

$$\mathbb{P}(X - \mathbb{E}[X] \geq \epsilon) \leq \begin{cases} \exp\left(-\dfrac{\epsilon^2}{2\tau^2}\right), & 0 \leq \epsilon \leq \dfrac{\tau^2}{b}, \\ \exp\left(-\dfrac{\epsilon}{2b}\right), & \epsilon > \dfrac{\tau^2}{b}. \end{cases} \tag{28}$$

## B    AN EXAMPLE OF THE POLICY-SPECIFIC NOTATIONS IN SEC. 3.1

Consider $d = 16$, $K = 10,000$. Then, we have $g_0 = 64$, $h_0 = 9936$, $M_1 = 4$, $M = 10$ and $\lambda \approx 27.65$. In the following table, we display the values of $s_p$ for $p = 1, \ldots, M_1$.

| $p$ | $s_p$ | $(s_{p-1} - g_0)/(s_p - g_0)$ |
|---|---|---|
| 1 | 1000 | N.A. |
| 2 | 423 | $\approx 27.65$ |
| 3 | 77 | $\approx 27.68$ |
| 4 $(= M_1)$ | 64 | $\approx 27.62$ |

For $p = M_1 + 1, \ldots, M$, we simply have $s_p = 2^{10-p}$, i.e., $s_5 = 32$, $s_6 = 16, \ldots, s_{10} = 1$.

Notice that the values in the third column of the above table (apart from the first row which is not applicable) are nearly equal to the value of $\lambda$. Based on the above table, we may bifurcate the operations of our algorithm into two distinct stages. Roughly speaking, in the first stage (i.e., phases 1 to $M_1$), the algorithm aims to reduce the number of arms to a predetermined quantity (i.e., $g_0$) as quickly as possible. Following this, in the second stage (i.e., phase $M_1 + 1$ to $M$), the algorithm iteratively halves the number of arms. This process is designed to gradually concentrate arm pulls on those arms with small sub-optimality gaps (including the best arm) as much as possible, thereby improving the accuracy of estimating the mean of the best arm and encouraging effective identification of the best arm.

## C  EQUIVALENCE OF OUR NOTION OF DP WITH THE NOTION OF TABLE-DP OF AZIZE AND BASU (2023)

In this section, we discuss the notion of *table-DP* introduced in the recent work by Azize and Basu (2023, Section 2), and we will show that our definition of differential privacy in Definition 2.1 is equivalent to the notion of table-DP. Firstly, to maintain notational consistency with Azize and Basu (2023), we introduce a *dual decision maker* who obtains the reward $X_{A_t,t}$ rather than the reward $X_{A_t,N_{A_t,t}}$ upon pulling arm $A_t$ at time step $t$; see Section 2 for other relevant notations. Then, for any policy $\pi$, we write $D(\pi)$ to denote its *dual policy* obtained by substituting its decision maker with the dual decision maker. Clearly, in the non-private setting, any policy is equivalent to its dual policy. We demonstrate below that the same conclusion holds under DP considerations too. To begin, we formally define DP for the dual policy.

**Definition C.1** (Column-neighbouring). We say that $\mathbf{x}, \mathbf{x}' \in \mathcal{X}$ are *column-neighbouring* if they differ in exactly one column, i.e., there exists $t \in [T]$ such that $\mathbf{x}_{\cdot,t} \neq \mathbf{x}'_{\cdot,t}$ and $\mathbf{x}_{\cdot,s} = \mathbf{x}'_{\cdot,s}$ for all $s \neq t$, where $\mathbf{x}_{\cdot,s} := (\mathbf{x}_{i,s})_{i=1}^K$ and $\mathbf{x}'_{\cdot,s} := (\mathbf{x}'_{i,s})_{i=1}^K$.

**Definition C.2** (Table-DP). Given any $\varepsilon > 0$ and a randomised policy $\mathcal{M} : \mathcal{X} \to [K]^{T+1}$, we say that the dual policy $D(\mathcal{M})$ satisfies $\varepsilon$-*table-DP* if, for any pair of column-neighbouring $\mathbf{x}, \mathbf{x}' \in \mathcal{X}$,

$$\mathbb{P}^{D(\mathcal{M})}\left(D(\mathcal{M})(\mathbf{x}) \in \mathcal{S}\right) \leq e^\varepsilon \, \mathbb{P}^{D(\mathcal{M})}\left(D(\mathcal{M})(\mathbf{x}') \in \mathcal{S}\right) \quad \forall \mathcal{S} \subset [K]^{T+1}.$$

For any $\mathbf{z} = (z_t)_{t=1}^{T+1} \in [K]^{T+1}$, we denote $[\mathbf{z}]_i^j := (z_t)_{t=i}^j \in [K]^{j-i+1}$. Then, we have the following alternative characterization of table-DP.

**Corollary C.3.** *Given any $\varepsilon > 0$ and a randomised policy $\mathcal{M} : \mathcal{X} \to [K]^{T+1}$, the dual policy $D(\mathcal{M})$ satisfies $\varepsilon$-table-DP if and only if for any $t \in [T]$ and any pair of column-neighbouring $\mathbf{x}, \mathbf{x}' \in \mathcal{X}$ with $\mathbf{x}_{\cdot,t} \neq \mathbf{x}'_{\cdot,t}$,*

$$\mathbb{P}^{D(\mathcal{M})}\left([D(\mathcal{M})(\mathbf{x})]_{t+1}^{T+1} \in \mathcal{S}\right) \leq e^\varepsilon \, \mathbb{P}^{D(\mathcal{M})}\left([D(\mathcal{M})(\mathbf{x}')]_{t+1}^{T+1} \in \mathcal{S}\right) \quad \forall \mathcal{S} \subset [K]^{T-t+1}.$$

In the following, we demonstrate that the notions of DP in Definition C.2 and Definition 2.1 are equivalent after some slight modifications in notations.

**Proposition C.4.** *Fix any policy $\pi$. If $\pi$ satisfies $\varepsilon$-DP, then $D(\pi)$ satisfies $\varepsilon$-table-DP.*

*Proof.* Fix any $\mathbf{z} = (z_t)_{t=1}^{T+1} \in [K]^{T+1}$, and any pair of column-neighbouring $\mathbf{x}^D, \mathbf{x}'^D \in \mathcal{X}$. For $i \in [K]$ and $t \in [T]$, we let

$$n_{i,t} := \sum_{t=1}^T \mathbf{1}_{\{z_t = i\}}. \tag{29}$$

Notice that $n_{i,t}$ is equal to $N_{i,t}$ if $A_\tau = z_\tau$ for all $\tau \leq t$. Then, we construct $\mathbf{x}, \mathbf{x}' \in \mathcal{X}$ by defining for all $t \in [T]$

$$\mathbf{x}_{z_t, n_{z_t,t}} := \mathbf{x}^D_{z_t,t} \quad \text{and} \quad \mathbf{x}'_{z_t, n_{z_t,t}} := \mathbf{x}'^D_{z_t,t}, \tag{30}$$

and for all $(i,j) \in [K] \times [T] \setminus \{(z_t, n_{z_t,t}) \mid t \in [T]\}$,

$$\mathbf{x}_{i,j} := 0 \quad \text{and} \quad \mathbf{x}'_{i,j} := 0.$$

Hence, the fact that $\mathbf{x}^D, \mathbf{x}'^D \in \mathcal{X}$ are column-neighbouring, implies that either $\mathbf{x}$ and $\mathbf{x}'$ are neighbouring or $\mathbf{x} = \mathbf{x}'$. That is, by the assumption that $\pi$ satisfies $\varepsilon$-DP, we have

$$\mathbb{P}^\pi\left(\pi(\mathbf{x}) = \mathbf{z}\right) \leq e^\varepsilon \mathbb{P}^\pi\left(\pi(\mathbf{x}') = \mathbf{z}\right). \tag{31}$$

In addition, by (29) and (30) we obtain that

$$\mathbb{P}^\pi\left(\pi(\mathbf{x}) = \mathbf{z}\right) = \mathbb{P}^{D(\pi)}\left(D(\pi)(\mathbf{x}^D) = \mathbf{z}\right) \quad \text{and}$$
$$\mathbb{P}^\pi\left(\pi(\mathbf{x}') = \mathbf{z}\right) = \mathbb{P}^{D(\pi)}\left(D(\pi)(\mathbf{x}'^D) = \mathbf{z}\right). \tag{32}$$

Finally, combining (31) and (32), we have

$$\mathbb{P}^{D(\pi)}\left(D(\pi)(\mathbf{x}^D) = \mathbf{z}\right) \leq e^\varepsilon \mathbb{P}^{D(\pi)}\left(D(\pi)(\mathbf{x}'^D) = \mathbf{z}\right),$$

which completes the desired proof. $\qquad\square$

**Proposition C.5.** *Fix any policy $\pi$. If $D(\pi)$ satisfies $\varepsilon$-table-DP, then $\pi$ satisfies $\varepsilon$-DP.*

*Proof.* Fix any $\mathbf{z} = (z_i)_{i=1}^{T+1} \in [K]^{T+1}$, and any pair of neighbouring $\mathbf{x}, \mathbf{x}' \in \mathcal{X}$. Again, for $i \in [K]$ and $t \in [T]$ we let

$$n_{i,t} := \sum_{t=1}^{T} \mathbf{1}_{\{z_t = i\}}. \tag{33}$$

Then, we construct $\mathbf{x}^D, \mathbf{x}'^D \in \mathcal{X}$ by defining for all $t \in [T]$

$$\mathbf{x}_{z_t,t}^D := \mathbf{x}_{z_t, n_{z_t,t}} \quad \text{and} \quad \mathbf{x}_{z_t,t}'^D := \mathbf{x}_{z_t, n_{z_t,t}}', \tag{34}$$

and for all $(i,j) \in [K] \times [T] \setminus \{(z_t, t) \mid t \in [T]\}$,

$$\mathbf{x}_{i,j}^D := 0 \quad \text{and} \quad \mathbf{x}_{i,j}'^D := 0.$$

Hence, the fact that $\mathbf{x}, \mathbf{x}' \in \mathcal{X}$ are neighbouring, implies that either $\mathbf{x}^D$ and $\mathbf{x}'^D$ are column-neighbouring or $\mathbf{x}^D = \mathbf{x}'^D$. That is, by the assumption that $D(\pi)$ satisfies $\varepsilon$-table-DP, we have

$$\mathbb{P}^{D(\pi)}\big(D(\pi)(\mathbf{x}^D) = \mathbf{z}\big) \le e^\varepsilon \mathbb{P}^{D(\pi)}\big(D(\pi)(\mathbf{x}'^D) = \mathbf{z}\big). \tag{35}$$

In addition, by (33) and (34) we obtain

$$\mathbb{P}^\pi\big(\pi(\mathbf{x}) = \mathbf{z}\big) = \mathbb{P}^{D(\pi)}\big(D(\pi)(\mathbf{x}^D) = \mathbf{z}\big) \quad \text{and}$$
$$\mathbb{P}^\pi\big(\pi(\mathbf{x}') = \mathbf{z}\big) = \mathbb{P}^{D(\pi)}\big(D(\pi)(\mathbf{x}'^D) = \mathbf{z}\big). \tag{36}$$

Finally, combining (35) and (36), we have

$$\mathbb{P}^\pi\big(\pi(\mathbf{x}) = \mathbf{z}\big) \le e^\varepsilon \mathbb{P}^\pi\big(\pi(\mathbf{x}') = \mathbf{z}\big),$$

which completes the proof. $\qquad\square$

Combining Propositions C.4 and C.5, we obtain the following corollary.

**Corollary C.6.** *A policy $\pi$ satisfies $\varepsilon$-DP if and only if $D(\pi)$ satisfies $\varepsilon$-table-DP.*

This proves the equivalence betweeen our notion of DP in Definition 2.1 and the notion of table-DP appearing in the work of Azize and Basu (2023).

## D  EXTENSION OF OUR RESULTS TO $(\varepsilon, \delta)$-DIFFERENTIAL PRIVACY

Below, we first define the $(\varepsilon, \delta)$-differential privacy constraint formally.

**Definition D.1** ($(\varepsilon, \delta)$-differential privacy). Given any $\varepsilon > 0$ and $\delta > 0$, a randomised policy $\mathcal{M} : \mathcal{X} \to [K]^{T+1}$ satisfies $(\varepsilon, \delta)$-differential privacy if, for any pair of neighbouring $\mathbf{x}, \mathbf{x}' \in \mathcal{X}$,

$$\mathbb{P}^\mathcal{M}(\mathcal{M}(\mathbf{x}) \in \mathcal{S}) \le e^\varepsilon \, \mathbb{P}^\mathcal{M}(\mathcal{M}(\mathbf{x}') \in \mathcal{S}) + \delta \quad \forall \mathcal{S} \subset [K]^{T+1}.$$

The above definition particularises to that of $\varepsilon$-differential privacy when $\delta = 0$. Therefore, it is clear that any policy that satisfies the $\varepsilon$-DP constraint automatically satisfies $(\varepsilon, \delta)$-DP constraint for all $\delta > 0$. In particular, our proposed DP-BAI policy satisfies the $(\varepsilon, \delta)$-DP constraint for all $\delta > 0$.

However, DP-BAI has been specifically designed for $(\varepsilon, 0)$-DP, and does not adjust the agent's strategy for varying values of $\delta$. Therefore, exclusively for the $(\varepsilon, \delta)$-differential privacy constraint with $\delta > 0$, we provide a variant of our proposed algorithm called DP-BAI-GAUSS that utilises the Gaussian mechanism (Dwork et al., 2014) and operates with additive Gaussian noise (in contrast to Laplacian noise under DP-BAI). The pseudo-code of DP-BAI-GAUSS is shown in Algorithm 2, where the differences from DP-BAI are highlighted in red.

**Proposition D.2.** *The* DP-BAI-GAUSS *policy with privacy and budget parameters $(\varepsilon, \delta, T)$ satisfies the $(\varepsilon, \delta)$-DP constraint, i.e., for any pair of neighbouring $\mathbf{x}, \mathbf{x}' \in \mathcal{X}$ and $\forall \mathcal{S} \subseteq [K]^{T+1}$,*

$$\mathbb{P}^{\Pi_{\text{DP-BAI-GAUSS}}}(\Pi_{\text{DP-BAI-GAUSS}}(\mathbf{x}) \in \mathcal{S}) \le e^\varepsilon \, \mathbb{P}^{\Pi_{\text{DP-BAI-GAUSS}}}(\Pi_{\text{DP-BAI-GAUSS}}(\mathbf{x}') \in \mathcal{S}) + \delta. \tag{37}$$

---

**Algorithm 2** DP-BAI-GAUSS

---

**Input:**
$\varepsilon, \delta$: differential privacy parameters; $T$: budget; $\{\mathbf{a}_i : i \in [K]\}$: $d$-dimensional feature vectors.

**Output:** $\hat{I}_T$: best arm.

1: Initialise $T_0 = 0$, $\mathcal{A}_1 = [K]$, $\mathbf{a}_i^{(0)} = \mathbf{a}_i$ for all $i \in [K]$. Set $M$ and $T'$ as in (8).
2: **for** $p \in \{1, 2, \ldots, M\}$ **do**
3:      Set $d_p = \dim(\mathrm{span}\{\mathbf{a}_i^{(p-1)} : i \in \mathcal{A}_p\})$.
4:      Obtain the new vector set $\{\mathbf{a}_i^{(p)} : i \in \mathcal{A}_p\}$ from the set $\{\mathbf{a}_i^{(p-1)} : i \in \mathcal{A}_p\}$ via (9).
5:      Compute $s_p$ using (7).
6:      **if** $d_p < \sqrt{s_p}$ **then**
7:          Construct a MAX-DET collection $\mathcal{B}_p \subset \mathcal{A}_p$.
8:          Pull each arm in $\mathcal{B}_p$ for $\lceil \frac{T'}{Md_p} \rceil$ many times. Update $T_p \leftarrow T_{p-1} + d_p \lceil \frac{T'}{Md_p} \rceil$.
9:          Obtain the empirical means $\{\hat{\mu}_i(p) : i \in \mathcal{B}_p\}$ via (10).
10:         Generate $\widetilde{\xi}_i^{(p)} \sim \mathrm{Gaussian}\left(0, \frac{2\log(1.25/\delta)}{(\varepsilon \lceil \frac{T'}{Md_p} \rceil)^2}\right)$ for $i \in \mathcal{B}_p$.
11:         Set $\widetilde{\mu}_i^{(p)} \leftarrow \hat{\mu}_i^{(p)} + \widetilde{\xi}_i^{(p)}$ for all $i \in \mathcal{B}_p$.
12:         For arm $i \in \mathcal{A}_p \setminus \mathcal{B}_p$, compute $\widetilde{\mu}_i^{(p)}$ via (12).
13:      **else**
14:         Pull each arm in $\mathcal{A}_p$ for $\lceil \frac{T'}{Ms_p} \rceil$ many times. Update $T_p \leftarrow T_{p-1} + s_p \lceil \frac{T'}{Ms_p} \rceil$
15:         Obtain the empirical means $\{\hat{\mu}_i^{(p)} : i \in \mathcal{A}_p\}$ via (10).
16:         Generate $\widetilde{\xi}_i^{(p)} \sim \mathrm{Gaussian}\left(0, \frac{2\log(1.25/\delta)}{(\varepsilon \lceil \frac{T'}{Ms_p} \rceil)^2}\right)$ for $i \in \mathcal{A}_p$.
17:         Set $\widetilde{\mu}_i^{(p)} \leftarrow \hat{\mu}_i^{(p)} + \widetilde{\xi}_i^{(p)}$ for all $i \in \mathcal{A}_p$.
18:      **end if**
19:      Compute $s_{p+1}$ using (7).
20:      $\mathcal{A}_{p+1} \leftarrow$ the set of $s_{p+1}$ arms with largest private empirical means among $\{\widetilde{\mu}_i^{(p)} : i \in \mathcal{A}_p\}$.
21: **end for**
22: $\hat{I}_T \leftarrow$ the only arm remaining in $\mathcal{A}_{M+1}$
23: **return** Best arm $\hat{I}_T$.

---

The proof of Proposition D.2 is deferred until Section I. Below, we present an upper bound on the error probability of DP-BAI-GAUSS.

**Proposition D.3.** *For any problem instance $v \in \mathcal{P}$,*

$$\mathbb{P}_v^{\Pi_{\text{DP-BAI-GAUSS}}}(\hat{I}_T \neq i^*(v)) \leq \exp\left(-\Omega\left(\frac{T}{H_{\text{BAI}} \log d} \wedge \left(\frac{T}{\sqrt{\log(\frac{1.25}{\delta})} H_{\text{pri}} \log d}\right)^2\right)\right),$$

*where $H_{\text{BAI}}$ and $H_{\text{pri}}$ are as defined in* (15).

The proof of Proposition D.3 follows along the lines of the proof of Theorem 4.2, by using sub-Gaussian concentration bounds in place of sub-exponential concentration bounds. The proof is deferred until Section J.

*Remark* 4. It is pertinent to highlight that the contribution of our algorithms do not lie in the introduction of a novel privacy mechanism such as Laplace and Gaussian mechanisms. Instead, we propose a new *sampling strategy* based on the MAX-DET principle for fixed budget BAI. This strategy can be seamlessly integrated into any differential privacy mechanism, including the Laplace and the Gaussian mechanisms.

Table 1 shows a comparison of the upper bounds for DP-BAI-GAUSS and DP-BAI (from Theorem 4.2). Because DP-BAI is $(\varepsilon, \delta)$-differentially private for all $\delta > 0$ as alluded to earlier, the preceding comparison is valid.

We observe that DP-BAI-GAUSS outperforms DP-BAI for large values of $\delta$ and $T$. In the small $\delta$ regime, however, DP-BAI performs better than DP-BAI-GAUSS. In particular, when $\delta = 0$, the

| Condition on $T$ and $\delta$ | DP-BAI-GAUSS | DP-BAI |
|---|---|---|
| $\left(\frac{T}{\sqrt{\log(\frac{5}{4\delta})}H_{\text{pri}}\,\log d}\right)^2 \geq \frac{T}{H_{\text{BAI}}\,\log d}$ | $\exp\left(-\Omega\left(\frac{T}{H_{\text{BAI}}\,\log d}\right)\right)$ ✓ | $\exp\left(-\Omega\left(\frac{T}{H\,\log d}\right)\right)$ |
| $\frac{T}{H\log d} \leq \left(\frac{T}{\sqrt{\log(\frac{5}{4\delta})}H_{\text{pri}}\log d}\right)^2 < \frac{T}{H_{\text{BAI}}\log d}$ | $\exp\left(-\Omega\left(\frac{T}{\sqrt{\log(\frac{5}{4\delta})}H_{\text{pri}}\log d}\right)^2\right)$ ✓ | $\exp\left(-\Omega\left(\frac{T}{H\log d}\right)\right)$ |
| $\left(\frac{T}{\sqrt{\log(\frac{5}{4\delta})}H_{\text{pri}}\log d}\right)^2 < \frac{T}{H\log d}$ | $\exp\left(-\Omega\left(\frac{T}{\sqrt{\log(\frac{5}{4\delta})}H_{\text{pri}}\log d}\right)^2\right)$ | $\exp\left(-\Omega\left(\frac{T}{H\log d}\right)\right)$ ✓ |
| $\delta = 0,\ T > 0$ | $\exp(-\Omega(1))$ (vacuous) | $\exp\left(-\Omega\left(\frac{T}{H\log d}\right)\right)$ ✓ |

Table 1: Comparison between the upper bounds of DP-BAI and DP-BAI-GAUSS. The ticks indicate the tighter of the two bounds under the given condition.

additive noise in the Gaussian mechanism is infinite and thus vacuous. That is, DP-BAI-GAUSS is not applicable when $\delta = 0$. The above trends are not surprising, given that DP-BAI is designed for $\delta = 0$ and is hence expected to work well for small values of $\delta$. Finally, in order to keep the analysis simple and bring out the main ideas clearly, we hence only discuss $\varepsilon$-DP criterion in our main text, and the discussion regarding of $(\varepsilon, \delta)$-DP is presented in this section of the appendix.

# E   A DIFFERENTIALLY PRIVATE VERSION OF OD-LINBAI (YANG AND TAN, 2022) AND ITS ANALYSIS

## E.1   DETAILS OF IMPLEMENTING DP-OD

To achieve $\varepsilon$-DP for DP-OD, a variant of OD-LINBAI (Yang and Tan, 2022), one of the natural solutions is to use the idea of Shariff and Sheffet (2018), which is to add random noise in the ordinary least square (OLS) estimator; we note that OD-LINBAI uses the OLS estimator as a subroutine, similar to the algorithm of Shariff and Sheffet (2018). Specifically, given a dataset $\{(\boldsymbol{x}_i, y_i)\}_{i=1}^n$, one of the steps in the computation of the standard OLS estimator (Weisberg, 2005, Chapter 2) is to evaluate the coefficient vector $\hat{\boldsymbol{\beta}}$ via

$$\boldsymbol{G}\hat{\boldsymbol{\beta}} = \boldsymbol{U},$$

where $\boldsymbol{G}$ is the *Gram* matrix in the OLS estimator and $\boldsymbol{U} = \sum_{i=1}^n \boldsymbol{x}_i y_i$ is the *moment* matrix. While Shariff and Sheffet (2018) add random noise to both $\boldsymbol{G}$ and $\boldsymbol{U}$, we do not add noise to $\boldsymbol{G}$ as the feature vectors are determined and known to the decision maker in our setting. Instead, we add independent Laplacian noise with parameter $\frac{L}{\varepsilon}$ to each element of $\boldsymbol{U}$ of the OLS estimator, with $L = \max_{i \in [K]} \|\mathbf{a}_i\|$; we use Laplacian noise instead of Gaussian or Wishart noise, noting from Shariff and Sheffet (2018, Section 4.2) and Sheffet (2015, Theorem 4.1) that the latter versions of noise can potentially be infinite and hence not particularly suitable for satisfying the $\varepsilon$-DP constraint. Using Shariff and Sheffet (2018, Claim 7), we may then prove that the above method satisfies the $\varepsilon$-DP constraint.

## E.2   A BRIEF ANALYSIS OF DP-OD

The error probability of identifying the best arm under DP-OD depends not only on the suboptimality gaps of the arms, but also inherently a function of the arm vectors, as demonstrated below.

**Proposition E.1.** *Let $K = 2$, $\mathbf{a}_1 = [x, 0]^\top$, $\mathbf{a}_2 = [0, y]^\top$ with $x, y > 0$, and $\theta^* = [(0.5 + \Delta)/x,\ 0.5/y]^\top$ for some fixed $\Delta > 0$. The upper bound on the error probability of* DP-OD *for the above bandit instance is*

$$\exp\left(-\Omega\left(\frac{T}{\Delta^{-2} + \frac{x \vee y}{x \wedge y}(\epsilon\Delta)^{-1}}\right)\right).$$

*Proof.* Note that there is only one round in DP-OD for the two-armed bandit instance. Let $T_1$ and $T_2$ be the number of arm pulls for arm 1 and arm 2, respectively. By the sampling strategy of DP-OD, we have $T_1 = \Theta(T)$ and $T_2 = \Theta(T)$ (as $T \to \infty$). From the description in Section E.1, we note that

the moment matrix is given by $\mathbf{U} = \sum_{i=1}^{T_1} \mathbf{a}_1 X_{1,i} + \sum_{i=1}^{T_2} \mathbf{a}_2 X_{2,i}$. We denote $\tilde{\mathbf{U}}$ to be the moment matrix with Laplacian noise, i.e.,

$$\widetilde{\mathbf{U}} = \mathbf{U} + [\widetilde{\xi}_1, \widetilde{\xi}_2]^\top, \tag{38}$$

where $\widetilde{\xi}_1$ and $\widetilde{\xi}_2$ are independent Laplacian noises with parameters $\frac{L}{\varepsilon}$ and $L = x \vee y$, respectively. The estimates of the expected reward of arm 1 and arm 2 are

$$\widetilde{\mu}_1 = \langle \mathbf{G}^{-1}\widetilde{\mathbf{U}}, \mathbf{a}_1 \rangle \tag{39}$$

and

$$\widetilde{\mu}_2 = \langle \mathbf{G}^{-1}\widetilde{\mathbf{U}}, \mathbf{a}_2 \rangle, \tag{40}$$

where the Gram matrix $\mathbf{G} = T_1 \mathbf{a}_1 \mathbf{a}_1^\top + T_2 \mathbf{a}_2 \mathbf{a}_2^\top$. Then, the event $\{\widetilde{\mu}_1 > \widetilde{\mu}_2\}$ implies that the decision maker successfully identifies the best arm. In other words,

$$\mathbb{P}(\hat{I}_T \neq 1) \leq \mathbb{P}(\widetilde{\mu}_1 \leq \widetilde{\mu}_2). \tag{41}$$

By (39) and (40), we have

$$\widetilde{\mu}_1 - \widetilde{\mu}_2 = \frac{1}{T_1}\sum_{i=1}^{T_1} X_{1,i} - \frac{1}{T_2}\sum_{i=1}^{T_2} X_{2,i} + \left(\frac{\widetilde{\xi}_1}{xT_1} - \frac{\widetilde{\xi}_2}{yT_2}\right).$$

Hence,

$$\mathbb{P}(\widetilde{\mu}_1 - \widetilde{\mu}_2 < 0) \leq \mathbb{P}\left(\sum_{i=1}^{T_1} \frac{X_{1,i}}{T_1} < \mu_1 - \frac{\Delta}{4}\right) + \mathbb{P}\left(\sum_{i=1}^{T_2} \frac{X_{2,i}}{T_2} > \mu_2 + \frac{\Delta}{4}\right)$$

$$+ \mathbb{P}\left(\frac{\widetilde{\xi}_1}{xT_1} < -\frac{\Delta}{4}\right) + \mathbb{P}\left(\frac{\widetilde{\xi}_2}{yT_2} > \frac{\Delta}{4}\right). \tag{42}$$

By Lemma A.1, we have

$$\mathbb{P}\left(\sum_{i=1}^{T_1} \frac{X_{1,i}}{T_1} < \mu_1 - \frac{\Delta}{4}\right) \leq \exp\left(-\frac{T_1}{8\Delta^2}\right), \tag{43}$$

and

$$\mathbb{P}\left(\sum_{i=1}^{T_2} \frac{X_{2,i}}{T_2} > \mu_2 + \frac{\Delta}{4}\right) \leq \exp\left(-\frac{T_2}{8\Delta^2}\right). \tag{44}$$

For sufficiently large $T_1$ and $T_2$, by Lemma A.4, we have

$$\mathbb{P}\left(\frac{\widetilde{\xi}_1}{xT_1} < -\frac{\Delta}{4}\right) \leq \exp\left(-\frac{T_1\Delta\varepsilon}{8(x \vee y)/x}\right), \tag{45}$$

and

$$\mathbb{P}\left(\frac{\widetilde{\xi}_2}{yT_2} > \frac{\Delta}{4}\right) \leq \exp\left(-\frac{T_2\Delta\varepsilon}{8(x \vee y)/y}\right). \tag{46}$$

Recall that $T_1 = \Theta(T)$ and $T_2 = \Theta(T)$ (as $T \to \infty$), and by combining (42), (43), (44), (45) and (46), we have

$$\mathbb{P}(\widetilde{\mu}_1 < \widetilde{\mu}_2) \leq \exp\left(-\Omega\left(\frac{T}{\Delta^{-2} + \frac{x \vee y}{x \wedge y}(\epsilon\Delta)^{-1}}\right)\right).$$

$\square$

Noting that $\frac{x \vee y}{x \wedge y}$ can be arbitrarily large, the above bound is inferior to the upper bound of DP-BAI (equal to $\exp(-\Omega(\frac{T}{\Delta^{-2}+(\epsilon\Delta)^{-1}}))$) for the above bandit instance). Figure 2 shows the empirical performances of DP-BAI and DP-OD by fixing $x$ and varying $y$ in this problem instance, where $\Delta = 0.05$, $x = 1$ is fixed, and $y > 1$ is allowed to vary, and the reward distribution is uniform distribution in $[0, 2\mu_i]$ for arm $i = 1, 2$. The success rates are obtained by averaging across 1000 independent trials. We observe that when the ratio $\frac{x \vee y}{x \wedge y}$ increases, the performance of DP-OD becomes worse. However, the performance of DP-BAI is essentially independent of this ratio as the theory suggests. The numerical results presented in Section 5 of our paper as well as this additional numerical study, showing the empirical performances of DP-BAI and DP-OD, reaffirm the inferiority of DP-OD.

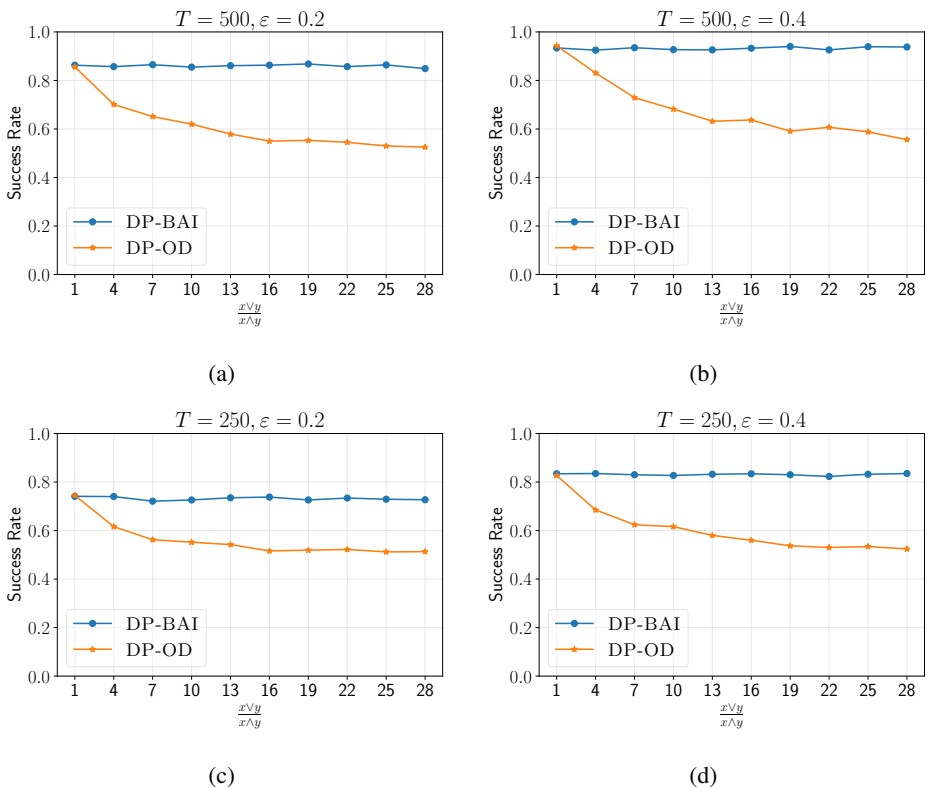

Figure 2: Comparison of DP-BAI to DP-OD for different values of $\frac{x \vee y}{x \wedge y}$ in the two-armed bandit instance introduced in Appendix E.2.

## F PROOF OF PROPOSITION 4.1

Let $N_i^{(p)} \coloneqq N_{i,T_p}$. Under the (random) process of DP-BAI, we introduce an auxiliary random mechanism $\tilde{\pi}$ whose input is $\mathbf{x} \in \mathcal{X}$, and whose output is $(N_i^{(P)}, \widetilde{\mu}_i^{(p)})_{i \in [K], p \in [M]}$, where we define $\widetilde{\mu}_i^{(p)} = 0$ if $i \notin \mathcal{A}_p$. By Dwork et al. (2014, Proposition 2.1), to prove Proposition 4.1, it suffices to show that $\tilde{\pi}$ is $\varepsilon$-differentially private, i.e., for all neighbouring $\mathbf{x}, \mathbf{x}' \in \mathcal{X}$,

$$\mathbb{P}^{\tilde{\pi}}(\tilde{\pi}(\mathbf{x}) \in \mathcal{S}) \leq e^{\varepsilon}\, \mathbb{P}^{\tilde{\pi}}(\tilde{\pi}(\mathbf{x}') \in \mathcal{S}) \quad \forall \mathcal{S} \subseteq \tilde{\Omega}, \tag{47}$$

where $\tilde{\Omega}$ is the set of all possible outputs of $\tilde{\pi}$. In the following, we write $\mathbb{P}_{\mathbf{x}}^{\tilde{\pi}}$ to denote the probability measure induced under $\tilde{\pi}$ with input $\mathbf{x} \in \mathcal{X}$.

Fix neighbouring $\mathbf{z}, \mathbf{z}' \in \mathcal{X}$ arbitrarily. There exists a unique pair $(\underline{i}, \underline{t})$ with $\mathbf{z}_{\underline{i},\underline{n}} \neq \mathbf{z}'_{\underline{i},\underline{n}}$. In addition, fix any $n_i^{(p)} \in \mathbb{N}$ and Borel set $\chi_i^{(p)} \subseteq [0,1]$ for $i \in [K]$ and $p \in [M]$ such that

$$0 \leq n_i^{(1)} \leq n_i^{(2)} \leq \ldots \leq n_i^{(M)} \leq T \quad \forall i \in [K].$$

Let

$$\underline{p} = \begin{cases} M+1, & \text{if } n_{\underline{i}}^{(M)} < \underline{n} \\ \min\{p \in [M] : n_i^{(p)} \geq \underline{n}\}, & \text{otherwise.} \end{cases} \tag{48}$$

For any $j \in [M]$, we define the event

$$D_j = \{\forall (i,p) \in [K] \times [j] : N_i^{(p)} = n_i^{(p)}, \widetilde{\mu}_i^{(p)} \in \chi_i^{(p)}\}$$

and

$$D_j' = D_{j-1} \cap \{N_j^{(p)} = n_j^{(p)}\}.$$

For $j' \in \{j+1, \ldots, M\}$

$$D_{j,j'} = \{\forall (i,p) : \in [K] \times \{j+1, \ldots, j'\} : \; N_i^{(p)} = n_i^{(p)}, \widetilde{\mu}_i^{(p)} \in \chi_i^{(p)}\}$$

In particular, $D_0$ and $D_{M,M}$ are events with probability 1, i.e.,

$$\mathbb{P}_{\mathbf{x}}^{\tilde{\pi}}(D_0 \cap D_{M,M}) = 1 \quad \forall \mathbf{x} \in \mathcal{X}.$$

Then, if $n_{\underline{i}}^{(M)} < \underline{n}$, by the definition of $\tilde{\pi}$ we can have

$$\mathbb{P}_{\mathbf{z}}^{\tilde{\pi}}(D_M) = \mathbb{P}_{\mathbf{z}'}^{\tilde{\pi}}(D_M). \tag{49}$$

In the following, we assume $n_{\underline{i}}^{(M)} \geq \underline{n}$ and we will show

$$\mathbb{P}_{\mathbf{z}}^{\tilde{\pi}}(D_M) \leq \exp(\varepsilon)\mathbb{P}_{\mathbf{z}'}^{\tilde{\pi}}(D_M) \tag{50}$$

which then implies that $\tilde{\pi}$ is $\varepsilon$-differentially private.

For $\mathbf{x} \in \{\mathbf{z}, \mathbf{z}'\}$, we denote

$$\mu_{\mathbf{x}} = \frac{1}{n_{\underline{i}}^{(p)} - n_{\underline{i}}^{(p-1)}} \sum_{j=n_{\underline{i}}^{(p-1)}+1}^{n_{\underline{i}}^{(p)}} \mathbf{x}_{\underline{i},j}$$

That is, $\mu_{\mathbf{x}}$ is the value of $\hat{\mu}_{\underline{i}}^{(p)}$ in the condition of $D'_{\underline{p}}$ under probability measure $\mathbb{P}_{\mathbf{x}}^{\tilde{\pi}}$.

Then, by the chain rule we have for both $\mathbf{x} \in \{\mathbf{z}, \mathbf{z}'\}$

$$\mathbb{P}_{\mathbf{x}}^{\tilde{\pi}}(D_M) = \mathbb{P}_{\mathbf{x}}^{\tilde{\pi}}\left(D'_{\underline{p}}\right) \mathbb{P}_{\mathbf{x}}^{\tilde{\pi}}\left(\widetilde{\mu}_{\underline{i}}^{(p)} \in \chi_i^{(p)} \mid D'_{\underline{p}}\right) \mathbb{P}_{\mathbf{x}}^{\tilde{\pi}}\left(D_{\underline{p},M} \mid D_{\underline{p}}\right)$$

$$= \mathbb{P}_{\mathbf{x}}^{\tilde{\pi}}\left(D'_{\underline{p}}\right) \int_{x \in \chi_i^{(p)}} f_{\mathrm{Lap}}(x - \mu_{\mathbf{x}})\mathbb{P}_{\mathbf{x}}^{\tilde{\pi}}\left(D_{\underline{p},M} \mid D'_{\underline{p}} \cap \{\widetilde{\mu}_{\underline{i}}^{(p)} = x\}\right) \mathrm{d}x \tag{51}$$

where $f_{\mathrm{Lap}}(\cdot)$ is the probability density function of Laplace distribution with parameter $\frac{1}{\varepsilon(n_{\underline{i}}^{(p)} - n_{\underline{i}}^{(p-1)})}$.
Note that by definition it holds

$$\begin{cases} \mathbb{P}_{\mathbf{z}}^{\tilde{\pi}}\left(D'_{\underline{p}}\right) = \mathbb{P}_{\mathbf{z}'}^{\tilde{\pi}}\left(D'_{\underline{p}}\right) \\ \mathbb{P}_{\mathbf{z}}^{\tilde{\pi}}\left(D_{\underline{p},M} \mid D'_{\underline{p}} \cap \{\widetilde{\mu}_{\underline{i}}^{(p)} = x\}\right) = \mathbb{P}_{\mathbf{z}'}^{\tilde{\pi}}\left(D_{\underline{p},M} \mid D'_{\underline{p}} \cap \{\widetilde{\mu}_{\underline{i}}^{(p)} = x\}\right) \quad \forall x \in \chi_i^{(p)} \\ f_{\mathrm{Lap}}(x - \mu_{\mathbf{z}}) \leq \exp(\varepsilon)f_{\mathrm{Lap}}(x - \mu_{\mathbf{z}'}) \quad \forall x \in \chi_i^{(p)} \end{cases} \tag{52}$$

Hence, combining (51) and (52), we have

$$\mathbb{P}_{\mathbf{z}}^{\tilde{\pi}}(D_M) \leq \exp(\varepsilon)\mathbb{P}_{\mathbf{z}'}^{\tilde{\pi}}(D_M)$$

which implies $\tilde{\pi}$ is $\varepsilon$-differentially private.

## G  PROOF OF THEOREM 4.2

**Lemma G.1.** *Fix $d' \in \mathbb{N}$ and a finite set $\mathcal{A} \subset \mathbb{R}^{d'}$ such that $\mathrm{span}(\mathcal{A}) = \mathbb{R}^{d'}$. Let $\mathcal{B} = \{\mathbf{b}_1, \ldots, \mathbf{b}_{d'}\}$ be a MAX-DET collection of $\mathcal{A}$. Fix an arbitrary $\mathbf{z} \in \mathcal{A}$, and let $\beta_1, \ldots, \beta_{d'} \in \mathbb{R}$ be the unique set of coefficients such that*

$$\mathbf{z} = \sum_{i=1}^{d'} \beta_i \mathbf{b}_i.$$

*Then, $|\beta_i| \leq 1$ for all $i \in [d']$.*

*Proof.* Fix $z \in \mathcal{A}$.Let $\boldsymbol{B} = [\boldsymbol{b}_1 \quad \boldsymbol{b}_2 \quad \dots \quad \boldsymbol{b}_{d'}]$ be the $d' \times d'$ matrix consisting of vectors in $\mathcal{B}$. Let $\boldsymbol{\beta} = [\beta_1 \quad \beta_2 \quad \dots \quad \beta_{d'}]^\top$ to be a vector consisting of the members $\beta_1, \dots, \beta_{d'}$.

Then, by the definition of $\beta_i$ for $i \in [d']$, we have

$$\boldsymbol{B}\boldsymbol{\beta} = \mathbf{z}.$$

In addition, let $\boldsymbol{B}^{(i)}$ be the matrix that is identical to $\boldsymbol{B}$ except the $i$-th column is replaced by $\mathbf{z}$, i.e., $\boldsymbol{B}^{(i)} = [\boldsymbol{b}_1 \quad \dots \quad \boldsymbol{b}_{i-1} \quad \mathbf{z} \quad \boldsymbol{b}_{i+1} \quad \dots \quad \boldsymbol{b}_{d'}]$, for $i \in [d']$. Note that by the definition of MAX-DET collection, we have

$$|\det(\boldsymbol{B}^{(i)})| \leq |\det(\boldsymbol{B})|. \tag{53}$$

By Cramer's Rule (Meyer, 2000, Chapter 6), we have

$$\beta_i = \frac{\det(\boldsymbol{B}^{(i)})}{\det(\boldsymbol{B})}. \tag{54}$$

Finally, combining (53) and (54), we have for all $i \in [d']$

$$|\beta_i| \leq 1$$

$\square$

**Lemma G.2.** *Fix an instance $v \in \mathcal{P}$ and $p \in [M]$. Let $i^*(v)$ denote the unique best arm under $v$. For any set $\mathcal{Q} \subset [K]$ with $|Q| = s_p$ and $i^*(v) \in Q$, let $d_\mathcal{Q} = \dim(\text{span}\{\mathbf{a}_i : i \in \mathcal{Q}\})$. We have*

$$\mathbb{P}_v^{\Pi_{\text{DP-BAI}}} \left( \widetilde{\mu}_j^{(p)} \geq \widetilde{\mu}_{i^*(v)}^{(p)} \mid \mathcal{A}_p = \mathcal{Q} \right)$$

$$\leq 2 \exp\left( -\frac{\Delta_j^2}{16} \left\lceil \frac{T'}{M(d_\mathcal{Q}^2 \wedge s_p)} \right\rceil \right) + 2 \exp\left( -\frac{\varepsilon \Delta_j}{16} \left\lceil \frac{T'}{M(d_\mathcal{Q}^2 \wedge s_p)} \right\rceil \right) \tag{55}$$

*for all $T'$ sufficiently large and $j \in \mathcal{Q} \setminus \{i^*(v)\}$.*

*Proof.* Fix $j \in \mathcal{Q} \setminus \{i^*(v)\}$. Notice that the sampling strategy of $\Pi_{\text{DP-BAI}}$ depends on the relation between $|\mathcal{Q}|$ and $d_\mathcal{Q}$. We consider two cases.

**Case 1**: $d_\mathcal{Q} > \sqrt{|\mathcal{Q}|}$.

In this case, note that each arm in $\mathcal{Q}$ is pulled $\left\lceil \frac{T'}{M|\mathcal{Q}|} \right\rceil$ many times. Let

$$E_1 := \{\hat{\mu}_j^{(p)} - \hat{\mu}_{i^*(v)}^{(p)} + \Delta_j \geq \Delta_j/2\}. \tag{56}$$

Let $\bar{X}_{i,s} = X_{i,s} - \mu_i$ for all $i \in [K]$ and $s \in [T]$. Notice that $\bar{X}_{i,s} \in [-1,1]$ for all $(i,s)$. Then,

$$\mathbb{P}_v^{\Pi_{\text{DP-BAI}}}(E_1 \mid \mathcal{A}_p = \mathcal{Q})$$

$$\leq \mathbb{P}_v^{\Pi_{\text{DP-BAI}}} \left( \sum_{s=N_{j,T_p}}^{N_{j,T_p} + \left\lceil \frac{T'}{M|\mathcal{Q}|} \right\rceil} \bar{X}_{j,s} - \sum_{s=N_{i^*(v),T_p}}^{N_{i^*(v),T_p} + \left\lceil \frac{T'}{M|\mathcal{Q}|} \right\rceil} \bar{X}_{i^*(v),s} \geq \frac{\Delta_j}{2} \left\lceil \frac{T'}{M|\mathcal{Q}|} \right\rceil \mid \mathcal{A}_p = \mathcal{Q} \right)$$

$$\overset{(a)}{\leq} \exp\left( -\frac{2\left(\frac{1}{2}\left\lceil \frac{T'}{M|\mathcal{Q}|}\right\rceil \Delta_j\right)^2}{2\left\lceil \frac{T'}{M|\mathcal{Q}|}\right\rceil} \right)$$

$$= \exp\left( -\frac{1}{4}\left\lceil \frac{T'}{M|\mathcal{Q}|}\right\rceil \Delta_j^2 \right), \tag{57}$$

where (a) follows Lemma A.1. Furthermore, let

$$E_2 := \{\xi_j^{(p)} - \xi_{i^*(v)}^{(p)} \geq \Delta_j/2\}. \tag{58}$$

We note that both $\xi_j^{(p)}$ and $\xi_{i^*(v)}^{(p)}$ are $(\tau^2, b)$ sub-exponential, where $\tau = b = \frac{2}{\left\lceil \frac{T'}{M|\mathcal{Q}|} \right\rceil \varepsilon}$. Lemma A.3 then yields that $\xi_j^{(p)} - \xi_{i^*(v)}^{(p)}$ is sub-exponential with parameters $\left( \left( \frac{2\sqrt{2}}{\left\lceil \frac{T'}{M|\mathcal{Q}|} \right\rceil \varepsilon} \right)^2, \frac{2}{\left\lceil \frac{T'}{M|\mathcal{Q}|} \right\rceil \varepsilon} \right)$. Subsequently, Lemma A.4 yields that for all sufficiently large $T'$,

$$
\mathbb{P}_v^{\Pi_{\text{DP-BAI}}} \left( E_2 \mid \mathcal{A}_p = \mathcal{Q} \right) \leq \exp \left( -\frac{\Delta_j/2}{\frac{4}{\left\lceil \frac{T'}{M|\mathcal{Q}|} \right\rceil \varepsilon}} \right)
$$
$$
= \exp \left( -\frac{\left\lceil \frac{T'}{M|\mathcal{Q}|} \right\rceil \varepsilon \Delta_j}{8} \right). \tag{59}
$$

We therefore have

$$
\mathbb{P}_v^{\Pi_{\text{DP-BAI}}} \left( \widetilde{\mu}_j^{(p)} \geq \widetilde{\mu}_{i^*(v)}^{(p)} \mid \mathcal{A}_p = \mathcal{Q} \right)
$$
$$
\leq \mathbb{P}_v^{\Pi_{\text{DP-BAI}}} \left( E_1 \cup E_2 \mid \mathcal{A}_p = \mathcal{Q} \right)
$$
$$
\overset{(a)}{\leq} \exp \left( -\frac{1}{4} \left\lceil \frac{T'}{M|\mathcal{Q}|} \right\rceil \Delta_j^2 \right) + \exp \left( -\frac{1}{8} \left\lceil \frac{T'}{M|\mathcal{Q}|} \right\rceil \varepsilon \Delta_j \right)
$$
$$
= \exp \left( -\frac{1}{4} \left\lceil \frac{T'}{M(d_Q^2 \wedge s_p)} \right\rceil \Delta_j^2 \right) + \exp \left( -\frac{1}{8} \left\lceil \frac{T'}{M(d_Q^2 \wedge s_p)} \right\rceil \varepsilon \Delta_j \right), \tag{60}
$$

where (a) follows from the union bound.

**Case 2**: $d_Q \leq \sqrt{|\mathcal{Q}|}$.

In this case, each arm in $\mathcal{B}_p \subset \mathcal{Q}$ is pulled for $\left\lceil \frac{T'}{Md_Q} \right\rceil$ times. Recall that we have $\mathbf{a}_j^{(p)} = \sum_{i \in \mathcal{B}_p} \alpha_{j,i} \mathbf{a}_i^{(p)}$. Using (9), this implies $\mathbf{a}_j = \sum_{i \in \mathcal{B}_p} \alpha_{j,i} \mathbf{a}_i$, which in turn implies that $\langle \mathbf{a}_j, \theta^* \rangle = \sum_{i \in \mathcal{B}_p} \alpha_{j,i} \langle \mathbf{a}_i, \theta^* \rangle$. Using (12), we then have

$$
\mathbb{E}_v^{\Pi_{\text{DP-BAI}}} \left( \widetilde{\mu}_j^{(p)} \mid \mathcal{A}_p = Q \right) = \mu_j. \tag{61}
$$

If $j \notin \mathcal{B}_p$, we denote

$$
\hat{\mu}_j^{(p)} = \sum_{\iota \in \mathcal{B}_p} \alpha_{j,\iota} \hat{\mu}_\iota^{(p)} \tag{62}
$$

and

$$
\widetilde{\xi}_j^{(p)} = \sum_{\iota \in \mathcal{B}_p} \alpha_{j,\iota} \widetilde{\xi}_\iota^{(p)}. \tag{63}
$$

Note that (62) and (63) still hold in the case of $j \in \mathcal{B}_p$. We define the event

$$
G_1 = \left\{ \hat{\mu}_j^{(p)} - \mu_j \geq \frac{\Delta_j}{4} \right\}.
$$

Then, we have

$$
\mathbb{P}_v^{\Pi_{\text{DP-BAI}}} \left( G_1 \mid \mathcal{A}_p = \mathcal{Q} \right)
$$
$$
= \mathbb{P}_v^{\Pi_{\text{DP-BAI}}} \left( \left\lceil \frac{T'}{Md_Q} \right\rceil (\hat{\mu}_j^{(p)} - \mu_j) \geq \left\lceil \frac{T'}{Md_Q} \right\rceil \frac{\Delta_j}{4} \,\middle|\, \mathcal{A}_p = \mathcal{Q} \right)
$$
$$
= \mathbb{P}_v^{\Pi_{\text{DP-BAI}}} \left( \sum_{\iota \in \mathcal{B}_p} \sum_{s = N_{\iota,T_{p-1}}+1}^{N_{\iota,T_{p-1}} + \left\lceil \frac{T'}{Md_Q} \right\rceil} \alpha_{j,\iota} (X_{\iota,s} - \mu_\iota) \geq \left\lceil \frac{T'}{Md_Q} \right\rceil \frac{\Delta_j}{4} \,\middle|\, \mathcal{A}_p = \mathcal{Q} \right)
$$
$$
\overset{(a)}{\leq} \exp \left( -\frac{2 \left( \left\lceil \frac{T'}{Md_Q} \right\rceil \Delta_j/4 \right)^2}{d_Q \left\lceil \frac{T'}{Md_Q} \right\rceil} \right)
$$

$$\leq \exp\left(-\frac{1}{8}\left\lceil\frac{T'}{Md_Q^2}-1\right\rceil\Delta_j^2\right)$$

$$= \exp\left(-\frac{1}{8}\left\lceil\frac{T'}{M(d_Q^2\wedge s_p)}-1\right\rceil\Delta_j^2\right) \tag{64}$$

where (a) follows Lemma G.1 and Lemma A.1. In addition, we define the event

$$G_2 = \left\{\widetilde{\xi}_j^{(p)}\geq\frac{\Delta_j}{4}\right\}.$$

By Lemma A.3 and Lemma G.1, we have $\widetilde{\xi}_j^{(p)} = \sum_{\iota\in\mathcal{B}_p}\alpha_{j,\iota}\widetilde{\xi}_\iota^{(p)}$ is a sub-exponential random variable with parameters $(\sum_{\iota\in\mathcal{B}_p}\alpha_{j,\iota}^2(\frac{2}{\left\lceil\frac{T'}{Md_Q}\right\rceil\varepsilon})^2, \max_{\iota\in\mathcal{B}_p}|\alpha_{j,\iota}|\frac{2}{\left\lceil\frac{T'}{Md_Q}\right\rceil\varepsilon})$. Then, by Lemma A.4, it holds that for $T'$ sufficiently large

$$\mathbb{P}_v^{\Pi_{\text{DP-BAI}}}\left(G_2\mid\mathcal{A}_p=\mathcal{Q}\right)\leq\exp\left(-\frac{\Delta_j/4}{\frac{4\max_{\iota\in\mathcal{B}_p}|\alpha_{j,\iota}|}{\left\lceil\frac{T'}{Md_Q}\right\rceil\varepsilon}}\right)$$

$$\leq\exp\left(-\frac{\left\lceil\frac{T'}{Md_Q}\right\rceil\varepsilon\Delta_j}{16}\right)$$

$$\leq\exp\left(-\frac{\left\lceil\frac{T'}{M(d_Q^2\wedge s_p)}\right\rceil\varepsilon\Delta_j}{16}\right). \tag{65}$$

Define the events

$$G_3 = \left\{\hat{\mu}_{i^*(v)}^{(p)}-\mu_{i^*(v)}\leq-\frac{\Delta_j}{4}\right\}\quad\text{and}\quad G_4 = \left\{\widetilde{\xi}_{i^*(v)}^{(p)}\leq-\frac{\Delta_j}{4}\right\}.$$

Similar to (64) and (65), we have

$$\mathbb{P}_v^{\Pi_{\text{DP-BAI}}}\left(G_3\mid\mathcal{A}_p=\mathcal{Q}\right)\leq\exp\left(-\frac{1}{8}\left\lceil\frac{T'}{M(d_Q^2\wedge s_p)}-1\right\rceil\Delta_j^2\right) \tag{66}$$

and for sufficiently large $T'$,

$$\mathbb{P}_v^{\Pi_{\text{DP-BAI}}}\left(G_4\mid\mathcal{A}_p=\mathcal{Q}\right)\leq\exp\left(-\frac{\left\lceil\frac{T'}{M(d_Q^2\wedge s_p)}\right\rceil\varepsilon\Delta_j}{16}\right). \tag{67}$$

Hence, in Case 2, for sufficiently large $T'$ we have

$$\mathbb{P}_v^{\Pi_{\text{DP-BAI}}}\left(\widetilde{\mu}_j^{(p)}\geq\widetilde{\mu}_{i^*(v)}^{(p)}\mid\mathcal{A}_p=\mathcal{Q}\right)$$

$$\leq\mathbb{P}_v^{\Pi_{\text{DP-BAI}}}\left(G_1\cup G_2\cup G_3\cup G_4\mid\mathcal{A}_p=\mathcal{Q}\right)$$

$$\leq 2\exp\left(-\frac{1}{8}\left\lceil\frac{T'}{M(d_Q^2\wedge s_p)}-1\right\rceil\Delta_j^2\right)+2\exp\left(-\frac{1}{16}\left\lceil\frac{T'}{M(d_Q^2\wedge s_p)}\right\rceil\varepsilon\Delta_j\right) \tag{68}$$

Finally, combining both Cases 1 and 2, for sufficiently large $T'$, we have

$$\mathbb{P}_v^{\Pi_{\text{DP-BAI}}}\left(\widetilde{\mu}_j^{(p)}\geq\widetilde{\mu}_{i^*(v)}^{(p)}\mid\mathcal{A}_p=\mathcal{Q}\right)$$

$$\leq 2\exp\left(-\frac{1}{8}\left\lceil\frac{T'}{M(d_Q^2\wedge s_p)}-1\right\rceil\Delta_j^2\right)+2\exp\left(-\frac{1}{16}\left\lceil\frac{T'}{M(d_Q^2\wedge s_p)}\right\rceil\varepsilon\Delta_j\right).$$

$$\leq 2\exp\left(-\frac{1}{16}\left\lceil\frac{T'}{M(d_Q^2\wedge s_p)}\right\rceil\Delta_j^2\right)+2\exp\left(-\frac{1}{16}\left\lceil\frac{T'}{M(d_Q^2\wedge s_p)}\right\rceil\varepsilon\Delta_j\right).$$

$$\square$$

**Lemma G.3.** *Fix instance $v \in \mathcal{P}$ and $p \in [M_1]$. Recall the definitions of $\lambda$ in (4) and $g_0$ in (8). For any set $\mathcal{Q} \subset [K]$ with $|\mathcal{Q}| = s_p$ and $i^*(v) \in Q$, it holds when $T'$ is sufficiently large*

$$\mathbb{P}_v^{\Pi_{\text{DP-BAI}}} \left( i^*(v) \notin \mathcal{A}_{p+1} \mid \mathcal{A}_p = \mathcal{Q} \right)$$
$$\leq 2\lambda \left( \exp\left( -\frac{1}{16} \left\lceil \frac{T'}{M(d_{\mathcal{Q}}^2 \wedge s_p)} \right\rceil \Delta_{(g_0)}^2 \right) + \exp\left( -\frac{1}{16} \left\lceil \frac{T'}{M(d_{\mathcal{Q}}^2 \wedge s_p)} \right\rceil \Delta_{(g_0)}\varepsilon \right) \right) \quad (69)$$

*Proof.* Fix $v \in \mathcal{P}$, $p \in [M_1]$ and $\mathcal{Q} \subset [K]$ with $|\mathcal{Q}| = s_p$ and $i^*(v) \in Q$. Let $\mathcal{Q}^{\text{sub}}$ be the set of $s_p - g_0 + 1$ arms in $\mathcal{Q}$ with largest suboptimal gap. In addition, let

$$N^{\text{sub}} = \sum_{j \in \mathcal{Q}^{\text{sub}}} \mathbf{1}_{\{\widetilde{\mu}_j^{(p)} \geq \widetilde{\mu}_{i^*(v)}^{(p)}\}}$$

be the number of arms in $\mathcal{Q}^{\text{sub}}$ with private empirical mean larger than the best arm. Then, we have

$$\mathbb{E}_v^{\Pi_{\text{DP-BAI}}} \left( N^{\text{sub}} \mid \mathcal{A}_p = \mathcal{Q} \right)$$
$$= \sum_{j \in \mathcal{Q}^{\text{sub}}} \mathbb{E}_v^{\Pi_{\text{DP-BAI}}} \left( \mathbf{1}_{\{\widetilde{\mu}_j^{(p)} \geq \widetilde{\mu}_{i^*(v)}^{(p)}\}} \mid \mathcal{A}_p = \mathcal{Q} \right)$$
$$\leq \sum_{j \in \mathcal{Q}^{\text{sub}}} \mathbb{P}_v^{\Pi_{\text{DP-BAI}}} \left( \widetilde{\mu}_j^{(p)} \geq \widetilde{\mu}_{i^*(v)}^{(p)} \mid \mathcal{A}_p = \mathcal{Q} \right)$$
$$\overset{(a)}{\leq} \sum_{j \in Q^{\text{sub}}} 2 \left( \exp\left( -\frac{1}{16} \left\lceil \frac{T'}{M(d_Q^2 \wedge s_p)} \right\rceil \Delta_j^2 \right) + \exp\left( -\frac{1}{16} \left\lceil \frac{T'}{M(d_Q^2 \wedge s_p)} \right\rceil \varepsilon\Delta_j \right) \right)$$
$$\overset{(b)}{\leq} 2|\mathcal{Q}^{\text{sub}}| \left( \exp\left( -\frac{1}{16} \left\lceil \frac{T'}{M(d_Q^2 \wedge s_p)} \right\rceil \Delta_{(g_0)}^2 \right) + \exp\left( -\frac{1}{16} \left\lceil \frac{T'}{M(d_Q^2 \wedge s_p)} \right\rceil \varepsilon\Delta_{(g_0)} \right) \right)$$
$$= 2(s_p - g_0 + 1) \left( \exp\left( -\frac{1}{16} \left\lceil \frac{T'}{M(d_Q^2 \wedge s_p)} \right\rceil \Delta_{(g_0)}^2 \right) \right.$$
$$\left. + \exp\left( -\frac{1}{16} \left\lceil \frac{T'}{M(d_Q^2 \wedge s_p)} \right\rceil \varepsilon\Delta_{(g_0)} \right) \right), \quad (70)$$

where (a) follows Lemma G.2, and (b) follows from the fact that $\min_{j \in \mathcal{Q}^{\text{sub}}} \Delta_j \geq \Delta_{(g_0)}$.

Then,

$$\mathbb{P}_v^{\Pi_{\text{DP-BAI}}} \left( i^*(v) \notin \mathcal{A}_{p+1} \mid \mathcal{A}_p = \mathcal{Q} \right)$$
$$\overset{(a)}{\leq} \mathbb{P}_v^{\Pi_{\text{DP-BAI}}} \left( N^{\text{sub}} \geq s_{p+1} - g_0 + 1 \mid \mathcal{A}_p = \mathcal{Q} \right)$$
$$\overset{(b)}{\leq} \frac{\mathbb{E}_v^{\Pi_{\text{DP-BAI}}} \left( N^{\text{sub}} \mid \mathcal{A}_p = \mathcal{Q} \right)}{s_{p+1} - g_0 + 1}$$
$$\overset{(c)}{\leq} \frac{2(s_p - g_0 + 1)}{s_{p+1} - g_0 + 1} \left( \exp\left( -\frac{1}{16} \left\lceil \frac{T'}{M(d_Q^2 \wedge s_p)} \right\rceil \Delta_{(g_0)}^2 \right) + \exp\left( -\frac{1}{16} \left\lceil \frac{T'}{M(d_Q^2 \wedge s_p)} \right\rceil \varepsilon\Delta_{(g_0)} \right) \right)$$
$$\overset{(d)}{\leq} \frac{2(h_p + 1)}{h_{p+1} + 1} \left( \exp\left( -\frac{1}{16} \left\lceil \frac{T'}{M(d_Q^2 \wedge s_p)} \right\rceil \Delta_{(g_0)}^2 \right) + \exp\left( -\frac{1}{16} \left\lceil \frac{T'}{M(d_Q^2 \wedge s_p)} \right\rceil \varepsilon\Delta_{(g_0)} \right) \right)$$
$$\overset{(e)}{\leq} 2\lambda \left( \exp\left( -\frac{1}{16} \left\lceil \frac{T'}{M(d_Q^2 \wedge s_p)} \right\rceil \Delta_{(g_0)}^2 \right) + \exp\left( -\frac{1}{16} \left\lceil \frac{T'}{M(d_Q^2 \wedge s_p)} \right\rceil \varepsilon\Delta_{(g_0)} \right) \right)$$

where (a) follows from the fact that $N^{\text{sub}} \geq s_{p+1} - g_0 + 1$ is a necessary condition for $i^*(v) \notin \mathcal{A}_{p+1}$ when $\mathcal{A}_p = \mathcal{Q}$, (b) follows Markov's inequality, (c) is obtained from (70). In addition, (d) is obtained from the definition in (7), and (e) is obtained from the definition in (6).

$\square$

**Lemma G.4.** *Fix instance $v \in \mathcal{P}$ and $p \in \{M_1 + 1, \ldots, M\}$. For any set $\mathcal{Q} \subset [K]$ with $|\mathcal{Q}| = s_p$ and $i^*(v) \in Q$, it holds that when $T'$ is sufficiently large*

$$\mathbb{P}_v^{\Pi_{\text{DP-BAI}}} \left( i^*(v) \notin \mathcal{A}_{p+1} \,\big|\, \mathcal{A}_p = \mathcal{Q} \right)$$
$$\leq 6 \left( \exp\left( -\frac{1}{16} \left\lceil \frac{T'}{M(d_Q^2 \wedge s_p)} \right\rceil \Delta_{(s_{p+2}+1)}^2 \right) + \exp\left( -\frac{1}{16} \left\lceil \frac{T'}{M(d_Q^2 \wedge s_p)} \right\rceil \varepsilon\Delta_{(s_{p+2}+1)} \right) \right), \tag{71}$$

*where we define $s_{M+2} = 1$.*

*Proof.* Fix $v \in \mathcal{P}$, $p \in [M_1]$ and $\mathcal{Q} \subset [K]$ with $|\mathcal{Q}| = s_p$ and $i^*(v) \in Q$. Similarly, let $\mathcal{Q}^{\text{sub}}$ be the set of $s_p - s_{p+2}$ arms in $\mathcal{Q}$ with the largest suboptimality gaps. Again, let

$$N^{\text{sub}} = \sum_{j \in \mathcal{Q}^{\text{sub}}} \mathbf{1}_{\{\widetilde{\mu}_j^{(p)} \geq \widetilde{\mu}_{i^*(v)}^{(p)}\}}.$$

Then, we have

$$\mathbb{E}_v^{\Pi_{\text{DP-BAI}}} \left( N^{\text{sub}} \,\big|\, \mathcal{A}_p = \mathcal{Q} \right)$$

$$= \sum_{j \in \mathcal{Q}^{\text{sub}}} \mathbb{E}_v^{\Pi_{\text{DP-BAI}}} \left( \mathbf{1}_{\{\widetilde{\mu}_j^{(p)} \geq \widetilde{\mu}_{i^*(v)}^{(p)}\}} \,\big|\, \mathcal{A}_p = \mathcal{Q} \right)$$

$$\leq \sum_{j \in \mathcal{Q}^{\text{sub}}} \mathbb{P}_v^{\Pi_{\text{DP-BAI}}} \left( \widetilde{\mu}_j^{(p)} \geq \widetilde{\mu}_{i^*(v)}^{(p)} \,\big|\, \mathcal{A}_p = \mathcal{Q} \right)$$

$$\overset{(a)}{\leq} \sum_{j \in Q^{\text{sub}}} 2 \left( \exp\left( -\frac{1}{16} \left\lceil \frac{T'}{M(d_Q^2 \wedge s_p)} \right\rceil \Delta_j^2 \right) + \exp\left( -\frac{1}{16} \left\lceil \frac{T'}{M(d_Q^2 \wedge s_p)} \right\rceil \varepsilon\Delta_j \right) \right)$$

$$\overset{(b)}{\leq} 2|\mathcal{Q}^{\text{sub}}| \left( \exp\left( -\frac{1}{16} \left\lceil \frac{T'}{M(d_Q^2 \wedge s_p)} \right\rceil \Delta_{(s_{p+2}+1)}^2 \right) + \exp\left( -\frac{1}{16} \left\lceil \frac{T'}{M(d_Q^2 \wedge s_p)} \right\rceil \varepsilon\Delta_{(s_{p+2}+1)} \right) \right)$$

$$= 2(s_p - s_{p+2}) \left( \exp\left( -\frac{1}{16} \left\lceil \frac{T'}{M(d_Q^2 \wedge s_p)} \right\rceil \Delta_{(s_{p+2}+1)}^2 \right) \right.$$

$$\left. + \exp\left( -\frac{1}{16} \left\lceil \frac{T'}{M(d_Q^2 \wedge s_p)} \right\rceil \varepsilon\Delta_{(s_{p+2}+1)} \right) \right), \tag{72}$$

where (a) follows from Lemma G.2, and (b) follows from the fact that $\min_{j \in \mathcal{Q}^{\text{sub}}} \Delta_j \geq \Delta_{s_{p+2}+1}$.

Similarly, we can have

$$\mathbb{P}_v^{\Pi_{\text{DP-BAI}}} \left( i^*(v) \notin \mathcal{A}_{p+1} \,\big|\, \mathcal{A}_p = \mathcal{Q} \right)$$

$$\overset{(a)}{\leq} \mathbb{P}_v^{\Pi_{\text{DP-BAI}}} \left( N^{\text{sub}} \geq s_{p+1} - s_{p+2} + 1 \,\big|\, \mathcal{A}_p = \mathcal{Q} \right)$$

$$\overset{(b)}{\leq} \frac{\mathbb{E}_v^{\Pi_{\text{DP-BAI}}} \left( N^{\text{sub}} \,\big|\, \mathcal{A}_p = \mathcal{Q} \right)}{s_{p+1} - s_{p+2} + 1}$$

$$\overset{(c)}{\leq} \frac{2(s_p - s_{p+2})}{s_{p+1} - s_{p+2} + 1} \left( \exp\left( -\frac{1}{16} \left\lceil \frac{T'}{M(d_Q^2 \wedge s_p)} \right\rceil \Delta_{(s_{p+2}+1)}^2 \right) \right.$$

$$\left. + \exp\left( -\frac{1}{16} \left\lceil \frac{T'}{M(d_Q^2 \wedge s_p)} \right\rceil \varepsilon\Delta_{(s_{p+2}+1)} \right) \right)$$

$$\overset{(d)}{\leq} 6 \left( \exp\left( -\frac{1}{16} \left\lceil \frac{T'}{M(d_Q^2 \wedge s_p)} \right\rceil \Delta_{(s_{p+2}+1)}^2 \right) + \exp\left( -\frac{1}{16} \left\lceil \frac{T'}{M(d_Q^2 \wedge s_p)} \right\rceil \varepsilon\Delta_{(s_{p+2}+1)} \right) \right),$$

where (a) follows from the fact that $N^{\text{sub}} \geq s_{p+1} - s_{p+2} + 1$ is the necessary condition of $i^*(v) \notin \mathcal{A}_{p+1}$ when $\mathcal{A}_p = \mathcal{Q}$, (b) follows from Markov's inequality, (c) follows (72), and (d) is obtained from the definition in (7).

$\square$

With the above ingredients in place, we are now ready to prove Theorem 4.2.

*Proof of Theorem 4.2.* Fix instance $v \in \mathcal{P}$. Recall that in DP-BAI, the decision maker eliminates arms in successive phases, and the decision maker can successfully identify the best arm if and only if it is not eliminated in any of the phases. That is,

$$\mathbb{P}_v^{\Pi_{\text{DP-BAI}}} \left( \hat{I}_T \neq i^*(v) \right) \leq \sum_{p=1}^{M} \mathbb{P}_v^{\Pi_{\text{DP-BAI}}} \left( i^*(v) \notin \mathcal{A}_{p+1} \mid i^*(v) \in \mathcal{A}_p \right). \tag{73}$$

Then, we divide the rightmost sum of the probabilities into two parts. Let

$$P_1 = \sum_{p=1}^{M_1} \mathbb{P}_v^{\Pi_{\text{DP-BAI}}} \left( i^*(v) \notin \mathcal{A}_{p+1} \mid i^*(v) \in \mathcal{A}_p \right)$$

and

$$P_2 = \sum_{p=M_1+1}^{M} \mathbb{P}_v^{\Pi_{\text{DP-BAI}}} \left( i^*(v) \notin \mathcal{A}_{p+1} \mid i^*(v) \in \mathcal{A}_p \right).$$

In the case of $K \leq \lceil d^2/4 \rceil$, by definition we have $M_1 = 0$, which implies that $P_1 = 0$. In the case of $K > \lceil d^2/4 \rceil$ by Lemma G.3, we obtain

$$\begin{aligned} P_1 &\leq 2M_1\lambda \left( \exp\left( -\frac{1}{16} \left\lceil \frac{T'}{Md^2} \right\rceil \Delta^2_{(\lceil d^2/4 \rceil)} \right) + \exp\left( -\frac{1}{16} \left\lceil \frac{T'}{Md^2} \right\rceil \varepsilon\Delta_{(\lceil d^2/4 \rceil)} \right) \right) \\ &\leq 2M_1\lambda \left( \exp\left( -\frac{T'}{64MH_{\text{BAI}}} \right) + \exp\left( -\frac{T'}{64MH_{\text{pri}}} \right) \right) \\ &\leq 4M_1\lambda \exp\left( -\frac{T'}{64MH} \right) \end{aligned} \tag{74}$$

In addition, by Lemma G.4

$$\begin{aligned} P_2 &\leq \sum_{p=M_1+1}^{M} 6 \left( \exp\left( -\frac{1}{16} \left\lceil \frac{T'}{Ms_p} \right\rceil \Delta^2_{(s_{p+2}+1)} \right) + \exp\left( -\frac{1}{16} \left\lceil \frac{T'}{Ms_p} \right\rceil \varepsilon\Delta_{(s_{p+2}+1)} \right) \right) \\ &\leq \sum_{p=M_1+1}^{M} 6 \left( \exp\left( -\frac{T'}{16Ms_p} \Delta^2_{(s_{p+2}+1)} \right) + \exp\left( -\frac{T'}{16Ms_p} \varepsilon\Delta_{(s_{p+2}+1)} \right) \right) \\ &\leq \sum_{p=M_1+1}^{M} 6 \left( \exp\left( -\frac{T'}{16M(4s_{p+2}+4)} \Delta^2_{(s_{p+2}+1)} \right) + \exp\left( -\frac{T'}{16M(4s_{p+2}+4)} \varepsilon\Delta_{(s_{p+2}+1)} \right) \right) \\ &= \sum_{i \in \{s_{p+2}+1: p \in \{M_1+1,\ldots,M\}\}} 6 \left( \exp\left( -\frac{T'}{64Mi} \Delta^2_{(i)} \right) + \exp\left( -\frac{T'}{64Mi} \varepsilon\Delta_{(i)} \right) \right) \\ &\leq 6(M-M_1) \left( \exp\left( -\frac{T'}{64MH_{\text{BAI}}} \right) + \exp\left( -\frac{T'}{64MH_{\text{pri}}} \right) \right) \\ &\leq 12(M-M_1) \exp\left( -\frac{T'}{64MH} \right) \end{aligned} \tag{75}$$

Finally, combining (74) and (75), for sufficiently large $T'$ we have

$$\begin{aligned} \mathbb{P}_v^{\Pi_{\text{DP-BAI}}} \left( \hat{I}_T \neq i^*(v) \right) &\leq P_1 + P_2 \\ &\leq (12M + 4M_1\lambda) \exp\left( -\frac{T'}{64MH} \right) \\ &\leq \exp\left( -\frac{T'}{65MH} \right). \end{aligned} \tag{76}$$

$\square$

# H    PROOF OF THEOREM 4.7

Let $\mathcal{P}'$ be the set of problem instances that meet the following properties: (a) the best arm is unique; (b) the feature vectors $\{\mathbf{a}_i\}_{i=1}^K \subset \mathbb{R}^d$ are mutually orthogonal; (c) $K = d > 3$ and (d) $\varepsilon < 1$. Hence, we have $\mathcal{P}' \subseteq \mathcal{P}$, and it suffices to consider the instances in $\mathcal{P}'$ to prove Theorem 4.7.

Note that $\log(d) > 1$, $H_{\mathrm{BAI}} > 1$ and $H_{\mathrm{pri}} > 1$ when $v \in \mathcal{P}'$. That is, for any $v \in \mathcal{P}', T > 0, c > 0$, and $\overline{\beta}_i, \underline{\beta}_i \in [0,1]$ with $\overline{\beta}_i \geq \underline{\beta}_i$ for $i \in \{1, 2, 3\}$, we have

$$\exp\left( -\frac{T}{c(\log d)^{\underline{\beta}_1}\left((H_{\mathrm{BAI}})^{\underline{\beta}_2} + (H_{\mathrm{pri}})^{\underline{\beta}_3}\right)} \right) \leq \exp\left( -\frac{T}{c(\log d)^{\overline{\beta}_1}\left((H_{\mathrm{BAI}})^{\overline{\beta}_2} + (H_{\mathrm{pri}})^{\overline{\beta}_3}\right)} \right).$$

Hence, besides we consider the instances in $\mathcal{P}'$, it still suffices to show that (23) holds in the cases of (i) $\beta_1 \in [0,1), \beta_2 = \beta_3 = 1$, (ii) $\beta_2 \in [0,1), \beta_1 = \beta_3 = 1$, and (iii) $\beta_3 \in [0,1), \beta_1 = \beta_2 = 1$. In the following, we will show that cases (i) and (ii) can be satisfied by following the argument of Carpentier and Locatelli (2016), while case (iii) is satisfied by the formula of *change-of-measure* introduced in Subsection 4.3.

First, we present the proof of Lemma 4.3 in the following.

*Proof of Lemma 4.3.* Fix a problem instance $v \in \mathcal{P}$, policy $\pi$, $n \in \mathbb{N}$, and $\iota \in [K]$, and event $E = \{\hat{I}_T \in \mathcal{A}\} \cap \{N_{\iota,T} < n\}$ for arbitrary $\mathcal{A} \subseteq [K]$.

By the definition of early stopping policy, we have

$$\mathbb{P}_v^\pi(\{N_{\iota,T} < n\}) = \mathbb{P}_v^{\mathrm{ES}(\pi,n,\iota)}(\{N_{\iota,T} < n\}) \tag{77}$$

and

$$\mathbb{P}_v^\pi(\{\hat{I}_T \in \mathcal{A}\} \mid \{N_{\iota,T} < n\}) = \mathbb{P}_v^{\mathrm{ES}(\pi,n,\iota)}(\{\hat{I}_T \in \mathcal{A}\} \mid \{N_{\iota,T} < n\}). \tag{78}$$

By the chain rule, we have

$$\mathbb{P}_v^\pi(E) = \mathbb{P}_v^\pi(\{N_{\iota,T} < n\})\mathbb{P}_v^\pi(\{\hat{I}_T \in \mathcal{A}\} \mid \{N_{\iota,T} < n\}) \tag{79}$$

and

$$\mathbb{P}_v^{\mathrm{ES}(\pi,n,\iota)}(E) = \mathbb{P}_v^{\mathrm{ES}(\pi,n,\iota)}(\{N_{\iota,T} < n\})\mathbb{P}_v^{\mathrm{ES}(\pi,n,\iota)}(\{\hat{I}_T \in \mathcal{A}\} \mid \{N_{\iota,T} < n\}). \tag{80}$$

That is,

$$\mathbb{P}_v^\pi(E) = \mathbb{P}_v^{\mathrm{ES}(\pi,n,\iota)}(E). \tag{81}$$

□

**Lemma H.1** (Adapted from Carpentier and Locatelli (2016)). *Fix $\tilde{K} > 0$, $\tilde{H} > 0$, and a non-private consistent policy $\pi$. For all sufficiently large $T$, there exists an instance $v \in \mathcal{P}'$ with $K \geq \tilde{K}$, $H_{\mathrm{BAI}} > \tilde{H}$ such that,*

$$\mathbb{P}_v^\pi(\hat{I}_T \neq i^*(v)) > \exp\left( -\frac{401T}{\log(K)H_{\mathrm{BAI}}(v)} \right). \tag{82}$$

Note that (82) still holds even with DP constraint, and the privacy parameter $\varepsilon$ will not affect $H_{\mathrm{BAI}}$. Hence, we can have the following corollary.

**Corollary H.2.** *Fix $\tilde{K} > 0$, $\tilde{H} > 0$, $\tilde{\varepsilon} > 0$ and a consistent policy $\pi$. For all sufficiently large $T$, there exists an instance $v \in \mathcal{P}'$ with $K \geq \tilde{K}$, $H_{\mathrm{BAI}} > \tilde{H}$ and $\varepsilon = \tilde{\varepsilon}$ such that,*

$$\mathbb{P}_v^\pi(\hat{I}_T \neq i^*(v)) > \exp\left( -\frac{401T}{\log(K)H_{\mathrm{BAI}}(v)} \right). \tag{83}$$

With the above Corollary, we are now ready to discuss Case (i) in Lemma H.3.

**Lemma H.3.** *Fix any $\beta \in [0,1)$, a consistent policy $\pi$, and a constant $c > 0$. For all sufficiently large $T$, there exists an instance $v \in \mathcal{P}'$ such that,*

$$\mathbb{P}_v^\pi(\hat{I}_T \neq i^*(v)) > \exp\left( -\frac{T}{c(\log K)^\beta(H_{\mathrm{BAI}}(v) + H_{\mathrm{pri}}(v))} \right). \tag{84}$$

*Proof.* Fix $c > 0$, $\beta \in [0, 1)$. Let $\tilde{v}(\tilde{K}, \tilde{H}, \tilde{\varepsilon}, T)$ to be the instance in Corollary H.1 that meets (83). Then, when $\tilde{K} > \exp\left(401^{\frac{1}{1-\beta}}(2c)^{\frac{\beta}{1-\beta}}\right)$, we have

$$\frac{\log(\tilde{K})}{401} > 2c(\log(\tilde{K}))^{\beta}. \tag{85}$$

In addition, by the definition of $H_{\mathrm{pri}}$ we have

$$\lim_{\tilde{\varepsilon} \to +\infty} H_{\mathrm{pri}}(\tilde{v}(\tilde{K}, \tilde{H}, \tilde{\varepsilon}, T)) = 0, \tag{86}$$

which implies that there exists $\varepsilon'(\tilde{K}, \tilde{H}, T)$ such that when $\tilde{\varepsilon} > \varepsilon'(\tilde{K}, \tilde{H}, T)$,

$$H_{\mathrm{pri}}(\tilde{v}(\tilde{K}, \tilde{H}, \tilde{\varepsilon}, T)) < H_{\mathrm{BAI}}(\tilde{v}(\tilde{K}, \tilde{H}, \tilde{\varepsilon}, T)), \tag{87}$$

since $H_{\mathrm{BAI}}$ would not be affected by the privacy parameter.

Finally, combining (83), (85) and (87), we have for all sufficiently large $T$

$$\mathbb{P}_v^{\pi}\left(\hat{I}_T \neq i^*(v)\right) > \exp\left(-\frac{T}{c(\log K)^{\beta}(H_{\mathrm{BAI}}(v) + H_{\mathrm{pri}}(v))}\right), \tag{88}$$

where $v = \tilde{v}(\tilde{K}, \tilde{H}, \tilde{\varepsilon}, T)$, $\tilde{\varepsilon} > \varepsilon'(\tilde{K}, \tilde{H}, T)$, and $\tilde{K} > \exp\left(401^{\frac{1}{1-\beta}}(2c)^{\frac{\beta}{1-\beta}}\right)$. $\qquad\square$

Similarly, we discuss Case (ii) in Lemma H.4.

**Lemma H.4.** *Fix any $\beta \in [0, 1)$, a consistent policy $\pi$, and a constant $c > 0$. For all sufficiently large $T$, there exists an instance $v \in \mathcal{P}'$ such that,*

$$\mathbb{P}_v^{\pi}\left(\hat{I}_T \neq i^*(v)\right) > \exp\left(-\frac{T}{c(\log K)(H_{\mathrm{BAI}}(v)^{\beta} + H_{\mathrm{pri}}(v))}\right). \tag{89}$$

*Proof.* Again, fix $c > 0$, $\beta \in [0, 1)$. Recall that $\tilde{v}(\tilde{K}, \tilde{H}, \tilde{\varepsilon}, T)$ is the instance in Corollary H.1 that meets (83). Similarly, when $\tilde{H} > (802c)^{\frac{1}{1-\beta}}$, we have

$$\frac{\tilde{H}}{401} > 2c(\tilde{H})^{\beta}. \tag{90}$$

In addition, we have

$$\lim_{\tilde{\varepsilon} \to +\infty} H_{\mathrm{pri}}(\tilde{v}(\tilde{K}, \tilde{H}, \tilde{\varepsilon}, T)) = 0, \tag{91}$$

which implies that there exists $\varepsilon'(\tilde{K}, \tilde{H}, T, \beta)$ such that when $\tilde{\varepsilon} > \varepsilon'(\tilde{K}, \tilde{H}, T, \beta)$,

$$H_{\mathrm{pri}}(\tilde{v}(\tilde{K}, \tilde{H}, \tilde{\varepsilon}, T)) < H_{\mathrm{BAI}}(\tilde{v}(\tilde{K}, \tilde{H}, \tilde{\varepsilon}, T))^{\beta}. \tag{92}$$

Finally, combining 83, (90) and (92), we have for all sufficiently large $T$

$$\mathbb{P}_v^{\pi}\left(\hat{I}_T \neq i^*(v)\right) > \exp\left(-\frac{T}{c(\log K)(H_{\mathrm{BAI}}(v)^{\beta} + H_{\mathrm{pri}}(v))}\right), \tag{93}$$

where $v = \tilde{v}(\tilde{K}, \tilde{H}, \tilde{\varepsilon}, T)$, $\tilde{\varepsilon} > \varepsilon'(\tilde{K}, \tilde{H}, T, \beta)$, and $\tilde{H} > (802c)^{\frac{1}{1-\beta}}$. $\qquad\square$

Furthermore, before we discuss (iii), we introduce the following lemma.

**Lemma H.5.** *Fix $\tilde{H} > 0$, $\beta \in [0, 1)$, a consistent policy $\pi$, and $\tilde{K} > 3$. For all sufficiently large $T$, there exists an instance $v \in \mathcal{P}'$ with $K = \tilde{K}$, $H_{\mathrm{pri}} \geq \tilde{H}$ and $H_{\mathrm{pri}} \geq (H_{\mathrm{BAI}})^{\frac{1}{\beta}}$ such that,*

$$\mathbb{P}_v^{\pi}\left(\hat{I}_T \neq i^*(v)\right) > \exp\left(-\frac{97T}{\log(K)H_{\mathrm{pri}}(v)}\right). \tag{94}$$

*Proof.* Fix $\tilde{H} > 0$, a consistent policy $\pi$, and $\tilde{K} > 3$. Fix $\gamma_1 \in (0, (\frac{1}{\tilde{K}})^{\tilde{K}+2})$, and let $\gamma_i = \gamma_{i-1}\tilde{K}$ for $i = 2, \ldots, \tilde{K}$. We construct $\tilde{K} + 1$ problem instances $v^{(0)}, v^{(1)}, \ldots, v^{(\tilde{K})} \in \mathcal{P}'$ with $K = \tilde{K}$ arms as follows. For instance $j = 0, \ldots, \tilde{K}$, the reward distribution of arm $i \in [\tilde{K}]$ is

$$\nu_i^{(j)} = \begin{cases} \mathrm{Ber}(\frac{1}{2} - \gamma_i) & \text{if } i \neq j \\ \mathrm{Ber}(\frac{1}{2} + \gamma_i) & \text{if } i = j \end{cases}, \tag{95}$$

where Ber is denoted to be Bernoulli distribution. In addition, the privacy parameters of these $\tilde{K} + 1$ instances are set to be the same, and it is denoted by $\tilde{\varepsilon}$. By the fact that $H_{\mathrm{pri}}(v^{(j)}) \to +\infty$ as $\tilde{\varepsilon} \to 0$ for all $j = 0, \ldots, \tilde{K}$. Then, we choose $\tilde{\varepsilon}$ to be any value such that for all $j = 0, \ldots, \tilde{K}$

$$H_{\mathrm{pri}}(v^{(j)}) \geq \tilde{H} \quad \text{and} \quad H_{\mathrm{pri}}(v^{(j)}) \geq (H_{\mathrm{BAI}}(v^{(j)}))^{\frac{1}{\beta}}. \tag{96}$$

By the fact that $\pi$ is a consistent policy, when $T$ is sufficiently large, we have

$$\mathbb{P}_{v^{(0)}}^{\pi}(\hat{I}_T = 1) > \frac{1}{2}. \tag{97}$$

Furthermore, we denote $n_i = 4\mathbb{E}_{v^{(0)}}^{\pi}(N_{i,T})$ to be the expected number of pulls for arm $i \in [\tilde{K}]$ in instance 0 with time budget $T$. Then, we define event $E_i = \{N_{i,T} < n_i\} \cap \{\hat{I}_T = 1\}$. It then holds that

$$\begin{aligned} \mathbb{P}_{v^{(0)}}^{\pi}(E_i) &= 1 - \mathbb{P}_{v^{(0)}}^{\pi}(\neg E_i) \\ &\overset{(a)}{\geq} 1 - \mathbb{P}_{v^{(0)}}^{\pi}(N_{i,T} \geq 4n_i) - \mathbb{P}_{v^{(0)}}^{\pi}(\hat{I}_T \neq 1) \\ &\overset{(b)}{\geq} 1 - \mathbb{P}_{v^{(0)}}^{\pi}(N_{i,T} \geq 4n_i) - \frac{1}{2} \\ &\overset{(c)}{\geq} 1 - \frac{1}{4} - \frac{1}{2} \\ &= \frac{1}{4}, \end{aligned} \tag{98}$$

where (a) follows from the union bound, (b) follows (97), and (c) is obtained from Markov's inequality. Then, by Lemma 4.3, we have

$$\mathbb{P}_{v^{(0)}}^{\mathrm{ES}(\pi, n_i, i)}(E_i) = \mathbb{P}_{v^{(0)}}^{\pi}(E_i) \geq \frac{1}{4}. \tag{99}$$

In addition, by Lemma 4.5, we have for $i = 1, \ldots, \tilde{K}$

$$\begin{aligned} \mathbb{P}_{v^{(i)}}^{\mathrm{ES}(\pi, n_i, i)}(E_i) &\geq \exp\left(-6\varepsilon n_i \mathrm{TV}(\nu_i^{(i)}, \nu_i^{(0)})\right)\mathbb{P}_{v^{(0)}}^{\mathrm{ES}(\pi, n_i, i)}(E_i) \\ &\overset{(a)}{\geq} \frac{1}{4} \exp\left(-6\varepsilon n_i \mathrm{TV}(\nu_i^{(i)}, \nu_i^{(0)})\right) \\ &\overset{(b)}{\geq} \frac{1}{4} \exp(-12\varepsilon n_i \gamma_i), \end{aligned} \tag{100}$$

where (a) follows (99), and (b) is obtained by the definition in (95).

In other words, we have

$$\begin{aligned} \mathbb{P}_{v^{(i)}}^{\pi}(\hat{I}_T \neq i^*(v^{(i)})) &\geq \mathbb{P}_{v^{(i)}}^{\pi}(E_i) \\ &\overset{(a)}{=} \mathbb{P}_{v^{(i)}}^{\mathrm{ES}(\pi, n_i, i)}(E_i) \\ &\overset{(b)}{\geq} \frac{1}{4} \exp(-12\varepsilon n_i \gamma_i), \end{aligned} \tag{101}$$

where (a) follows Lemma 4.3, and (b) follows (100).

Observe that for any $\{x_i\}_{i=1}^{\tilde{K}-1} \subset \mathbb{R}$ and $\{y_i\}_{i=1}^{\tilde{K}-1} \subset \mathbb{R}^+$, it holds that $\min_{i \in [\tilde{K}-1]} \frac{x_i}{y_i} \leq \frac{\sum_{i \in [\tilde{K}-1]} x_i}{\sum_{i \in [\tilde{K}-1]} y_i}$. Then, by letting $x_i = n_{i+1}$ and $y_i = \frac{1}{\varepsilon \gamma_{i+1} H_{\mathrm{pri}}(v^{(i+1)})}$, we conclude that there exists $i' \in \{2, \ldots, \tilde{K}\}$ such that

$$n_{i'} \varepsilon \gamma_{i'} H_{\mathrm{pri}}(v^{(i')}) \leq \left(\sum_{i=2}^{\tilde{K}} n_i\right) \bigg/ \left(\sum_{i=2}^{\tilde{K}} \frac{1}{\varepsilon \gamma_i H_{\mathrm{pri}}(v^{(i)})}\right)$$

$$\overset{(a)}{\le} 4T/\Big(\sum_{i=2}^{\tilde{K}} \frac{1}{\varepsilon\gamma_i H_{\mathrm{pri}}(v^{(i)})}\Big)$$

$$\overset{(b)}{\le} 4T/\Big(\sum_{i=2}^{\tilde{K}} \frac{1}{i}\Big)$$

$$\overset{(c)}{\le} \frac{4T}{\log(\tilde{K}) - \log(2)}$$

$$\overset{(d)}{\le} \frac{8T}{\log(\tilde{K})} \tag{102}$$

where (a) follows from the fact that $\sum_{i=2}^{\tilde{K}} \mathbb{E}_{v^{(0)}}(N_{i,T}) \le T$, and (b) is obtained from $H_{\mathrm{pri}}(v^{(i)}) \le \frac{i}{\varepsilon\gamma_i}$, (c) follows from the fact that $\log(\tilde{K}) - \log(2) = \int_2^{\tilde{K}} \frac{1}{x}\,\mathrm{d}x < \sum_{i=2}^{\tilde{K}} \frac{1}{i}$, and (d) is obtained by the assumption that $\tilde{K} > 3$.

Combining (101) and (102), we have

$$\begin{aligned}
\mathbb{P}_{v^{(i')}}^{\pi}(\hat{I}_T \ne i^*(v^{(i')})) &\ge \frac{1}{4}\exp(-12\varepsilon n_{i'}\gamma_{i'}) \\
&= \frac{1}{4}\exp\left(-\frac{12}{H_{\mathrm{pri}}(v^{(i')})}\varepsilon n_{i'}\gamma_i H_{\mathrm{pri}}(v^{(i')})\right) \\
&\ge \frac{1}{4}\exp\left(-\frac{96T}{\log(\tilde{K})H_{\mathrm{pri}}(v^{(i')})}\right).
\end{aligned} \tag{103}$$

That is, when $T$ is sufficiently large, we have

$$\mathbb{P}_{v^{(i')}}^{\pi}(\hat{I}_T \ne i^*(v^{(i')})) \ge \exp\left(-\frac{97T}{\log(\tilde{K})H_{\mathrm{pri}}(v^{(i')})}\right).$$

$\square$

With Lemma H.5, we are now ready to show that Theorem 4.7 holds in Case (iii) in the following lemma.

**Lemma H.6.** *Fix any $\beta \in [0, 1)$, a consistent policy $\pi$, and a constant $c > 0$. For all sufficiently large $T$, there exists an instance $v \in \mathcal{P}'$ such that,*

$$\mathbb{P}_v^{\pi}(\hat{I}_T \ne i^*(v)) > \exp\left(-\frac{T}{c(\log K)(H_{\mathrm{BAI}}(v) + H_{\mathrm{pri}}(v)^\beta)}\right). \tag{104}$$

*Proof.* Fix $c > 0$, $\beta \in [0, 1)$. Similarly, let $\tilde{v}(\tilde{K}, \tilde{H}, \beta, T)$ be the instance specified in Lemma H.5 that meets (94). Then, when $\tilde{H} > (194c)^{\frac{1}{1-\beta}}$, we have

$$\frac{\tilde{H}}{97} > 2c(\tilde{H})^\beta. \tag{105}$$

Finally, combining (94), (96) and (105), we have for all sufficiently large $T$

$$\mathbb{P}_v^{\pi}(\hat{I}_T \ne i^*(v)) > \exp\left(-\frac{T}{c(\log K)(H_{\mathrm{BAI}}(v) + H_{\mathrm{pri}}(v)^\beta)}\right), \tag{106}$$

where $v = \tilde{v}(\tilde{K}, \tilde{H}, \beta, T)$, and $\tilde{H} > (194c)^{\frac{1}{1-\beta}}$. $\square$

*Proof of Theorem 4.7.* Finally, by combining Lemmas H.3, H.4 and H.6, we see that Theorem 4.7 holds. $\square$

# I   PROOF OF PROPOSITION D.2

The proof of Proposition D.2 shares some similarities as the proof of Proposition 4.1, and the difference is using the property of Gaussian mechanism instead of Laplacian mechanism.

Let $N_i^{(p)} := N_{i,T_p}$. Under the (random) process of DP-BAI-GAUSS, we introduce an auxiliary random mechanism $\tilde{\pi}$ whose input is $\mathbf{x} \in \mathcal{X}$, and whose output is $(N_i^{(P)}, \widetilde{\mu}_i^{(p)})_{i \in [K], p \in [M]}$, where we define $\widetilde{\mu}_i^{(p)} = 0$ if $i \notin \mathcal{A}_p$. By Dwork et al. (2014, Proposition 2.1), to prove Proposition D.2, it suffices to show that $\tilde{\pi}$ is $(\varepsilon, \delta)$-differentially private, i.e., for all neighbouring $\mathbf{x}, \mathbf{x}' \in \mathcal{X}$,

$$\mathbb{P}^{\tilde{\pi}}(\tilde{\pi}(\mathbf{x}) \in \mathcal{S}) \le e^\varepsilon \, \mathbb{P}^{\tilde{\pi}}(\tilde{\pi}(\mathbf{x}') \in \mathcal{S}) + \delta \quad \forall \mathcal{S} \subseteq \tilde{\Omega}, \tag{107}$$

where $\tilde{\Omega}$ is the set of all possible outputs of $\tilde{\pi}$. In the following, we write $\mathbb{P}_{\mathbf{x}}^{\tilde{\pi}}$ to denote the probability measure induced under $\tilde{\pi}$ with input $\mathbf{x} \in \mathcal{X}$.

Fix neighbouring $\mathbf{z}, \mathbf{z}' \in \mathcal{X}$ arbitrarily. There exists a unique pair $(\underline{i}, \underline{n})$ with $\mathbf{z}_{\underline{i},\underline{n}} \ne \mathbf{z}'_{\underline{i},\underline{n}}$. In addition, fix any $n_i^{(p)} \in \mathbb{N}$ and Borel set $\chi_i^{(p)} \subseteq [0, 1]$ for $i \in [K]$ and $p \in [M]$ such that

$$0 \le n_i^{(1)} \le n_i^{(2)} \le \ldots \le n_i^{(M)} \le T \quad \forall i \in [K].$$

Let

$$\underline{p} = \begin{cases} M + 1, & \text{if } n_{\underline{i}}^{(M)} < \underline{n} \\ \min\{p \in [M] : n_{\underline{i}}^{(p)} \ge \underline{n}\}, & \text{otherwise.} \end{cases} \tag{108}$$

For any $j \in [M]$, we define the event

$$D_j = \{(i, p) \in [K] \times [j] : \ N_i^{(p)} = n_i^{(p)}, \widetilde{\mu}_i^{(p)} \in \chi_i^{(p)}\}$$

and

$$D'_j = D_{j-1} \cap \{N_{\underline{i}}^{(j)} = n_{\underline{i}}^{(j)}\} \cap \{i \in [K] \setminus \{\underline{i}\} : \ N_i^{(j)} = n_i^{(j)}, \widetilde{\mu}_i^{(j)} \in \chi_i^{(j)}\}.$$

For $j' \in \{j + 1, \ldots, M\}$

$$D_{j,j'} = \{(i, p) \in [K] \times \{j + 1, \ldots, j'\} : \ N_i^{(p)} = n_i^{(p)}, \widetilde{\mu}_i^{(p)} \in \chi_i^{(p)}\}$$

In particular, $D_0$ and $D_{M,M}$ are events with probability 1, i.e.,

$$\mathbb{P}_{\mathbf{x}}^{\tilde{\pi}}(D_0 \cap D_{M,M}) = 1 \quad \forall \mathbf{x} \in \mathcal{X}.$$

Then, if $n_{\underline{i}}^{(M)} < \underline{n}$, by the definition of $\tilde{\pi}$, we have

$$\mathbb{P}_{\mathbf{z}}^{\tilde{\pi}}(D_M) = \mathbb{P}_{\mathbf{z}'}^{\tilde{\pi}}(D_M). \tag{109}$$

In the following, we assume $n_{\underline{i}}^{(M)} \ge \underline{n}$ and we will show

$$\mathbb{P}_{\mathbf{z}}^{\tilde{\pi}}(D_M) \le \exp(\varepsilon) \mathbb{P}_{\mathbf{z}'}^{\tilde{\pi}}(D_M) \tag{110}$$

which then implies that $\tilde{\pi}$ is $\varepsilon$-differentially private.

For $\mathbf{x} \in \{\mathbf{z}, \mathbf{z}'\}$, we denote

$$\mu_{\mathbf{x}} = \frac{1}{n_{\underline{i}}^{(p)} - n_{\underline{i}}^{(p-1)}} \sum_{j=n_{\underline{i}}^{(p-1)}+1}^{n_{\underline{i}}^{(p)}} \mathbf{x}_{\underline{i},j}$$

That is, $\mu_{\mathbf{x}}$ is the value of $\hat{\mu}_{\underline{i}}^{(p)}$ in the condition of $D'_{\underline{p}}$ under probability measure $\mathbb{P}_{\mathbf{x}}^{\tilde{\pi}}$.

Then, by the chain rule we have for both $\mathbf{x} \in \{\mathbf{z}, \mathbf{z}'\}$

$$\mathbb{P}_{\mathbf{x}}^{\tilde{\pi}}(D_M) = \mathbb{P}_{\mathbf{x}}^{\tilde{\pi}}\left(D'_{\underline{p}}\right) \mathbb{P}_{\mathbf{x}}^{\tilde{\pi}}\left(\widetilde{\mu}_{\underline{i}}^{(p)} \in \chi_i^{(p)} \mid D'_{\underline{p}}\right) \mathbb{P}_{\mathbf{x}}^{\tilde{\pi}}\left(D_{\underline{p},M} \mid D_{\underline{p}}\right)$$

$$= \mathbb{P}_{\mathbf{x}}^{\tilde{\pi}}\left(D'_{\underline{p}}\right) \int_{x \in \chi_{\underline{i}}^{(p)}} f_{\text{Gauss}}(x - \mu_{\mathbf{x}}) \mathbb{P}_{\mathbf{x}}^{\tilde{\pi}}\left(D_{\underline{p},M} \mid D'_{\underline{p}} \cap \{\widetilde{\mu}_{\underline{i}}^{(p)} = x\}\right) \mathrm{d}x \tag{111}$$

where $f_{\text{Gauss}}(\cdot)$ is the probability density function of Gaussian distribution with zero mean and variance $\frac{2\log(1.25/\delta)}{(\varepsilon(n_{\underline{i}}^{(p)} - n_{\underline{i}}^{(p-1)}))^2}$. Note that by definition it holds

$$
\begin{cases}
\mathbb{P}_{\mathbf{z}}^{\tilde{\pi}}\left(D_{\underline{p}}'\right) = \mathbb{P}_{\mathbf{z}'}^{\tilde{\pi}}\left(D_{\underline{p}}'\right) \\
\mathbb{P}_{\mathbf{z}}^{\tilde{\pi}}\left(D_{\underline{p},M} \mid D_{\underline{p}}' \cap \{\widetilde{\mu}_{\underline{i}}^{(p)} = x\}\right) = \mathbb{P}_{\mathbf{z}'}^{\tilde{\pi}}\left(D_{\underline{p},M} \mid D_{\underline{p}}' \cap \{\widetilde{\mu}_{\underline{i}}^{(p)} = x\}\right) \le 1 \quad \forall x \in \chi_i^{(p)}
\end{cases}
\tag{112}
$$

By a property of Gaussian mechanism (Dwork et al., 2014, Appendix A), there exists sets $\gamma^+, \gamma^- \subseteq \chi_i^{(p)}$ with $\gamma^+ \cap \gamma^- = \emptyset$ and $\gamma^+ \cup \gamma^- = \chi_{\underline{i}}^{(p)}$, such that

$$
\begin{cases}
\int_{x \in \gamma^-} f_{\text{Gauss}}(x - \mu_{\mathbf{x}})\, dx \le \frac{\delta}{2} \quad \forall \mathbf{x} \in \{\mathbf{z}, \mathbf{z}'\} \\
\int_{x \in \gamma^+} f_{\text{Gauss}}(x - \mu_{\mathbf{z}})\, dx \le \exp(\varepsilon) \int_{x \in \gamma^+} f_{\text{Gauss}}(x - \mu_{\mathbf{z}'})\, dx
\end{cases}
\tag{113}
$$

Hence, combining (111), (112) and (113), we have

$$
\mathbb{P}_{\mathbf{z}}^{\tilde{\pi}}\left(D_M\right) \le \exp(\varepsilon)\mathbb{P}_{\mathbf{z}'}^{\tilde{\pi}}\left(D_M\right) + \delta
$$

which implies that $\tilde{\pi}$ is $(\varepsilon, \delta)$-differentially private.

## J    PROOF OF PROPOSITION D.3

The proof of Proposition D.3 shares some similarities as the proof of Theorem 4.2, and the difference is using the concentration bound of Gaussian random variables instead of Sub-Exponential random variables.

**Lemma J.1.** *Fix an instance $v \in \mathcal{P}$ and $p \in [M]$. For any set $\mathcal{Q} \subset [K]$ with $|Q| = s_p$ and $i^*(v) \in Q$, let $d_{\mathcal{Q}} = \dim(\text{span}\{\mathbf{a}_i : i \in \mathcal{Q}\})$. We have*

$$
\mathbb{P}_v^{\Pi_{\text{DP-BAI-GAUSS}}}\left(\widetilde{\mu}_j^{(p)} \ge \widetilde{\mu}_{i^*(v)}^{(p)} \mid \mathcal{A}_p = \mathcal{Q}\right)
$$

$$
\le 2\exp\left(-\frac{\Delta_j^2}{16}\left\lceil \frac{T'}{M(d_Q^2 \wedge s_p)}\right\rceil\right) + 2\exp\left(-\frac{\varepsilon^2\Delta_j^2}{32\log(1.25/\delta)}\left\lceil \frac{T'}{M(d_Q^2 \wedge s_p)}\right\rceil^2\right)
\tag{114}
$$

*for all $j \in \mathcal{Q} \setminus \{i^*(v)\}$.*

*Proof.* Fix $j \in \mathcal{Q} \setminus \{i^*(v)\}$. Notice that the sampling strategy of $\Pi_{\text{DP-BAI-GAUSS}}$ depends on the relation between $|\mathcal{Q}|$ and $d_{\mathcal{Q}}$. We consider two cases.

**Case 1**: $d_Q > \sqrt{|\mathcal{Q}|}$.

In this case, note that each arm in $\mathcal{Q}$ is pulled $\left\lceil \frac{T'}{M|\mathcal{Q}|}\right\rceil$ times. Let

$$
E_1 := \{\hat{\mu}_j^{(p)} - \hat{\mu}_{i^*(v)}^{(p)} + \Delta_j \ge \Delta_j/2\}.
\tag{115}
$$

Let $\bar{X}_{i,s} = X_{i,s} - \mu_i$ for all $i \in [K]$ and $s \in [T]$. Notice that $\bar{X}_{i,s} \in [-1,1]$ for all $(i,s)$. Then,

$$
\mathbb{P}_v^{\Pi_{\text{DP-BAI-GAUSS}}}\left(E_1 \mid \mathcal{A}_p = \mathcal{Q}\right)
$$

$$
\le \mathbb{P}_v^{\Pi_{\text{DP-BAI-GAUSS}}}\left(\sum_{s=N_{j,T_p}}^{N_{j,T_p}+\left\lceil \frac{T'}{M|\mathcal{Q}|}\right\rceil} \bar{X}_{j,s} - \sum_{s=N_{i^*(v),T_p}}^{N_{i^*(v),T_p}+\left\lceil \frac{T'}{M|\mathcal{Q}|}\right\rceil} \bar{X}_{i^*(v),s} \ge \frac{\Delta_j}{2}\left\lceil \frac{T'}{M|\mathcal{Q}|}\right\rceil \,\Big|\, \mathcal{A}_p = \mathcal{Q}\right)
$$

$$
\overset{(a)}{\le} \exp\left(-\frac{2\left(\frac{1}{2}\left\lceil \frac{T'}{M|\mathcal{Q}|}\right\rceil \Delta_j\right)^2}{2\left\lceil \frac{T'}{M|\mathcal{Q}|}\right\rceil}\right)
$$

$$
= \exp\left(-\frac{1}{4}\left\lceil \frac{T'}{M|\mathcal{Q}|}\right\rceil \Delta_j^2\right),
\tag{116}
$$

where (a) follows from Lemma A.1. Furthermore, let

$$E_2 := \{\xi_j^{(p)} - \xi_{i^*(v)}^{(p)} \geq \Delta_j/2\}. \tag{117}$$

We note that both $\xi_j^{(p)}$ and $\xi_{i^*(v)}^{(p)}$ under DP-BAI-GAUSS are Gaussian random variables with zero mean and variance $\frac{2\log(1.25/\delta)}{(\varepsilon\lceil\frac{T'}{M|\mathcal{Q}|}\rceil)^2}$. Hence, $\xi_j^{(p)} - \xi_{i^*(v)}^{(p)}$ is a Gaussian random variable with zero mean and variance $\frac{4\log(1.25/\delta)}{(\varepsilon\lceil\frac{T'}{M|\mathcal{Q}|}\rceil)^2}$

Subsequently,

$$\mathbb{P}_v^{\Pi_{\text{DP-BAI-GAUSS}}}\left(E_2 \mid \mathcal{A}_p = \mathcal{Q}\right) \leq \exp\left(-\frac{(\Delta_j\varepsilon\lceil\frac{T'}{M|\mathcal{Q}|}\rceil)^2}{32\log(1.25/\delta)}\right)$$

We therefore have

$$\begin{aligned}
&\mathbb{P}_v^{\Pi_{\text{DP-BAI-GAUSS}}}\left(\widetilde{\mu}_j^{(p)} \geq \widetilde{\mu}_{i^*(v)}^{(p)} \mid \mathcal{A}_p = \mathcal{Q}\right) \\
&\leq \mathbb{P}_v^{\Pi_{\text{DP-BAI-GAUSS}}}\left(E_1 \cup E_2 \mid \mathcal{A}_p = \mathcal{Q}\right) \\
&\overset{(a)}{\leq} \exp\left(-\frac{1}{4}\left\lceil\frac{T'}{M(d_Q^2 \wedge s_p)}\right\rceil \Delta_j^2\right) + \exp\left(-\frac{(\Delta_j\varepsilon\lceil\frac{T'}{M|\mathcal{Q}|}\rceil)^2}{32\log(1.25/\delta)}\right),
\end{aligned} \tag{118}$$

where (a) follows from the union bound.

__Case 2__: $d_Q \leq \sqrt{|\mathcal{Q}|}$.

Similarly, in this case, each arm in $\mathcal{B}_p \subset \mathcal{Q}$ is pulled for $\left\lceil\frac{T'}{Md_Q}\right\rceil$ times. Recall that we have $\mathbf{a}_j^{(p)} = \sum_{i\in\mathcal{B}_p}\alpha_{j,i}\mathbf{a}_i^{(p)}$. Using (9), this implies $\mathbf{a}_j = \sum_{i\in\mathcal{B}_p}\alpha_{j,i}\mathbf{a}_i$, which in turn implies that $\langle\mathbf{a}_j, \theta^*\rangle = \sum_{i\in\mathcal{B}_p}\alpha_{j,i}\langle\mathbf{a}_i, \theta^*\rangle$. Using (12), we then have

$$\mathbb{E}_v^{\Pi_{\text{DP-BAI-GAUSS}}}\left(\widetilde{\mu}_j^{(p)} \mid \mathcal{A}_p = Q\right) = \mu_j. \tag{119}$$

If $j \notin \mathcal{B}_p$, we denote

$$\hat{\mu}_j^{(p)} = \sum_{\iota\in\mathcal{B}_p}\alpha_{j,\iota}\hat{\mu}_\iota^{(p)} \tag{120}$$

and

$$\widetilde{\xi}_j^{(p)} = \sum_{\iota\in\mathcal{B}_p}\alpha_{j,\iota}\widetilde{\xi}_\iota^{(p)}. \tag{121}$$

Note that (120) and (121) still hold in the case of $j \in \mathcal{B}_p$. We define the event

$$G_1 = \left\{\hat{\mu}_j^{(p)} - \mu_j \geq \frac{\Delta_j}{4}\right\}.$$

Then, we have

$$\begin{aligned}
&\mathbb{P}_v^{\Pi_{\text{DP-BAI-GAUSS}}}\left(G_1 \mid \mathcal{A}_p = \mathcal{Q}\right) \\
&= \mathbb{P}_v^{\Pi_{\text{DP-BAI-GAUSS}}}\left(\left\lceil\frac{T'}{Md_Q}\right\rceil(\hat{\mu}_j^{(p)} - \mu_j) \geq \left\lceil\frac{T'}{Md_Q}\right\rceil\frac{\Delta_j}{4}\mid \mathcal{A}_p = \mathcal{Q}\right) \\
&= \mathbb{P}_v^{\Pi_{\text{DP-BAI-GAUSS}}}\left(\sum_{\iota\in\mathcal{B}_p}\sum_{s=N_{\iota,T_{p-1}}+1}^{N_{\iota,T_{p-1}}+\lceil\frac{T'}{Md_Q}\rceil}\alpha_{j,\iota}(X_{\iota,s} - \mu_\iota) \geq \left\lceil\frac{T'}{Md_Q}\right\rceil\frac{\Delta_j}{4}\mid \mathcal{A}_p = \mathcal{Q}\right) \\
&\overset{(a)}{\leq} \exp\left(-\frac{2(\lceil\frac{T'}{Md_Q}\rceil\Delta_j/4)^2}{d_Q\lceil\frac{T'}{Md_Q}\rceil}\right)
\end{aligned}$$

$$\leq \exp\left(-\frac{1}{8}\left\lceil\frac{T'}{Md_Q^2}-1\right\rceil\Delta_j^2\right)$$

$$= \exp\left(-\frac{1}{8}\left\lceil\frac{T'}{M(d_Q^2\wedge s_p)}-1\right\rceil\Delta_j^2\right) \tag{122}$$

where (a) follows Lemma G.1 and Lemma A.1. In addition, we define the event

$$G_2 = \left\{\widetilde{\xi}_j^{(p)}\geq\frac{\Delta_j}{4}\right\}.$$

By the fact that $\widetilde{\xi}_\iota^{(p)}$ is a Gaussian random variable with zero mean and variance $\frac{2\log(1.25/\delta)}{(\varepsilon\lceil\frac{T'}{Md_Q}\rceil)^2}$, we have $\widetilde{\xi}_j^{(p)} = \sum_{\iota\in\mathcal{B}_p}\alpha_{j,\iota}\widetilde{\xi}_\iota^{(p)}$ is a Gaussian random variable with zero mean and variance $\sum_{\iota\in\mathcal{B}_p}\alpha_{j,\iota}^2\frac{2\log(1.25/\delta)}{(\varepsilon\lceil\frac{T'}{Md_Q}\rceil)^2}$. Note that $\sum_{\iota\in\mathcal{B}_p}\alpha_{j,\iota}^2\leq d_Q$, then we can have

$$\mathbb{P}_v^{\Pi_{\text{DP-BAI-GAUSS}}}\left(G_2\mid\mathcal{A}_p=\mathcal{Q}\right)\leq\exp\left(-\frac{(\Delta_j\varepsilon\lceil\frac{T'}{Md_Q}\rceil)^2}{32d_Q\log(1.25/\delta)}\right)$$

$$\leq\exp\left(-\frac{(\Delta_j\varepsilon\lceil\frac{T'}{Md_Q^2}\rceil)^2}{32\log(1.25/\delta)}\right) \tag{123}$$

Define the events

$$G_3 = \left\{\hat{\mu}_{i^*(v)}^{(p)}-\mu_{i^*(v)}\leq-\frac{\Delta_j}{4}\right\} \quad\text{and}\quad G_4 = \left\{\widetilde{\xi}_{i^*(v)}^{(p)}\leq-\frac{\Delta_j}{4}\right\}.$$

Similar to (122) and (123), we have

$$\mathbb{P}_v^{\Pi_{\text{DP-BAI-GAUSS}}}\left(G_3\mid\mathcal{A}_p=\mathcal{Q}\right)\leq\exp\left(-\frac{1}{8}\left\lceil\frac{T'}{M(d_Q^2\wedge s_p)}-1\right\rceil\Delta_j^2\right)$$

and

$$\mathbb{P}_v^{\Pi_{\text{DP-BAI-GAUSS}}}\left(G_4\mid\mathcal{A}_p=\mathcal{Q}\right)\leq\exp\left(\frac{(\Delta_j\varepsilon\lceil\frac{T'}{Md_Q^2}\rceil)^2}{32\log(1.25/\delta)}\right).$$

Hence, in Case 2, we have

$$\mathbb{P}_v^{\Pi_{\text{DP-BAI-GAUSS}}}\left(\widetilde{\mu}_j^{(p)}\geq\widetilde{\mu}_{i^*(v)}^{(p)}\mid\mathcal{A}_p=\mathcal{Q}\right)$$

$$\leq\mathbb{P}_v^{\Pi_{\text{DP-BAI-GAUSS}}}\left(G_1\cup G_2\cup G_3\cup G_4\mid\mathcal{A}_p=\mathcal{Q}\right)$$

$$\leq 2\exp\left(-\frac{1}{8}\left\lceil\frac{T'}{M(d_Q^2\wedge s_p)}-1\right\rceil\Delta_j^2\right)+2\exp\left(\frac{(\Delta_j\varepsilon\lceil\frac{T'}{Md_Q^2}\rceil)^2}{32\log(1.25/\delta)}\right)$$

Finally, combining both Cases 1 and 2, for sufficiently large $T'$, we have

$$\mathbb{P}_v^{\Pi_{\text{DP-BAI-GAUSS}}}\left(\widetilde{\mu}_j^{(p)}\geq\widetilde{\mu}_{i^*(v)}^{(p)}\mid\mathcal{A}_p=\mathcal{Q}\right)$$

$$\leq 2\exp\left(-\frac{1}{8}\left\lceil\frac{T'}{M(d_Q^2\wedge s_p)}-1\right\rceil\Delta_j^2\right)+2\exp\left(-\frac{(\Delta_j\varepsilon\lceil\frac{T'}{M(d_Q^2\wedge s_p)}\rceil)^2}{32\log(1.25/\delta)}\right)$$

$$\leq 2\exp\left(-\frac{1}{16}\left\lceil\frac{T'}{M(d_Q^2\wedge s_p)}\right\rceil\Delta_j^2\right)+2\exp\left(-\frac{(\Delta_j\varepsilon\lceil\frac{T'}{M(d_Q^2\wedge s_p)}\rceil)^2}{32\log(1.25/\delta)}\right).$$

$$\square$$

**Lemma J.2.** *Fix instance $v \in \mathcal{P}$ and $p \in [M_1]$. Recall the definitions of $\lambda$ in (4) and $g_0$ in (8). For any set $\mathcal{Q} \subset [K]$ with $|\mathcal{Q}| = s_p$ and $i^*(v) \in Q$, it holds*

$$\mathbb{P}_v^{\Pi_{\text{DP-BAI-Gauss}}}\left(i^*(v) \notin \mathcal{A}_{p+1} \mid \mathcal{A}_p = \mathcal{Q}\right)$$

$$\leq 2\lambda\left(\exp\left(-\frac{1}{16}\left\lceil\frac{T'}{M(d_{\mathcal{Q}}^2 \wedge s_p)}\right\rceil\Delta_{(g_0)}^2\right) + \exp\left(-\frac{\left(\left\lceil\frac{T'}{M(d_{\mathcal{Q}}^2 \wedge s_p)}\right\rceil\Delta_{(g_0)}\varepsilon\right)^2}{32\log(1.25/\delta)}\right)\right)$$

*Proof.* Fix $v \in \mathcal{P}$, $p \in [M_1]$ and $\mathcal{Q} \subset [K]$ with $|\mathcal{Q}| = s_p$ and $i^*(v) \in Q$. Let $\mathcal{Q}^{\text{sub}}$ be the set of $s_p - g_0 + 1$ arms in $\mathcal{Q}$ with largest suboptimal gap. In addition, let

$$N^{\text{sub}} = \sum_{j \in \mathcal{Q}^{\text{sub}}} \mathbf{1}_{\{\widetilde{\mu}_j^{(p)} \geq \widetilde{\mu}_{i^*(v)}^{(p)}\}}$$

be the number of arms in $\mathcal{Q}^{\text{sub}}$ with private empirical mean larger than the best arm. Then, we have

$$\mathbb{E}_v^{\Pi_{\text{DP-BAI-Gauss}}}\left(N^{\text{sub}} \mid \mathcal{A}_p = \mathcal{Q}\right)$$

$$= \sum_{j \in \mathcal{Q}^{\text{sub}}} \mathbb{E}_v^{\Pi_{\text{DP-BAI-Gauss}}}\left(\mathbf{1}_{\{\widetilde{\mu}_j^{(p)} \geq \widetilde{\mu}_{i^*(v)}^{(p)}\}} \mid \mathcal{A}_p = \mathcal{Q}\right)$$

$$\leq \sum_{j \in \mathcal{Q}^{\text{sub}}} \mathbb{P}_v^{\Pi_{\text{DP-BAI-Gauss}}}\left(\widetilde{\mu}_j^{(p)} \geq \widetilde{\mu}_{i^*(v)}^{(p)} \mid \mathcal{A}_p = \mathcal{Q}\right)$$

$$\overset{(a)}{\leq} \sum_{j \in Q^{\text{sub}}} 2\left(\exp\left(-\frac{1}{16}\left\lceil\frac{T'}{M(d_Q^2 \wedge s_p)}\right\rceil\Delta_j^2\right) + \exp\left(-\frac{(\Delta_j\varepsilon\lceil\frac{T'}{M(d_Q^2 \wedge s_p)}\rceil)^2}{32\log(1.25/\delta)}\right)\right)$$

$$\overset{(b)}{\leq} 2|\mathcal{Q}^{\text{sub}}|\left(\exp\left(-\frac{1}{16}\left\lceil\frac{T'}{M(d_Q^2 \wedge s_p)}\right\rceil\Delta_{(g_0)}^2\right) + \exp\left(-\frac{(\Delta_{(g_0)}\varepsilon\lceil\frac{T'}{M(d_Q^2 \wedge s_p)}\rceil)^2}{32\log(1.25/\delta)}\right)\right)$$

$$= 2(s_p - g_0 + 1)\left(\exp\left(-\frac{1}{16}\left\lceil\frac{T'}{M(d_Q^2 \wedge s_p)}\right\rceil\Delta_{(g_0)}^2\right)\right.$$

$$\left. + \exp\left(-\frac{(\Delta_{(g_0)}\varepsilon\lceil\frac{T'}{M(d_Q^2 \wedge s_p)}\rceil)^2}{32\log(1.25/\delta)}\right)\right), \tag{124}$$

where (a) follows Lemma J.1, and (b) follows from the fact that $\min_{j \in \mathcal{Q}^{\text{sub}}} \Delta_j \geq \Delta_{(g_0)}$.

Then,

$$\mathbb{P}_v^{\Pi_{\text{DP-BAI-Gauss}}}\left(i^*(v) \notin \mathcal{A}_{p+1} \mid \mathcal{A}_p = \mathcal{Q}\right)$$

$$\overset{(a)}{\leq} \mathbb{P}_v^{\Pi_{\text{DP-BAI-Gauss}}}\left(N^{\text{sub}} \geq s_{p+1} - g_0 + 1 \mid \mathcal{A}_p = \mathcal{Q}\right)$$

$$\overset{(b)}{\leq} \frac{\mathbb{E}_v^{\Pi_{\text{DP-BAI-Gauss}}}\left(N^{\text{sub}} \mid \mathcal{A}_p = \mathcal{Q}\right)}{s_{p+1} - g_0 + 1}$$

$$\overset{(c)}{\leq} \frac{2(s_p - g_0 + 1)}{s_{p+1} - g_0 + 1}\left(\exp\left(-\frac{1}{16}\left\lceil\frac{T'}{M(d_Q^2 \wedge s_p)}\right\rceil\Delta_{(g_0)}^2\right) + \exp\left(-\frac{(\Delta_{(g_0)}\varepsilon\lceil\frac{T'}{M(d_Q^2 \wedge s_p)}\rceil)^2}{32\log(1.25/\delta)}\right)\right)$$

$$\overset{(d)}{\leq} \frac{2(h_p + 1)}{h_{p+1} + 1}\left(\exp\left(-\frac{1}{16}\left\lceil\frac{T'}{M(d_Q^2 \wedge s_p)}\right\rceil\Delta_{(g_0)}^2\right) + \exp\left(-\frac{(\Delta_{(g_0)}\varepsilon\lceil\frac{T'}{M(d_Q^2 \wedge s_p)}\rceil)^2}{32\log(1.25/\delta)}\right)\right)$$

$$\overset{(e)}{\leq} 2\lambda\left(\exp\left(-\frac{1}{16}\left\lceil\frac{T'}{M(d_Q^2 \wedge s_p)}\right\rceil\Delta_{(g_0)}^2\right) + \exp\left(-\frac{(\Delta_{(g_0)}\varepsilon\lceil\frac{T'}{M(d_Q^2 \wedge s_p)}\rceil)^2}{32\log(1.25/\delta)}\right)\right)$$

where $(a)$ follows from the fact that $N^{\mathrm{sub}} \geq s_{p+1} - g_0 + 1$ is a necessary condition for $i^*(v) \notin \mathcal{A}_{p+1}$ when $\mathcal{A}_p = \mathcal{Q}$, $(b)$ follows from Markov's inequality, and $(c)$ follows from (124). In addition, $(d)$ is obtained from the definition in (7), and $(e)$ is obtained from the definition in (6).

$\square$

**Lemma J.3.** *Fix instance* $v \in \mathcal{P}$ *and* $p \in \{M_1 + 1, \dots, M\}$. *For any set* $\mathcal{Q} \subset [K]$ *with* $|\mathcal{Q}| = s_p$ *and* $i^*(v) \in Q$, *it holds*

$$
\mathbb{P}_v^{\Pi_{\text{DP-BAI-GAUSS}}} \left( i^*(v) \notin \mathcal{A}_{p+1} \mid \mathcal{A}_p = \mathcal{Q} \right)
$$
$$
\leq 6 \left( \exp\left( -\frac{1}{16} \left\lceil \frac{T'}{M(d_{\mathcal{Q}}^2 \wedge s_p)} \right\rceil \Delta_{(s_{p+2}+1)}^2 \right) + \exp\left( -\frac{(\Delta_{(s_{p+2}+1)}\varepsilon\lceil \frac{T'}{M(d_{\mathcal{Q}}^2 \wedge s_p)}\rceil)^2}{32\log(1.25/\delta)} \right) \right),
\tag{125}
$$

*where we define* $s_{M+2} = 1$.

*Proof.* Fix $v \in \mathcal{P}$, $p \in [M_1]$ and $\mathcal{Q} \subset [K]$ with $|\mathcal{Q}| = s_p$ and $i^*(v) \in Q$. Similarly, let $\mathcal{Q}^{\mathrm{sub}}$ be the set of $s_p - s_{p+2}$ arms in $\mathcal{Q}$ with the largest suboptimality gaps. Again, let

$$
N^{\mathrm{sub}} = \sum_{j \in \mathcal{Q}^{\mathrm{sub}}} \mathbf{1}_{\{\widetilde{\mu}_j^{(p)} \geq \widetilde{\mu}_{i^*(v)}^{(p)}\}}.
$$

Then, we have

$$
\mathbb{E}_v^{\Pi_{\text{DP-BAI-GAUSS}}} \left( N^{\mathrm{sub}} \mid \mathcal{A}_p = \mathcal{Q} \right)
$$
$$
= \sum_{j \in \mathcal{Q}^{\mathrm{sub}}} \mathbb{E}_v^{\Pi_{\text{DP-BAI-GAUSS}}} \left( \mathbf{1}_{\{\widetilde{\mu}_j^{(p)} \geq \widetilde{\mu}_{i^*(v)}^{(p)}\}} \mid \mathcal{A}_p = \mathcal{Q} \right)
$$
$$
\leq \sum_{j \in \mathcal{Q}^{\mathrm{sub}}} \mathbb{P}_v^{\Pi_{\text{DP-BAI-GAUSS}}} \left( \widetilde{\mu}_j^{(p)} \geq \widetilde{\mu}_{i^*(v)}^{(p)} \mid \mathcal{A}_p = \mathcal{Q} \right)
$$
$$
\overset{(a)}{\leq} \sum_{j \in Q^{\mathrm{sub}}} 2 \left( \exp\left( -\frac{1}{16} \left\lceil \frac{T'}{M(d_Q^2 \wedge s_p)} \right\rceil \Delta_j^2 \right) + \exp\left( -\frac{(\Delta_j\varepsilon\lceil \frac{T'}{M(d_Q^2 \wedge s_p)}\rceil)^2}{32\log(1.25/\delta)} \right) \right)
$$
$$
\overset{(b)}{\leq} 2|\mathcal{Q}^{\mathrm{sub}}| \left( \exp\left( -\frac{1}{16} \left\lceil \frac{T'}{M(d_Q^2 \wedge s_p)} \right\rceil \Delta_{(s_{p+2}+1)}^2 \right) + \exp\left( -\frac{(\Delta_{(s_{p+2}+1)}\varepsilon\lceil \frac{T'}{M(d_Q^2 \wedge s_p)}\rceil)^2}{32\log(1.25/\delta)} \right) \right)
$$
$$
= 2(s_p - s_{p+2}) \left( \exp\left( -\frac{1}{16} \left\lceil \frac{T'}{M(d_Q^2 \wedge s_p)} \right\rceil \Delta_{(s_{p+2}+1)}^2 \right) \right.
$$
$$
\left. + \exp\left( -\frac{(\Delta_{(s_{p+2}+1)}\varepsilon\lceil \frac{T'}{M(d_Q^2 \wedge s_p)}\rceil)^2}{32\log(1.25/\delta)} \right) \right),
\tag{126}
$$

where (a) follows from Lemma J.1, and (b) follows from the fact that $\min_{j \in \mathcal{Q}^{\mathrm{sub}}} \Delta_j \geq \Delta_{s_{p+2}+1}$.

By a similar calculation,

$$
\mathbb{P}_v^{\Pi_{\text{DP-BAI-GAUSS}}} \left( i^*(v) \notin \mathcal{A}_{p+1} \mid \mathcal{A}_p = \mathcal{Q} \right)
$$
$$
\overset{(a)}{\leq} \mathbb{P}_v^{\Pi_{\text{DP-BAI-GAUSS}}} \left( N^{\mathrm{sub}} \geq s_{p+1} - s_{p+2} + 1 \mid \mathcal{A}_p = \mathcal{Q} \right)
$$
$$
\overset{(b)}{\leq} \frac{\mathbb{E}_v^{\Pi_{\text{DP-BAI-GAUSS}}} \left( N^{\mathrm{sub}} \mid \mathcal{A}_p = \mathcal{Q} \right)}{s_{p+1} - s_{p+2} + 1}
$$
$$
\overset{(c)}{\leq} \frac{2(s_p - s_{p+2})}{s_{p+1} - s_{p+2} + 1} \left( \exp\left( -\frac{1}{16} \left\lceil \frac{T'}{M(d_Q^2 \wedge s_p)} \right\rceil \Delta_{(s_{p+2}+1)}^2 \right) \right.
$$
$$
\left. + \exp\left( -\frac{(\Delta_{(s_{p+2}+1)}\varepsilon\lceil \frac{T'}{M(d_Q^2 \wedge s_p)}\rceil)^2}{32\log(1.25/\delta)} \right) \right)
$$

$$\overset{(d)}{\leq} 6 \left( \exp \left( -\frac{1}{16} \left\lceil \frac{T'}{M(d_Q^2 \wedge s_p)} \right\rceil \Delta_{(s_{p+2}+1)}^2 \right) + \exp \left( -\frac{(\Delta_{(s_{p+2}+1)} \varepsilon \lceil \frac{T'}{M(d_Q^2 \wedge s_p)} \rceil)^2}{32 \log(1.25/\delta)} \right) \right),$$

where (a) follows from the fact that $N^{\text{sub}} \geq s_{p+1} - s_{p+2} + 1$ is the necessary condition of $i^*(v) \notin \mathcal{A}_{p+1}$ when $\mathcal{A}_p = \mathcal{Q}$, (b) follows from Markov's inequality, (c) follows from (126), and (d) is obtained from the definition in (7). $\qquad\square$

With the above ingredients in place, we are now ready to prove Proposition D.3.

*Proof of Proposition D.3.* Fix instance $v \in \mathcal{P}$. Recall that in DP-BAI-GAUSS, the decision maker eliminates arms in successive phases, and the decision maker can successfully identify the best arm if and only if it is not eliminated in any of the phases. That is,

$$\mathbb{P}_v^{\Pi_{\text{DP-BAI-GAUSS}}} \left( \hat{I}_T \neq i^*(v) \right) \leq \sum_{p=1}^{M} \mathbb{P}_v^{\Pi_{\text{DP-BAI-GAUSS}}} \left( i^*(v) \notin \mathcal{A}_{p+1} \,\big|\, i^*(v) \in \mathcal{A}_p \right).$$

Then, we split the rightmost sum of the probabilities into two parts. Let

$$P_1 = \sum_{p=1}^{M_1} \mathbb{P}_v^{\Pi_{\text{DP-BAI-GAUSS}}} \left( i^*(v) \notin \mathcal{A}_{p+1} \,\big|\, i^*(v) \in \mathcal{A}_p \right)$$

and

$$P_2 = \sum_{p=M_1+1}^{M} \mathbb{P}_v^{\Pi_{\text{DP-BAI-GAUSS}}} \left( i^*(v) \notin \mathcal{A}_{p+1} \,\big|\, i^*(v) \in \mathcal{A}_p \right).$$

When $K \leq \lceil d^2/4 \rceil$, by definition, we have $M_1 = 0$, which implies that $P_1 = 0$. When $K > \lceil d^2/4 \rceil$ by Lemma J.2, we obtain

$$P_1 \leq 2M_1 \lambda \left( \exp \left( -\frac{1}{16} \left\lceil \frac{T'}{Md^2} \right\rceil \Delta_{(\lceil d^2/4 \rceil)}^2 \right) + \exp \left( -\frac{(\Delta_{(\lceil d^2/4 \rceil)} \varepsilon \lceil \frac{T'}{Md^2} \rceil)^2}{32 \log(1.25/\delta)} \right) \right)$$

$$\leq 2M_1 \lambda \left( \exp \left( -\frac{T'}{64MH_{\text{BAI}}} \right) + \exp \left( -\left( \frac{T'}{64MH_{\text{pri}} \sqrt{\log(1.25/\delta)}} \right)^2 \right) \right). \quad (127)$$

In addition, by Lemma J.3

$$P_2 \leq \sum_{p=M_1+1}^{M} 6 \left( \exp \left( -\frac{1}{16} \left\lceil \frac{T'}{Ms_p} \right\rceil \Delta_{(s_{p+2}+1)}^2 \right) + \exp \left( -\frac{(\Delta_{(s_{p+2}+1)} \varepsilon \lceil \frac{T'}{Ms_p} \rceil)^2}{32 \log(1.25/\delta)} \right) \right)$$

$$\leq \sum_{p=M_1+1}^{M} 6 \left( \exp \left( -\frac{T'}{16M(4s_{p+2}+4)} \Delta_{(s_{p+2}+1)}^2 \right) + \exp \left( -\frac{(\Delta_{(s_{p+2}+1)} \varepsilon \frac{T'}{M(4s_{p+2}+4)})^2}{32 \log(1.25/\delta)} \right) \right)$$

$$\leq \sum_{i \in \{s_{p+2}+1: p \in \{M_1+1,\dots,M\}\}} 6 \left( \exp \left( -\frac{T'}{64Mi} \Delta_{(i)}^2 \right) + \exp \left( -\left( \frac{T' \varepsilon \Delta_{(i)}}{64Mi \sqrt{\log(1.25/\delta)}} \right)^2 \right) \right)$$

$$\leq 6(M - M_1) \left( \exp \left( -\frac{T'}{64MH_{\text{BAI}}} \right) + \exp \left( -\left( \frac{T'}{64MH_{\text{pri}} \sqrt{\log(1.25/\delta)}} \right)^2 \right) \right). \quad (128)$$

Finally, combining (127) and (128), for sufficiently large $T'$ we have

$$\mathbb{P}_v^{\Pi_{\text{DP-BAI-GAUSS}}} \left( \hat{I}_T \neq i^*(v) \right)$$

$$\leq P_1 + P_2$$

$$\leq \exp \left( -\Omega \left( \frac{T}{H_{\text{BAI}} \log d} \wedge \left( \frac{T}{\sqrt{\log(\frac{1.25}{\delta})} H_{\text{pri}} \log d} \right)^2 \right) \right).$$

This completes the proof. $\qquad\square$

