# OpenReview forum: "Fixed-Budget Differentially Private Best Arm Identification"
_ICLR.cc/2024/Conference — ICLR 2024 poster_

### Official Review · Reviewer_fyMv · 2023-10-28

**Soundness:** 3 good
**Presentation:** 3 good
**Contribution:** 3 good
**Rating:** 8
**Confidence:** 3

**Summary:**

This paper studies the problem of identifying the best arm in a differentially private manner. This paper focuses on the pure DP setting. Here, the privacy model is as follows. At each time step T, there are K arms. Each arm has a potential reward x_{t, k}. The algorithm is DP if, changing only one of the x_{t, k} causes the trajectory of arms that are selected to satisfy the usual eps-DP property.

The authors prove tight upper and lower bounds for this problem. In particular, they determine a parameter, similar to the parameter in the non-private setting, which essentially governs the error rate of the algorithm.

**Strengths:**

This paper studies an interesting problem and will be of interest to researchers working on DP and bandits. The authors also prove tight results so I believe this is a significant contribution to the literature. The writing quality and clarity is good. The lower bound is a nice technical contribution.

**Weaknesses:**

Unless I misunderstood, two datasets are neighboring if the set of possible rewards differ in only one location. I'm curious what the motivation for this is. For example, if we use clinical trials as the motivating example then each arm may correspond to a different treatment. In this case, I would view two datasets as neighboring if the set of observations differ at one step which could mean that all K arms at a single time step have different rewards. I am curious how this impacts the results of the paper.

**Questions:**

See weaknesses above.

---

> ### Author Response · Authors · 2023-11-16
>
> We are grateful for your positive feedback on our paper and that you find value in our work. We present our response to your comments below.
>
> ---
>
>  **Q1**: "Unless I misunderstood, two datasets are neighboring if the set of possible rewards differ in only one location. I'm curious what the motivation for this is. For example, if we use clinical trials as the motivating example then each arm may correspond to a different treatment. In this case, I would view two datasets as neighboring if the set of observations differ at one step which could mean that all $K$ arms at a single time step have different rewards. I am curious how this impacts the results of the paper."
>
> **A1**: We thank the reviewer for this insightful question. In the revised version of our paper, we have created a new Appendix C to reply to this question. Briefly, our definition of DP is equivalent to that of **table-DP** appearing in [1] (by making a correspondence between the notations), and the latter precisely satisfies the reviewer's expectation of ``I would view two datasets as neighboring if the set of observations differ at one step which could mean that all $K$ arms at a single time step have different rewards.'' See Definition C.2 as well as the subsequent analyses for the details.
>
> **Reference**
>
> [1] Azize, A. and Basu, D. (2023). Interactive and concentrated differential privacy for bandits. In Sixteenth European Workshop on Reinforcement Learning.

---

> > ### Comment · Reviewer_fyMv · 2023-11-21
> >
> > Thanks for the response! I have no further questions.

---

### Official Review · Reviewer_mZMs · 2023-10-31

**Soundness:** 3 good
**Presentation:** 3 good
**Contribution:** 2 fair
**Rating:** 5
**Confidence:** 4

**Summary:**

This work studied the way to adopt differential privacy on the Best Arm Identification policy. The main trick is to append each arm's empirical mean with proper Laplace distribution.

**Strengths:**

The work provide comprehensive details on establishing its theoretical claims.

**Weaknesses:**

The empirical evaluation is limited on a particular synthetic data instance.

**Questions:**

1. Motivation to consider DP-BAI algorithm. For my understanding, DP is for preventing potential privacy risk when sharing statistics of a dataset. I would be great for the author to provide motivation to consider DP in bandit problem, especially on the best arm identification task.

2. Intuition on scheme (7). I had difficulty to make sense on  the scheme (7) and would be thankful is author can provide explanations.

---

> ### Author Response · Authors · 2023-11-16
> **Response to Reviewer mZMs (part 1/2)**
>
> We thank the reviewer for engaging with our manuscript and providing useful feedback. We summarise your comments/questions and respond to them below.
>
>
> ---
>
>
>  **Q1**: The empirical evaluation is limited on a particular synthetic data instance.
>
> **A1**: We acknowledge the limitation pointed out by the reviewer, and this is indeed an aspect that we are mindful of. The focus of our paper is primarily theoretical, aiming to establish and explore conceptual frameworks within differentially private BAI. Although the key intent of our work is to provide a strong theoretical foundation for future research and discussion, keeping in mind the importance of empirical validation, we benchmark the performance of our algorithm against existing non-private algorithms or the private adaptations thereof. We sincerely hope that the reviewer perceives this stance of ours optimistically.
>
> ---
>
>  **Q2**: ``It would be great for the authors to provide motivation to consider DP in bandit problem, especially on the best arm identification task.''
>
> **A2**: Our conceptualization of differential privacy within the context of bandit problems aligns closely with established literature [1,2,3,4,5,6], particularly drawing inspiration from the work of Nikolakakis et al. [3]. While the primary objective of our work is fixed-budget BAI, it is noteworthy that the objectives in other relevant studies center around either regret minimization or fixed-confidence BAI. Importantly, these alternative settings necessitate unique analytical methodologies, yielding results distinct in nature from our own.
>
> The underlying principle of privacy in the aforementioned body of literature, including our work, may be articulated as follows: an alteration in one of the rewards received by the learning agent should not largely influence subsequent decisions regarding the pulling of arms. To illustrate this point further, certain works posit a scenario where each time step is associated with an independent "user"; pulling arm $i$ at time $t$ provides the agent with user $t$'s "score" of arm $i$. If the score from a preceding user significantly impacts subsequent arm selections by the agent, there exists a potential inference risk for later users regarding the (private) scores of their predecessors through observations of the arm selections. Such privacy leakage can be mitigated by the application of differential privacy mechanisms tailored for bandit scenarios. In the context of the BAI task, the "agent" can be perceived as a surveyor seeking to identify the optimal item from a set of $K$ items in a market survey, while the "users" can be likened to customers participating in the survey.

---

> > ### Author Response · Authors · 2023-11-16
> > **Response to Reviewer mZMs (part 2/2)**
> >
> > **Q3**: "I had difficulty to make sense on the scheme (7) and would be thankful if author can provide explanations."
> >
> > **A3**: We thank the reviewer for the comment. Scheme (7) defines the variable $s_p$, which is the pre-configured number of active arms in phase $p$.
> > Following are some details of (7) assuming $K \gg d^2$.
> > Note that for $p=1$, we have $s_1=h_0+g_0=K$, where $g_0=\Theta(d^2)$, i.e., there $K$ active arms in phase 1 (the default initialisation).
> > Then, for phase $p \in \{2,\ldots,M_1\}$, we have from (7) that
> > $$
> > s_{M_1}=g_0, \quad
> > \forall p < M_1, ~ \frac{ s_{p+1}-g_0 }{ s_{p}-g_0 }=\frac{h_{p+1}}{h_{p}} \approx \frac{1}{\lambda} ,
> > $$
> > which means that if $s_p \gg g_0=\Theta(d^2)$ for some $p$, then the agent eliminates approximately $1-\frac{1}{\lambda}$ fraction of the current active arms at end of phase $p$. Therefore, from phase $1$ until phase $M_1$, the number of active arms reduces from $K$ to $\Theta(d^2)$, and the shrinkage rate is approximately $1-\frac{1}{\lambda}$ in a majority of these $M_1$ phases. Here, $\lambda$ is chosen to ensure that $M_1 = \Theta(\log d)$.
> > Lastly, for phase $p\in \{M_1+1,\ldots,M\}$, we have from (7) that $\frac{s_{p+1}}{s_p}\approx \frac{1}{2}$. That is, the agent eliminates approximately half of the active arms in each phase, and $M$ is chosen in such a way that $s_{M+1}=1$, i.e., only one active arm remains after $M$ phases. For clarity, we provide a numerical illustration of the above specifications in the newly created Appendix B of the revised manuscript.
> >
> >
> > **References**
> >
> > [1] Mishra, N. and  Thakurta, A. (2015, July). (Nearly) optimal differentially private stochastic multi-arm bandits. In Proceedings of the Thirty-First Conference on Uncertainty in Artificial Intelligence (pp. 592-601).
> >
> > [2] Sajed, T. and  Sheffet, O. (2019, May). An optimal private stochastic-mab algorithm based on optimal private stopping rule. In International Conference on Machine Learning (pp. 5579-5588). PMLR.
> >
> > [3] Nikolakakis, K. E., Kalogerias, D. S., Sheffet, O.,   Sarwate, A. D. (2021). Quantile multi-armed bandits: Optimal best-arm identification and a differentially private scheme. IEEE Journal on Selected Areas in Information Theory, 2(2), 534-548.
> >
> > [4] Azize, A. and Basu, D. (2022). When privacy meets partial information: A refined analysis of differentially private bandits. Advances in Neural Information Processing Systems, 35, 32199-32210.
> >
> > [5] Solanki, S., Kanaparthy, S., Damle, S. and  Gujar, S. (2022, September). Differentially Private Federated Combinatorial Bandits with Constraints. In Joint European Conference on Machine Learning and Knowledge Discovery in Databases (pp. 620-637). Cham: Springer Nature Switzerland.
> >
> > [6] Azize, A. and Basu, D. (2023). Interactive and concentrated differential privacy for bandits. In Sixteenth European Workshop on Reinforcement Learning.

---

> ### Author Response · Authors · 2023-11-19
>
> Dear Reviewer mZMs,
>
> We hope this message finds you well. The rebuttal period is scheduled to conclude on November 22nd, which is **ending**. We are committed to engaging all reviewers with discussions. We hope that our responses are satisfactory; if you need further clarifications, please let us know. Thank you for your time and attention.
>
> Best regards,
>
> Authors of 3619

---

### Official Review · Reviewer_Nv4R · 2023-10-31

**Soundness:** 4 excellent
**Presentation:** 4 excellent
**Contribution:** 3 good
**Rating:** 8
**Confidence:** 3

**Summary:**

The authors propose and analyze an algorithm for the fixed-budget best-arm identification problem with differential privacy constraints. The algorithm is based on differentially private version of MAX-DET rather than adapting established fixed-budget BAI algorithms. In the appendix the authors provide a general analysis of natural possible extension of one such algorithm and clearly benchmark their algorithm with this extension. The lower bounds also depend on new notions of complexity which incorporate differential privacy as a constraint in policy design.

**Strengths:**

I think the authors provide fundamental analysis of the problem in terms of the upper and lower bound. The main strength of the contribution is demonstrated by proving that the algorithm matches instance optimal bounds. Further, the algorithm idea is new itself and clearly outperforms existing benchmarks (and straightforward adaptations thereof).

**Weaknesses:**

I think the paper can be written more intuitively given that it has a lot of parameters. For example, while the algorithm is stated clearly, I am unsure why it works intuitively. What makes the apparent dimension go down for the first few rounds? How does decreasing the span basis vectors of the arm space lead to convergence to the optimal arm.

**Questions:**

--
I am skeptical about their definition of DP since it is defined over length $T$ sequences. I would expect that in the online setting this definition would only be defined over a sample of sequence starting from the time when the reward sequences differ as done in joint differential privacy.

--
I would like an intuitive explanation of why their algorithm works on a small example somewhere in the paper (maybe, in an appendix).

--
Finally, the central idea of the paper is to use D-optimal design rather than G-optimal design. These ideas are theoretically equivalent (see Proposition~3 in Soare et al. NeurIPs 2014) but  this paper demonstrates a dramatic performance improvement in their numerical experiments. What is the intuitive explanation for this?

---

> ### Author Response · Authors · 2023-11-16
>
> We thank the reviewer for recognizing the value of our work and for your insightful comments. Below are our responses to your questions.
>
> ---
>
> **Q1**: ``I am skeptical about their definition of DP since it is defined over length $T$
> sequences. I would expect that in the online setting this definition would only be defined over a sample of sequence starting from the time when the reward sequences differ as done in joint differential privacy.''
>
> **A1**: We thank the reviewer for this question. Our definition of DP indeed aligns with your expectation of ``over a sample of sequence starting from the time when the reward sequences differ''. Intuitively, this is because the future rewards have no impact on the past sequence of arm pulls. Below are further details.
>
> Our definition of DP does not explicitly entail time steps; note that $X_{i,j}$ in Section 2 represents the $j$-th reward from arm $i$ (rather than the classical representation of reward from arm $i$ at time step $j$). In spite of the minor difference in the representation of rewards, our definition of DP is, in fact, equivalent to the notion of DP arising from the classical representation of rewards (which we call **$\varepsilon$-table-DP** borrowing the terminology from Azize et al. [1]). We prove this formally in the newly created Appendix C of the revised manuscript by introducing a **dual decision maker** who observes reward $X_{A_t,t}$ at time $t$ instead of $X_{A_t, N_{A_t, t}}$ as in our work. We then show that for any $\varepsilon$-DP policy $\pi$ of the decision maker as defined in our work, its dual decision maker's policy (or simply the dual policy) $D(\pi)$ meets the $\varepsilon$-table-DP criterion, and vice-versa (see Propositions C.4 and C.5 in the revised paper).
>
> We note here that table-DP is based on ``a sample of sequence starting from the time when the reward sequences differ'' highlighted by the reviewer; see Corollary C.3 of the newly created Appendix C for further details.
>
> ---
>  **Q2**: ``I would like an intuitive explanation of why their algorithm works on a small example somewhere in the paper (maybe, in an appendix).What makes the apparent dimension go down for the first few rounds? How does decreasing the span basis vectors of the arm space lead to convergence to the optimal arm.''
>
> **A2**: We thank the reviewer for the suggestion. We have now provided additional explanations in the newly created Appendix B of our revised paper (highlighted in blue).  Firstly, we would like to clarify that in the first few rounds (precisely $M_1$ rounds), the dimension may not go down as the number of arms is at least $\Theta(d^2)$. Secondly, as pointed out by the reviewer, "decreasing the span" is indeed critical in identifying the best arm. This is because "decreasing the span" increases the probability that the private empirical mean of the best arm surpasses that of others, as is shown in Lemma E.2 of the original manuscript ( or Lemma G.2 of the revised version).
>
> We remain open to elucidating any particular facet of our algorithm should the reviewer express a desire for further explanations.
>
> ---
>  **Q3**: Finally, the central idea of the paper is to use D-optimal design rather than G-optimal design. These ideas are theoretically equivalent (see Proposition~3 in Soare et al. NeurIPS 2014) but this paper demonstrates a dramatic performance improvement in their numerical experiments. What is the intuitive explanation for this?
>
> **A3**: As pointed out by the reviewer, D-optimal design and G-optimal design are indeed theoretically equivalent. However, our Max-Det principle is rather different from D-optimal design. Note that $\gamma^*: \mathcal{A} \rightarrow [0,1]$ is a D-optimal design of vector set $\mathcal{A}$ if
> $$
> \gamma^* = \arg\max_{\gamma} {\rm det}(V(\gamma)),
> \quad\mbox{where} \quad
> V(\gamma) := \sum_{a\in \mathcal{A}} \gamma(a) \\, aa^\top.
> $$
> There are two primary differences between D-optimal design and the Max-Det collection (defined in Definition 3.1). Firstly, the matrices of which the determinants are calculated are different. In D-optimal design, this matrix is formed by the weighted sum of some rank-one matrices of the form $aa^\top$ for $a \in \mathcal{A}$, whereas in Definition 3.1, this matrix is formed by stacking up a subset of vectors from $\mathcal{A}$ as its columns. Secondly, in D-optimal design, the weight of each vector (i.e., $\gamma^*(a)$ for $a\in \mathcal{A}$) may be any value in  $[0,1]$. In contrast, in Definition 3.1, this ``weight'' of each vector in $\mathcal{A}$ can be regarded as being $0$ or $1$, depending on whether or not it belongs to the Max-Det collection.
>
> We appreciate the reviewer's question for providing us with the opportunity to clarify the differences between D-optimal design and our Max-Det principle.
>
> **Reference**
>
>
> [1] Azize, A. and Basu, D. (2023). Interactive and concentrated differential privacy for bandits. In Sixteenth European Workshop on Reinforcement Learning.

---

### Official Review · Reviewer_pqgE · 2023-11-10

**Soundness:** 3 good
**Presentation:** 2 fair
**Contribution:** 3 good
**Rating:** 6
**Confidence:** 2

**Summary:**

This paper investigates the best arm identification (BAI) problem in multi-armed bandits with differential privacy (DP) guarantees. It focuses on the fixed-budget BAI setting, aiming to minimize the error probability of selecting a suboptimal arm within a set number of arm pulls (budget). Notably, the paper introduces DP-BAI, the first algorithm to achieve DP guarantees in the fixed-budget BAI context. This algorithm outperforms a naive approach that directly incorporates a DP mechanism into the existing state-of-the-art non-private BAI algorithm.

**Strengths:**

This paper introduces the first algorithm to solve the BAI problem within a fixed-budget constraint and under DP guarantees. The paper also offers an extensive theoretical analysis, establishing an upper bound on the error probability for the new algorithm, which is adaptive to the complexity of the problem measured by $H_{BAI}, H_{pri}$. Furthermore, it provides a matching lower bound, demonstrating that the algorithm attains optimal performance in this specific setting.

**Weaknesses:**

The paper lacks a detailed comparison with existing non-private fixed-budget BAI works. For example, a natural question is: Does the error probability of DP-BAI converge to that of the state-of-the-art non-private counterpart when $\epsilon \to \infty$? Such an analysis would be valuable in understanding the trade-offs between privacy and performance.

Moreover, I personally believe that the writing of this paper, especially in the algorithm description section, could be improved. Currently the presentation has a lot of notations, many without adequate explanations. A more detailed explanation of each notation and some high-level insights would significantly improve the paper's readability and effectively convey the core ideas.

**Questions:**

- see Weakness 1: How does DP-BAI compare to OD-LinBAI, especially when $\epsilon\to\infty$?

- Page 4, Definition 3.1: Why is a Max-Det collection of $\mathcal{A}$ always linearly independent raises questions. Does this implicitly assume that $\mathcal{A}$ spans $\mathbb{R}^{d'}$?

- Regarding Algorithm 1:
  - Lines 8, 14: The phrasing in these lines is confusing and seems inconsistent with earlier descriptions. The term "pull each arm XX times" is ambiguous. From the description, it appears that, in line 8, every arm in $B_p$ is pulled $T'/Md_p$ times, totaling $T'/M$ arm pulls. Meanwhile, in line 14, it seems there are a total of $T'/Ms_p$ arm pulls, with each pull randomly choosing an arm from $A_p$.
  - Can the authors provide some high-level intuition behind the choice of $g_i$ and $h_i$ as described in equations (5) and (6)?

---

> ### Author Response · Authors · 2023-11-16
>
> We extend our sincere gratitude for your careful review of our paper. We present our response to your comments below.
>
> ---
> **Q1**: ``How does DP-BAI compare to OD-LinBAI, especially when $\varepsilon \rightarrow \infty$.''
>
> **A1**:  When $\varepsilon \rightarrow \infty$, the upper bound on the error probability of our DP-BAI algorithm is given by
> $$
> \\mathbb{P} \\left(\\hat{I}\_T \neq i^* \\right) \le \\exp \\left(-\Omega \\left(\frac{T}{\\log(d) H\_{\\rm BAI} } \\right) \\right).
> $$
> In contrast, the upper bound of OD-LinBAI is given by
> $$
> \\mathbb{P} \\left(\\hat{I}\_T \neq i^* \\right) \le \exp \\left(-\Omega \\left (\frac{T}{\log(d) H\_{2,{\\rm lin}}} \\right) \\right).
> $$
>
> Note that $H\_{\\rm BAI}$ is at least as large as  $H\_{2,{\\rm lin}}$ because  $H\_{\\rm BAI}=\\max\_{2\le i \le (d^2 \land K) }\frac{i}{\Delta_i^2} \ge H_{2,{\rm lin}}=\max_{2\le i \le (d \land K) }\frac{i}{\Delta_i^2}$. Hence, OD-LinBAI outperforms DP-BAI in the non-private setting (i.e., when $\varepsilon \rightarrow \infty$). However, DP-BAI outperforms DP-OD (a private adaptation of OD-LinBAI) as demonstrated in Appendix C of our original manuscript (or Appendix E of the revised version). These results are not surprising as OD-LinBAI is specifically designed for non-private settings, whereas DP-BAI is specifically designed for private settings. However, we would like to emphasize that $H_{\\rm BAI}$ shares the same form as $H_{2, {\\rm lin}}$ in that, for small $d$, it only depends on the first $d^2$ gaps ($d$ for OD-LinBAI).
>
> ---
>
>  **Q2**: ``Does this implicitly assume that $\mathcal{A}$ spans $R^{d'}$?''
>
> **Q3**: ``The phrasing in these lines is confusing and seems inconsistent with earlier descriptions.''
>
> **Response to Q2 and Q3**: The reviewer is verily right in these two questions. We apologize for the typos, and have revised the above highlighted sentences in Section 3 (please see the parts in blue) accordingly. Briefly: (1) Yes, we only consider the case in which  $\mathcal{A}$ spans $\mathbb{R}^{d'}$. (2) Yes, your understanding of Line 8 is correct, and "randomly" was a typo appearing in the earlier description of Line 14; we have now rectified this in the revised version. We express our sincere gratitude to the reviewer for their careful reading of our manuscript.
>
> ---
>
> **Q4**: ``Can the authors provide some high-level intuition behind the choice of $g_i$ and $h_i$ as described in equations (5) and (6)?''
>
> **A4**: The rationale behind the choices of $g_i$ and $h_i$ is rooted in their role as auxiliary parameters governing the number of active arms in each phase, as outlined in (7). Adhering to the principle of successive elimination (SE) [1,2], our algorithm aims to eliminate a specific proportion of active arms in each of the $\Theta(\log d)$ phases. The determination of this proportion in each phase is crucial for achieving the overall performance guarantee. In essence, when the number of active arms in any given phase significantly exceeds $\Theta(d^2)$, the agent eliminates approximately $1-\frac{1}{\lambda}$ proportion of active arms; otherwise, it eliminates roughly half of the active arms. The threshold $\Theta(d^2)$ is pivotal in the Max-Det principle analysis. The parameter $\lambda$ in (4) is chosen to ensure that $\Theta(\log d)$ phases transpire before the number of active arms reduces to $\Theta(d^2)$ in cases where $K \gg d^2$. This leads to $g_i \approx g_{i-1}/2$ and $h_i \approx h_{i-1}/\lambda$ in equations (5) and (6), respectively. These specifications are numerically illustrated in the newly created Appendix B for clarity.
>
>
> **References**
>
> [1] Karnin, Z., Koren, T., and Somekh, O. (2013, May). Almost optimal exploration in multi-armed bandits. In International Conference on Machine Learning, pp. 1238-1246. PMLR.
>
> [2] Yang, J., and Tan, V. (2022). Minimax Optimal Fixed-Budget Best Arm Identification in Linear Bandits. Advances in Neural Information Processing Systems, pp. 12253-12266.

---

> > ### Comment · Reviewer_pqgE · 2023-11-16
> > **Response to Rebuttal**
> >
> > Thank you for the detailed and adequate response.

---

### Official Review · Reviewer_9gBt · 2023-11-11

**Soundness:** 3 good
**Presentation:** 3 good
**Contribution:** 3 good
**Rating:** 8
**Confidence:** 4

**Summary:**

This paper is the first to study best arm identification (BAI) in linear bandits under a fixed DP budget budget. With extensive research focusing on DP MAB, this paper tries to answer this problem through a different lens of "pure exploration". This work serves as a valuable complement to the current body of literature.

**Strengths:**

To this end, the paper proposes a policy satisfying $\epsilon$-DP, thus providing an upper bound of the decaying speed of the error probability. The paper also provides an almost-matching lower bound. Empirical evaluation is also provided to show the effectiveness of the algorithm.

**Weaknesses:**

Although this is a nice work, I still suggest the paper provide more discussion on the connections between this problem to 1) BAI in the fixed-confidence regime, 2) and generally, MAB under DP. I understand superficially speaking they are different problems, but it is not very clear (to me) whether or not there exist some connections  deeper. For example, there might be a simple adaptation of previous algorithms to suit this setting.

**Questions:**

Please refer to weakness

---

> ### Author Response · Authors · 2023-11-16
> **Response to Reviewer 9gBt (part 1/2)**
>
> We are grateful for your valuable feedback. We are truly honored and encouraged by the positive evaluation and constructive feedback you provided. Below is our response.
>
> ---
>
> **Q**: ``I still suggest the paper provide more discussion on the connections between this problem to 1) BAI in the fixed-confidence regime, 2) and generally, MAB under DP. I understand superficially speaking they are different problems, but it is not very clear (to me) whether or not there exist some connections deeper. For example, there might be a simple adaptation of previous algorithms to suit this setting.''
>
> **A**: Thank you for the suggestion. Below are some comparisons between the different bandit settings.
>
> (1) **Fixed-budget BAI and regret minimization**: While regret minimization inherently entails an exploration vs. exploitation dilemma, BAI (encompassing fixed-budget and fixed-confidence regimes) on the other hand is centered around pure exploration. As astutely demonstrated in the work of Zhong et al. [1], no algorithm can perform optimally for both of the above objectives simultaneously. Thus, for instance, an algorithm achieving the minimum probability of error in identifying the best arm with a fixed budget is inevitably accompanied by a larger regret value, and vice versa; for more details, see [1, Theorem 5.1]. Consequently, in the nuanced context of our work, which involves additional considerations such as differential privacy, a mere transposition of ideas from regret minimization proves insufficient to devise algorithms with near-optimal performance for the rather distinct objective of BAI.
>
>
> (2) **Fixed-budget BAI and fixed-confidence BAI**: In the domain of pure exploration problems, two pivotal considerations, namely (a) the probability of error and (b) the stopping time required to ascertain the best arm, compete with one another. In the fixed-budget regime, a deterministic time (a.k.a. budget) $T$ is fixed, and the goal is to minimise the probability of error in determining the best arm after $T$ time steps. On the other hand, in the fixed-confidence regime, given a threshold $\delta \in (0,1)$ (a.k.a. confidence level), the goal is to minimise the expected stopping time subject to the error probability not exceeding $\delta$.
> A notable challenge in fixed-confidence scenarios lies in the determination of the expected stopping time, which is typically unknown to the agent. This is because the upper bound of the expected stopping time is, in general, a function of an unknown problem-specific ``hardness'' parameter, as evidenced in various works such as [2,3,4,5]. Consequently, establishing a straightforward relationship between $\delta$ in the fixed-confidence regime and the budget $T$ in the fixed-budget regime is substantially intricate.
>
> Furthermore, it is noteworthy that while the asymptotic complexity of fixed-confidence BAI has been comprehensively characterised in the asymptotic limit as $\delta \downarrow 0$ by Garivier and Kaufmann [2], a corresponding characterisation for fixed-budget BAI (where the asymptotics is as $T \to \infty$) remains an open problem, as articulated by Qin [6].
> Hence, the transposition of ideas from fixed-confidence BAI into the realm of fixed-budget BAI appears nontrivial due to the above fundamental disparities.

---

> > ### Author Response · Authors · 2023-11-16
> > **Response to Reviewer 9gBt (part 2/2)**
> >
> > Finally, we would like to emphasize that fixed-budget BAI remains a considerable challenge in multi-armed bandits, and continues to be actively investigated; see, for instance, the recent works [6,7,8,9,10]. To the best of our knowledge, our work is the first attempt towards incorporating DP considerations within the framework of fixed-budget BAI. The amalgamation of these two aspects, namely fixed-budget BAI and DP, introduces numerous analytical challenges. Despite these challenges, we provide a complete characterization of the exponent of the probability of error up to universal constants.
> >
> > [1] Zhong, Z., Cheung, W. C., and Tan, V. (2023). Achieving the Pareto frontier of regret minimization and best arm identification in multi-armed bandits. Transactions on Machine Learning Research.
> >
> > [2] Garivier, A., and Kaufmann, E. (2016, June). Optimal best arm identification with fixed confidence. In Conference on Learning Theory (pp. 998-1027). PMLR.
> >
> > [3] Wang, P. A., Tzeng, R. C., and Proutiere, A. (2021). Fast pure exploration via Frank--Wolfe. Advances in Neural Information Processing Systems, 34, 5810-5821.
> >
> > [4] Nikolakakis, K. E., Kalogerias, D. S., Sheffet, O., Sarwate, A. D. (2021). Quantile multi-armed bandits: Optimal best-arm identification and a differentially private scheme. IEEE Journal on Selected Areas in Information Theory, 2(2), 534-548.
> >
> > [5] Hou, Y., Tan, V. Y., and Zhong, Z. (2022). Almost optimal variance-constrained best arm identification. IEEE Transactions on Information Theory, 69(4), 2603-2634.
> >
> > [6] Qin, C. (2022, September). Open Problem: Optimal Best Arm Identification with Fixed-Budget. In Conference on Learning Theory (pp. 5650-5654). PMLR.
> >
> > [7] Yang, J., and Tan, V. (2022). Minimax Optimal Fixed-Budget Best Arm Identification in Linear Bandits. Advances in Neural Information Processing Systems, 35, 12253-12266.
> >
> > [8] Komiyama, J., Tsuchiya, T., and Honda, J. (2022). Minimax optimal algorithms for fixed-budget best arm identification. Advances in Neural Information Processing Systems, 35, 10393-10404.
> >
> > [9] Barrier, A., Garivier, A., and Stoltz, G. (2023, February). On best-arm identification with a fixed budget in non-parametric multi-armed bandits. In International Conference on Algorithmic Learning Theory (pp. 136-181). PMLR.
> >
> > [10] Lalitha, A. L., Kalantari, K., Ma, Y., Deoras, A., and Kveton, B. (2023, July). Fixed-Budget Best-Arm Identification with Heterogeneous Reward Variances. In Uncertainty in Artificial Intelligence (pp. 1164-1173). PMLR.

---

### Meta-Review · Area_Chair_oYxT · 2023-12-06

**Metareview:**

This paper studies best arm identification (BAI) in linear bandits with differential privacy (DP) constraint. Here there are $k$ arms with feature vectors $a_1, \\dots, a_K \\in \\mathbb{R}^d$ where the reward in each round is $x_i = a_i \\cdot \\theta$ for some unknown $\theta$; the differential privacy constraint is with respect to changing a single reward value. The goal is to output the arm with largest (expected) reward with as high a probability as possible.

The paper provides an algorithm and a lower bound for the problem. As usual, the algorithm keeps a list of active arms and gradually reduces this list. The main innovation is that, instead of querying all arms (and add noise to their rewards to achieve DP), the authors show that, when the number of active arms is larger than $\\approx d^2$, it suffices to query only a small subset of just $d$ arms; this subset is selected by maximizing the determinant of the submatrix corresponding to the selected arms. The rewards of other arms can be deduced from these by taking appropriate linear combinations. The authors give an analysis of this algorithm where the probabilistic error bound depends on the appropriate notion of (instance-wise) gap. The authors then give a lower bound that, for regime of constant $\\epsilon > 0$, matches the upper bound to within roughly a squared factor in the exponent of the error probability. Finally, the authors provide empirical evaluation on synthetic data; the algorithm is shown to be much superior compared to adding noise trivially to the best known non-private algorithm (Yang and Tan, NeurIPS 2022), and the algorithm nearly matches the non-private algorithm already for moderate value of $\\epsilon$ (between $1, 2$).

## Strengths

- BAI is a natural setting and this paper is the first paper to study BAI with DP constraints. (Previous DP bandit papers focus on regret minimization.)

- As far as I can tell, the idea of using determinant maximization is novel in the context of BAI.

- The paper provides rigorous analysis of upper and lower bounds that nearly matches for the most interesting regime of $\\epsilon > 0$.

- The paper also provides clear explanation and analysis on why trivially adding noise to the best known non-private algorithm (Yang and Tan, NeurIPS 2022) leads to a worst algorithm than the one proposed here.

- The algorithm is simple and practical; this is shown by the empirical results in the paper.

## Weaknesses

- The performance of the algorithm still does not match the best known non-private algorithm as we take $\\epsilon \\to \\infty$.

- The writing can be improved. E.g. algorithms are not explained in words and most analyses are given in the appendix without much intuition given in the main body.

**Justification For Why Not Higher Score:**

Given that the performance of the algorithm still does not match the best known non-private algorithm as we take $\\epsilon \\to \\infty$, it is likely that this is not yet the best approach to tackle the problem.

**Justification For Why Not Lower Score:**

As stated above, this is a natural and well-studied setting in bandits literature that is somehow overlooked by the DP community so far. I think this is a good first paper for the topic of DP-BAI, which provides some initial (and innovative) results on the problem. The analyses are rigorous and, as far as I can tell, seem correct.

---

### Decision · Program_Chairs · 2024-01-16

Accept (poster)